# Fine-Grained Theoretical Analysis of Federated Zeroth-Order Optimization

**Jun Chen**[1], **Hong Chen**[1,2,5,6]*, **Bin Gu**[3,4], **Hao Deng**[1]*

[1]College of Informatics, Huazhong Agricultural University, China
[2]Engineering Research Center of Intelligent Technology for Agriculture, China
[3]School of Artificial Intelligence, Jilin University, China
[4]Mohamed bin Zayed University of Artificial Intelligence
[5]Shenzhen Institute of Nutrition and Health, Huazhong Agricultural University, China
[6]Shenzhen Branch, Guangdong Laboratory for Lingnan Modern Agriculture,
Genome Analysis Laboratory of the Ministry of Agriculture,
Agricultural Genomics Institute at Shenzhen, Chinese Academy of Agricultural Sciences, China
`cj850487243@163.com`, `chenh@mail.hzau.edu.cn`,
`jsgubin@gmail.com`, `dengh@mail.hzau.edu.cn`

## Abstract

Federated zeroth-order optimization (FedZO) algorithm enjoys the advantages of both zeroth-order optimization and federated learning, and has shown exceptional performance on black-box attack and softmax regression tasks. However, there is little generalization analysis for FedZO, and its analysis on computing convergence rate is slower than the corresponding first-order optimization setting. This paper aims to establish systematic theoretical assessments of FedZO by developing the analysis technique of on-average model stability. We establish the first generalization error bound of FedZO under the Lipschitz continuity and smoothness conditions. Then, refined generalization and optimization bounds are provided by replacing bounded gradient with heavy-tailed gradient noise and utilizing the second-order Taylor expansion for gradient approximation. With the help of a new error decomposition strategy, our theoretical analysis is also extended to the asynchronous case. For FedZO, our fine-grained analysis fills the theoretical gap on the generalization guarantees and polishes the convergence characterization of the computing algorithm.

## 1 Introduction

Federated learning collaborates multiple local clients to train a global model without sharing local raw data, which often enjoys great ability in protecting data privacy [1]. The core training steps of federated learning include local clients receiving the global model from the central server, the local models being updated by the global information and local data, and the updated local models being uploaded to renew the global model. Based on this building-block process, rich federated learning algorithms have been formulated to match different motivations, where their properties on privacy protection [2, 3, 4, 5] and convergence rate [6, 7, 8] are investigated. In general, the existing algorithms of federated learning depend heavily on the gradient information of loss function, see e.g., [1, 6, 7, 8]. Indeed, there are some learning scenarios where the gradient or Hessian information

---

*Corresponding authors.

37th Conference on Neural Information Processing Systems (NeurIPS 2023).

is either unobtainable or too expensive to obtain, such as reinforcement learning [9, 10, 11], meta learning [12], black-box adversarial attacks [13, 14] and hyperparameter tuning [15]. To alleviate the dependence on the gradient or Hessian information, a federated zeroth-order optimization (FedZO) algorithm is proposed in [16] by integrating zeroth-order optimization into federated learning. In theory, the analysis of optimization convergence of FedZO has been established in [16], which shows the linear speedup in terms of the numbers of local iterations and clients participating in the update for the global iteration. However, there is little generalization analysis for federated zeroth-order optimization.

The goal of generalization analysis is to evaluate the capability of the empirical risk to approach the theoretical optimal risk on a given dataset. To characterize the generalization performance theoretically, there are four popular analysis techniques including uniform convergence approaches [17, 18, 19, 20], operator approximation techniques [21, 22], information-theoretic tools [23, 24, 25, 26, 27], and algorithmic stability analysis [28, 29, 30, 31]. Usually, stability-based generalization assessment enjoys some benefits over other tools. Firstly, in contrast to uniform convergence approaches, the theoretical guarantees derived from algorithmic stability analysis are independent of the capacity measurement of hypothesis function space [32]. Secondly, algorithmic stability analysis is available in a wide range of applications rather than only some models enjoying operator representation [33, 34]. Finally, data distribution does not affect the results of algorithmic stability analysis, whereas information-theoretic tools are typically sensitive to data distribution [25, 35]. To the best of our knowledge, there is only one work on the stability-based generalization analysis for the zeroth-order optimization, i.e., zeroth-order stochastic search (ZoSS) method [36]. Following this line, it is natural to further investigate the generalization guarantees of the FedZO algorithm. However, due to the essential difference between FedZO and ZoSS, the previous analysis technique in [36] can not be used for federated learning directly. In this paper, we develop the fine-grained error decomposition and estimations to overcome the challenge induced by the federated algorithmic formulation.

As a general assumption for loss function, Lipschitz continuity has been employed in many stability-based generalization assessments, see e.g., [36, 37, 38, 39, 40]. However, the index of Lipschitz continuity is likely too large or even infinite for some learning problems, which makes the previous results invalid [32]. With the help of the on-average model stability tool, [32] established the fine-grained generalization analysis of stochastic gradient descent (SGD) by removing the Lipschitz continuity and the convexity of each loss function, and relaxing the smoothness to Holder continuity. Moreover, [38] additionally considered the bounded variance of the stochastic gradient to get the generalization bounds of non-convex SGD with high probability. Inspired by [32] and [38], this paper considers the heavy-tailed gradient noise (Assumption 2) as a refined version of bounded variance of gradient and adopts the second-order Taylor expansion for gradient approximation to remove bounded gradient condition. Conclusively, we fill the theoretical gap in the generalization analysis of the FedZO algorithm [16] and its asynchronous version. The main contributions of this paper are outlined as follows.

- *Generalization bounds of FedZO*. We provide the first generalization bound of general FedZO after building the relationships between generalization error and $\ell_1$ on-average model stability. To alleviate the restriction of bounded gradient, we further get a refined generalization bound and an optimal optimization bound under the condition of heavy-tailed gradient noise and the second-order Taylor expansion of gradient approximation.

- *Learning guarantees of asynchronous FedZO*. For asynchronous FedZO, we design a new error decomposition strategy to bridge the relationships between each local model parameter and the global model parameter in each iteration. Then, the generalization and optimization bounds are derived for the asynchronous case. In particular, our fine-grained error bounds are tight even compared with the previous results for SGD implemented by the first-order optimization [32, 37, 38] and zeroth-order optimization [36].

## 2   Preliminaries

This section introduces some notations and definitions preparing for our theoretical analysis. Besides, we also give some detailed explanations for every symbol in *Appendix A*.

Considering a zeroth-order optimization algorithm for federated learning, we rely on $N$ clients to independently train a global model. In each iteration, a subset of $M$ clients is randomly selected to use their local data to obtain the corresponding gradient information. Then, this information is transmitted to the central server to update the global model parameter. For any $i \in [N]$, let $z_i$ be a random variable obeying data distribution $\mathcal{D}_i$ associated with the $i$-th client. Denote $S := \{S_i\}_{i=1}^N$ as the total dataset, where $S_i := \{z_{i1}, ..., z_{in}\}$ is the $i$-th local dataset and each sample $z_{ij}$ is drawn independently from the distribution $\mathcal{D}_i, i \in [N]$.

Federated learning aims to optimize model parameters with the collaboration among all local clients, i.e., minimizing the following population risk

$$F(w) := \frac{1}{N} \sum_{i=1}^N F_i(w_i) = \frac{1}{N} \sum_{i=1}^N \mathbb{E}_{z_i \sim \mathcal{D}_i} \left[ f_i(w_i; z_i) \right], \tag{1}$$

where $w, w_i \in \mathcal{W} \in \mathbb{R}^d$ denotes the training parameters for the global model and the $i$-th local model in a $d$-dimensional hypothesis space respectively, $F(w)$ and $F_i(w_i)$ indicate the population risks for the global model and the local model of the $i$-th client respectively, $f_i$ denotes the loss function of the $i$-th local client, and $\mathbb{E}_{z_i \sim \mathcal{D}_i}[\cdot]$ denotes the conditional expectation with respect to (w.r.t.) the sample $z_i$. Generally, the unattainability of the local population risk $F_i(w_i)$ forces us to train the model by minimizing the following empirical approximation of population risk $F(w)$

$$F_S(w) := \frac{1}{N} \sum_{i=1}^N F_{S_i}(w_i) = \frac{1}{nN} \sum_{i=1}^N \sum_{j=1}^n f_i(w_i; z_{ij}), \tag{2}$$

where $F_S(w)$ and $F_{S_i}(w_i)$ indicate the empirical risks for the global model and the local model of the $i$-th client respectively. Note that, for the update of the global model, the contribution of each client is treated equally. This paper considers the learning scenarios where the gradients (or Hessian information) of local loss functions are either unobtainable or too expensive to obtain [16]. Naturally, the first-order gradient estimation $\tilde{\nabla} f_i$, defined as, for any $t, b_2 \in \mathbb{N}, m \in [b_1], b_1 \in [n]([n] := \{1, ..., n\})$,

$$\tilde{\nabla} f_i \left( w_i^t; z_{i,m}^t, \{v_{i,l}^t\}_{l=1}^{b_2}, \mu \right) = \frac{1}{b_2} \sum_{l=1}^{b_2} \frac{v_{i,l}^t}{\mu} \left( f_i \left( w_i^t + \mu v_{i,l}^t; z_{i,m}^t \right) - f_i \left( w_i^t; z_{i,m}^t \right) \right),$$

is chosen to update the model parameters of federated learning, where $w_i^t$ is the local model parameter of the $i$-th client at the $t$-th iteration, $\left\{ z_{i,m}^t \right\}_{m=1}^{b_1}$ and $\left\{ v_{i,l}^t \right\}_{l=1}^{b_2}$ are two sets of independent and identically distributed (i.i.d.) random samples and i.i.d. random direction vectors (satisfying the $d$-dimensional uniform distribution), and $\mu$ represents the distance between two model parameters ($w_i^t + \mu v_{i,l}^t$ and $w_i^t$) used to estimate gradient in the $l$-th direction. In our analysis, the following second-order Taylor expansion is employed to approximate $\tilde{\nabla} f_i$,

$$\tilde{\nabla} f_i \left( w_i^t; z_{ij}^t \right) = \frac{1}{b_2} \sum_{l=1}^{b_2} \left( \langle \nabla f_i(w_i^t; z_{i,j}^t), v_{i,l}^t \rangle v_{i,l}^t + \left( \frac{\mu}{2} (v_{i,l}^t)^\top \nabla_{w_i}^2 f_i(w_i; z_{ij}^t) |_{w_i = w_{i,l}^{t*}} v_{i,l}^t \right) v_{i,l}^t \right).$$

Note that, the number $b_2$ of random direction vectors is set to be greater than the hypothesis space dimension $d$ in this paper, such as the requirement of [13]($b_2 = 2d$).

The update process of FedZO is formulated as follows,

$$w^{t+1} = w^t - \frac{\eta_t}{b_1 M} \sum_{i \in \mathcal{M}_t} \sum_{m=1}^{b_1} \tilde{\nabla} f_i \left( w_i^t; z_{i,m}^t, \{v_{i,l}^t\}_{l=1}^{b_2}, \mu \right), \quad w_i^t = w^t, \tag{3}$$

where $\eta_t$ and $\mathcal{M}_t \in [N](|\mathcal{M}_t| = M)$ denote the step size and the collection of selected client indices in the $t$-th iteration, respectively. In particular, we also consider the asynchronous FedZO algorithm.

Let

$$w^* \in \arg \min_{w \in \mathcal{W}} F(w) \text{ and } w(S) \in \arg \min_{w \in \mathcal{W}} F_S(w), \tag{4}$$

where $F(w), F_S(w)$ are defined in (1) and (2). Denote $A(S) = w^T$ as the output of the global model after $T$ iterations with any federated learning algorithm $A$ (including Algorithm 1, i.e.,

synchronous FedZO). Typically, the excess risk of $A(S)$ is measured by $\mathbb{E}[F(A(S)) - F(w^*)]$ and can be decomposed as

$$\mathbb{E}[F(A(S)) - F(w^*)] \le \mathbb{E}[F(A(S)) - F_S(A(S))] + \mathbb{E}[F_S(A(S)) - F_S(w(S))], \quad (5)$$

where $\mathbb{E}[\cdot]$ denotes the expectation w.r.t. all randomness. The first part of (5) (called generalization error) measures the expected gap between population risk and empirical risk, while the second part of (5) (called optimization error) measures the divergence between the trained model and the theoretically optimal model. In this paper, we simultaneously bound the generalization error and optimization error of FedZO.

**Remark 1.** *The update strategy* (3) *is associated with a minibatch of i.i.d. samples. Due to the interaction among samples, it is hard to establish the stability-based generalization analysis of the minibatch case directly. Fortunately, an equivalent formula can tackle this difficulty by means of the properties of binomial distribution (see* Appendix B.3*).*

**Remark 2.** *Compared with synchronous FedZO ([16] in Algorithm 1), there are several essential differences for its asynchronous one described in Section 4. Firstly, we assume that all clients participate throughout the entire update process of the global model parameters. Secondly, once some client finishes gradient computation, this gradient will be transmitted to update the global model immediately without waiting for other clients. Then, the updated parameters of the global model will be back to the corresponding client, not the other ones. As a result, the local model of some client may be inconsistent with other clients within the same iteration.*

---

**Algorithm 1** Synchronous FedZO

---

**Require:** $w^1$: initial global model; $\eta_1$: initial learning rate; $b_1, b_2$: minibatch sizes for samples and direction vectors respectively; $\mu$: positive step size in the definition of the derivative; $M$: number of clients selected to update the global model in each iteration
    **for all** $t = 1, ..., T - 1$ **do**
        Randomly select a clients set $\mathcal{M}_t$, let $\eta_t = \eta_1/t$
        **for all** $i \in \mathcal{M}_t$ **in parallel do**
            Let $w_i^t = w^t$
            Generate $\{z_{i,m}^t\}_{m=1}^{b_1}$ and $\{v_{i,l}^t\}_{l=1}^{b_2}$ from $\mathcal{D}_i$ and $d$-dimensional uniform distribution
            Compute $\sum_{m=1}^{b_1} \tilde{\nabla} f_i$ and upload it to global model
        **end for**
        Update $w^t$ to $w^{t+1}$ by Eq. (3)
    **end for**
**Ensure:** Final global model $w^T$

---

Inspired by Definition 4 in [32], we introduce a new definition of $\ell_1$ on-average model stability for federated learning.

**Definition 1.** *The federated learning algorithm $A$ is $\ell_1$ on-average model $\epsilon$-stable if*

$$\mathbb{E}\left[\frac{1}{nN}\sum_{i=1}^{N}\sum_{j=1}^{n}\left\|A(S^{(j_i)}) - A(S)\right\|\right] \le \epsilon, \quad (6)$$

*where $\|\cdot\|$ is the Euclidean distance $\|\cdot\|_2$, $S = \{S_i\}_{i=1}^{N}$, $S^{(j_i)} = \left\{S_1, ..., S_{i-1}, S_i^{(j)}, S_{i+1}, ..., S_N\right\}$, $S_i = \{z_{i1}, ..., z_{in}\}$, $S_i^{(j)} = \left\{z_{i1}, ..., z_{i(j-1)}, z_{ij}', z_{i(j+1)}, ..., z_{in}\right\}$ ($z_{ij}'$ is drawn from $\mathcal{D}_i, \forall i \in [N], j \in [n]$).*

In Definition 1, we assume that only one sample of one client is perturbed, not other clients. Particularly, as $N = 1$, the current definition is consistent with the standard on-average model stability in [32, 38]. Compared with several other stability tools, the advantages of on-average model stability are listed as follows. First, on-average model stability is a weaker stability tool than uniform (model) stability. Second, on-average model stability measures the stability of model parameters $w$ instead of function values $f(w)$ like on-average stability, which can improve our analysis.

**Definition 2.** *[41] For some $\theta, a, b > 0$ and all $x > 0$, if a random variable $X$ satisfying*

$$\mathbb{P}\left(|X| \geq x\right) \leq a \exp(-bx^{\frac{1}{\theta}}),$$

*we call it sub-Weibull random variable (denoted as $X \sim subW(\theta)$), where $\theta$ is a tail parameter.*

Sub-Weibull distribution is a heavier-tailed (longer-tailed) distribution than the sub-Gaussian $\left(\theta = \frac{1}{2}\right)$ and sub-Exponential $(\theta = 1)$ distribution, which matches some real applications [42] and has been used for learning theory analysis recently [43, 44]. This paper utilizes some sub-Weibull properties, derived from the moment generating function (MGF), to relax the bounded gradient assumption used in [36, 37, 38, 39, 40].

## 3 Evaluating the Generalization of FedZO via Stability

This section establishes the relationships between the generalization error of FedZO and the on-average model stability in Theorem 1. Detailed proofs are provided in *Appendix B.2*.

This paper makes the following assumptions for the loss function $f_i$ of $i$-th local client ($i \in [N]$).

**Assumption 1.** *(a) (L-Lipschitz continuity). For any $w_i, \bar{w}_i \in \mathcal{W}, i \in [N]$ and some $L > 0$, $f_i$ satisfies $\nabla f_i(w_i; z_i) \leq L$, that is*

$$|f_i(w_i; z_i) - f_i(\bar{w}_i; z_i)| \leq L \left\| w_i - \bar{w}_i \right\|, \ \ \forall z_i \sim \mathcal{D}_i.$$

*(b) ($\beta$-Smoothness). For any $w_i, \bar{w}_i \in \mathcal{W}, i \in [N]$ and some $\beta > 0$, $f_i$ satisfies*

$$\left\| \nabla f_i(w_i; z_i) - \nabla f_i(\bar{w}_i, z_i) \right\| \leq \beta \left\| w_i - \bar{w}_i \right\|, \ \ \forall z_i \sim \mathcal{D}_i.$$

The requirement of Lipschitz continuity and smoothness is general in statistical learning theory, see e.g., [37, 38, 39, 40]. Usually, it appears to be justifiable when the hypothesis function space is uniformly bounded. However, the Lipschitz constant $L$ may be infinite for some learning tasks with unbounded hypothesis function space [32]. In this paper, two attempts are proposed to remove the Lipschitz assumption. Firstly, we focus on the zeroth-order optimization problem, where the gradient information is unavailable. We approximate the first-order gradient using a second-order Taylor expansion to avoid the dependence on the Lipschitz constant (see the proofs of Theorems 2, 3 and 5). Secondly, we use a weaker gradient-related assumption, i.e., the heavy-tailed gradient noise assumption, to replace the Lipschitz assumption in our generalization analysis (see the proof of Theorem 1 (b)).

**Assumption 2.** *(Heavy-tailed gradient noise). Let the tail parameter $\theta > \frac{1}{2}$, the number of iterations $t > 0$ and the client index $i \in [N]$. For any $w_i^t \in \mathcal{W}, z_i^t \in S_i, S_i \in \mathcal{D}_i^n$, we assume that the gradient noise $\nabla f_i(w_i^t; z_i^t) - \nabla F_{S_i}(w_i^t)$ is a sub-Weibull random vector, i.e., $\nabla f_i(w_i^t; z_i^t) - \nabla F_{S_i}(w_i^t) \sim subW(\theta)$.*

From Theorem 2.1 in [41], we deduce that the MGF of $\| \nabla f_i(w_i^t; z_i^t) - \nabla F_{S_i}(w_i^t) \|^{\frac{1}{\theta}}$ can be bounded at some point, i.e., $\mathbb{E}\left[ \exp\left( \frac{\| \nabla f_i(w_i^t; z_i^t) - \nabla F_{S_i}(w_i^t) \|}{K} \right)^{\frac{1}{\theta}} \right] \leq 2$ for some $K > 0$. Moreover, a random variable $X$ is defined as $K$-sub-Weibull($\theta$) if $\mathbb{E}\left[ \exp\left( |X|/K \right)^{1/\theta} \right] \leq 2$ [45]. Therefore, the gradient noise in Assumption 2 can be also denoted as $\nabla f_i(w_i^t; z_i^t) - \nabla F_{S_i}(w_i^t) \sim subW(\theta, K)$. In this paper, a lemma (Lemma 2 in *Appendix B.1*) related to the bounded $p$-th norm of the sub-Weibull variable is used to build a novel relationship (Theorem 1 (b)) between $\ell_1$ on-average model stability and generalization error.

**Assumption 3.** *(PL condition). For any $w \in \mathcal{W}, S \in \bigcup_{i=1}^{N} \mathcal{D}_i^n$ and some $\alpha > 0$, the empirical risk $F_S(w)$ satisfies*

$$\mathbb{E}\left[ \|\nabla F_S(w)\|^2 \right] \geq 2\alpha \mathbb{E}\left[ (F_S(w) - F_S(w(S))) \right].$$

Assumption 3 elucidates that the quadratic gradient of the empirical risk enjoys a linearly decreasing lower bound [46]. Without the setting of convexity, the gradient of empirical risk $\|\nabla F_S(w)\| = 0$ is just a sufficient condition for a local optimal model instead of the guarantee to find a global optimal

parameter. This assumption implies that all local optimal parameters are global optimal parameters, that is, $\|\nabla F_S(w)\| = 0$ implies that $\mathbb{E}[F_S(w)] = \mathbb{E}[F(w(S))] = \mathbb{E}[F(w^*)]$ holds in expectation. For this reason, we can study the optimization error with the form $F_S(w) - F_S(w(S))$ rather than $|\nabla F_S(w)|$ [44] under the non-convex condition.

Now we state two quantitative relationships between generalization error and $\ell_1$ on-average model stability.

**Theorem 1.** *Let $S, S^{(j_i)}$ be given in Definition 1 and $A$ be a federated learning algorithm.*

*(a) Let Assumption 1 (a) hold and $A$ be $\ell_1$ on-average model $\epsilon$-stable. Then, for some $L > 0$,*

$$|\mathbb{E}[F(A(S)) - F_S(A(S))]| \leq \frac{L}{nN} \sum_{i=1}^{N} \sum_{j=1}^{n} \mathbb{E}\left[\left\|A(S^{(j_i)}) - A(S)\right\|\right] \leq L\epsilon.$$

*(b) Let Assumption 2 hold and $A$ be $\ell_1$ on-average model $\epsilon$-stable. Then, for some $\theta > \frac{1}{2}, K > 0$,*

$$|\mathbb{E}[F(A(S)) - F_S(A(S))]| \leq \frac{(4\theta)^{\theta} K}{nN} \sum_{i=1}^{N} \sum_{j=1}^{n} \mathbb{E}\left[\left\|A(S^{(j_i)}) - A(S)\right\|\right] + 2\mathbb{E}[F_S(A(S))]$$

$$\leq (4\theta)^{\theta} K\epsilon + 2\mathbb{E}[F_S(A(S))].$$

Theorem 1 provides the generalization bounds of the FedZO algorithm by the $\ell_1$ on-average model stability. Essentially, Theorem 1 (a) is consistent with Theorem 2 (a) in [32] and Theorem 1 (b) states a refined upper bound independent of the Lipschitz constant $L$. Explicitly, in Theorem 1 (b), the tail parameter $\theta$ in the first term is typically bounded [41] and the second term $\mathbb{E}\left[F_S(A(S))\right]$ has no significant negative impact on the upper bound [32].

In the sequel, we derive the generalization bounds by integrating Theorem 1 and estimations of $\ell_1$ on-average model stability. Meanwhile, some optimization bounds for FedZO are also provided. Table 1 and Table 2 report the comparisons of our results with the related theoretical analysis for ZoSS [36], AD-SGD [47], AD-PSGD [48], EF-ZO-SGD, FED-ZO-SGD [49], FedZO [16] and SGD [37, 38, 39, 40, 50] without the convexity requirement of loss function.

### 3.1 Generalization Analysis of General FedZO

This subsection considers the FedZO under the general setting.

**Theorem 2.** *Let $\{w^t\}$ and $\{\bar{w}^t\}$ be produced by FedZO (3) on $S$ and $S^{(j_i)}$ respectively, where $\eta_t = \eta_1 t^{-1}$, $\eta_1 \leq (2a_1)^{-1}$ with $a_1 = \left(1 + \sqrt{d/b_2}\right)\beta$. After $T$ iterations, we get the global parameters $A(S) = w^T$ and $A(S^{(j_i)}) = \bar{w}^T$ respectively. Under Assumption 1, there holds*

$$\frac{1}{nN} \sum_{i=1}^{N} \sum_{j=1}^{n} \mathbb{E}\left[\left\|w^T - \bar{w}^T\right\|\right] \leq (e(T-1))^{a_1\eta_1} a_2 \eta_1 \log(e(T-1)),$$

*where $a_2 = 2(nN)^{-1}L + \mu\beta + 2(nN)^{-1}L\sqrt{d/b_2}$.*

Theorem 2 states a $\ell_1$ on-average model stability bound with order $\mathcal{O}\left(\left((nN)^{-1}L + \mu\right)T^{\frac{1}{2}}\log T\right)$ under mild conditions of parameter selection. When $\mu = \mathcal{O}\left((nN)^{-1}\right)$, the upper bound is equal to $\mathcal{O}\left((nN)^{-1}LT^{\frac{1}{2}}\log T\right)$, which is comparable with the existing stability bounds for ZoSS [36] and SGD [37, 39, 40, 51, 52] under similar choices of step sizes.

The following corollary is derived by integrating Theorem 1 (a) and Theorem 2.

**Corollary 1.** *Under the same conditions of Theorem 2, for FedZO (3), there holds*

$$|\mathbb{E}[F(w^T) - F_S(w^T)]| \leq \mathcal{O}\left(L\left((nN)^{-1}L + \mu\right)T^{\frac{1}{2}}\log T\right).$$

When $\mu = \mathcal{O}\left((nN)^{-1}\right)$, it provides the generalization bound $\mathcal{O}\left((nN)^{-1}L^2 T^{\frac{1}{2}}\log T\right)$ for the general FedZO algorithm. When $\beta c/(\beta c + 1) \geq \frac{1}{2}$ for some positive constant $c$, Corollary 1

Table 1: Comparisons of stability-based generalization bounds under the non-convex condition (Thm.-Theorem; Cor.-Corollary; $c$-a positive constant; $v^2$-the upper bound of the variance of gradient; $\lambda$-a parameter characterizing the properties of decentralized topology; $\delta$-high probability; $*$-high probability bound; B.-bounded loss function; Uni.-uniform stability; $\ell_1$-$\ell_1$ on-average model stability; $\sqrt{}$-has such a property; $\times$-hasn't such a property; $\Gamma_{b_2}^d = \left(\sqrt{(3d-1)/b_2}+1\right)$; $\hat{a}(N,T,t_0) = \left(1+\sqrt{\log T}/(nN)\right)\sqrt{\log T}+t_0)$.

| Algorithm Result | Generalization bound | Tool | Assumptions | | | |
|---|---|---|---|---|---|---|
| | | | $L$ | $\theta$ | $v^2$ | B. |
| SGD ($\eta_t \leq c/t$) [37] (Thm. 3.12) | $\mathcal{O}\left(\frac{L^{\frac{2}{\beta c+1}}T^{\frac{\beta c}{\beta c+1}}}{n}\right)$ | Uni. | $\sqrt{}$ | $\times$ | $\times$ | $\sqrt{}$ |
| SGD ($\eta_t \leq \frac{c}{(t+2)\log(t+2)}$) [38] (Thm. 3) | $*\mathcal{O}\left(\sqrt{\frac{L\left(\sqrt{\mathbb{E}[v^2]}+\log T\right)}{n\delta}}\right)$ | $l_1$ | $\sqrt{}$ | $\times$ | $\sqrt{}$ | $\sqrt{}$ |
| SGD ($\eta_t \leq c/t$) [39] (Thm. 3.5) | $\mathcal{O}\left(\frac{L^{\frac{2}{\beta c+1}}T^{\frac{\beta c}{\beta c+1}}}{n}\right)$ | Uni. | $\sqrt{}$ | $\times$ | $\times$ | $\sqrt{}$ |
| SGD ($\eta_t \leq \frac{2t+1}{2\alpha(t+1)^2}$) [40] (Thm. 15) | $\mathcal{O}\left(\frac{T^{\frac{\beta}{\beta+\alpha}}}{n}\right)$ | Uni. | $\sqrt{}$ | $\times$ | $\times$ | $\sqrt{}$ |
| ZoSS ($\eta_t \leq c/\left(t\Gamma_{b_2}^d\right)$) [36] (Thm. 5) | $\mathcal{O}\left(\frac{L^{\frac{2}{\beta c+1}}T^{\frac{\beta c}{\beta c+1}}}{n}\right)$ | Uni. | $\sqrt{}$ | $\times$ | $\times$ | $\sqrt{}$ |
| ZoSS ($\eta_t \leq c/t$) [36] (Cor. 6) | $\mathcal{O}\left(\frac{L^2 T}{n}\right)$ | Uni. | $\sqrt{}$ | $\times$ | $\times$ | $\sqrt{}$ |
| ZoSS ($\eta_t \leq c/(t\Gamma_{b_2}^d)$) [36] (Thm. 8) | $\mathcal{O}\left(\frac{L^2 T^{\beta c}}{n}\min\left\{c+\beta^{-1}, c\log(eT)\right\}\right)$ | Uni. | $\sqrt{}$ | $\times$ | $\times$ | $\times$ |
| ZoSS ($\eta_t \leq c/(T\Gamma_{b_2}^d)$) [36] (Thm. 7) | $\mathcal{O}\left(\frac{L^2}{n}\right)$ | Uni. | $\sqrt{}$ | $\times$ | $\times$ | $\times$ |
| AD-SGD ($\eta_t = \eta_1$) [47] (Cor. 2) | $\mathcal{O}\left(\frac{n\eta_1-\lambda}{n(1-\lambda)}L^2\left(1+\frac{\beta\eta_1}{M}\right)^T\right)$ | Uni. | $\sqrt{}$ | $\times$ | $\times$ | $\times$ |
| AD-SGD ($\eta_t = \frac{M}{\beta(t+1)}$) [47] (Cor. 2) | $\mathcal{O}\left(\frac{nM-\lambda}{n(1-\lambda)}L^2 T\right)$ | Uni. | $\sqrt{}$ | $\times$ | $\times$ | $\times$ |
| FedZO ($\eta_t \leq \eta_1/t$) Ours (Cor. 1) | $\mathcal{O}\left(\left(\frac{L}{nN}+\mu\right)LT^{\frac{1}{2}}\log T\right)$ | $\ell_1$ | $\sqrt{}$ | $\times$ | $\times$ | $\times$ |
| FedZO ($\eta_t \leq \frac{\eta_1}{t}$) Ours (Cor. 2) | $\mathcal{O}\left(\left(\frac{\sqrt{\log T}}{nN}+1\right)(4\theta)^\theta \mu T^{\frac{1}{4}}\log T\right)$ | $\ell_1$ | $\times$ | $\sqrt{}$ | $\times$ | $\times$ |
| FedZO ($\eta_t \leq \frac{\eta_1}{t}$) Ours (Cor. 3) | $\mathcal{O}\left(\hat{a}(N,T,t_0)(4\theta)^\theta \mu T^{\frac{1}{2}}\sqrt{\log T}\right)$ | $\ell_1$ | $\times$ | $\sqrt{}$ | $\times$ | $\times$ |

aligns with the previous generalization bounds of SGD algorithms for pointwise learning (Theorem 3.12 in [37]) and pairwise learning (Theorem 3.5 in [39] and Theorem 15 in [40]). [36] provided the first generalization error analysis for the minibatch ZoSS algorithm with both unbounded and bounded non-convex loss functions. Specifically, they presented the generalization bound $n^{-1}(2+C)L^2(eT)^{\beta c}\min\left\{c+\beta^{-1}, c\log(eT)\right\}$ ($C$ is a positive constant) with the decreasing step size $\eta_t \leq c/\left(t\Gamma_{b_2}^d\right)$, where $\Gamma_{b_2}^d = \left(\sqrt{(3d-1)/b_2}+1\right)$. Under the constant step sizes $\eta_t \leq \log(1+\beta c)/\left(T\beta\Gamma_{b_2}^d\right)$ and $\eta_t \leq c/\left(T\Gamma_{b_2}^d\right)$, they also showed the generalization bound $n^{-1}(2+C)cL^2$ and $(n\beta)^{-1}L^2(2+C)\left(e^{\beta c}-1\right)$. When $\beta c \geq \frac{1}{2}$, our generalization bound can match their first result. Indeed, we also can get similar bounds as [36] for the special setting of constant step size. Detail comparisons are also provided in Table 1.

## 3.2 Learning Guarantees of FedZO with Heavy tails

Inspired by [43, 44], we further investigate the learning guarantees of FedZO with the smooth, heavy-tailed loss function and PL condition.

Table 2: Comparisons of optimization bounds under the non-convex condition (Thm.-Theorem; Cor.-Corollary; D-ZO-PD (Distributed ZO Primal-Dual); $r, C_\lambda$-some constants; $\lambda$-a parameter characterizing the properties of decentralized topology; $\sigma$-the upper bound of the square of gradient; B.-bounded loss function; $\sqrt{}$-has such a property; $\times$-hasn't such a property).

| Algorithm Result | Optimization bound | Step size | Assumptions | | | | |
|---|---|---|---|---|---|---|---|
| | | | $L$ | $\theta$ | $\beta$ | B. | $\sigma$ |
| AD-SGD [47] (Thm. 8) | $\mathcal{O}\left(\left(r + \frac{C_\lambda}{\lambda t_0} + \frac{t_0}{M}\right)(\log T)^{-1}\right)$ | $\eta_t = \mathcal{O}\left(\frac{M}{t+1}\right)$ | $\checkmark$ | $\times$ | $\checkmark$ | $\times$ | $\times$ |
| AD-PSGD [48](Cor. 2) | $\mathcal{O}\left(T^{-\frac{1}{2}}\right)$ | $\eta_t = \mathcal{O}\left(\frac{n}{b_1(\sqrt{T}+1)}\right)$ | $\times$ | $\times$ | $\checkmark$ | $\times$ | $\checkmark$ |
| EF-ZO-SGD [49](Thm. 1) | $\mathcal{O}\left(T^{-\frac{1}{2}}d^{\frac{1}{2}} + T^{-1}d\right)$ | $\eta_t = \mathcal{O}\left(\sqrt{\frac{1}{dT}}\right)$ | $\checkmark$ | $\times$ | $\checkmark$ | $\times$ | $\times$ |
| FED-EF -ZO-SGD [49](Thm. 2) | $\mathcal{O}\left(T^{-\frac{1}{2}}d^{\frac{1}{2}} + T^{-\frac{3}{2}}d^{\frac{3}{2}}\right)$ | $\eta_t = \mathcal{O}\left(\sqrt{\frac{1}{dT}}\right)$ | $\checkmark$ | $\times$ | $\checkmark$ | $\times$ | $\times$ |
| FedZO [16] (Cor. 2) | $\mathcal{O}\left((MT)^{-\frac{1}{2}} + d\mu^2\right)$ | $\eta_t \leq \sqrt{\frac{Mb_1b_2}{dT}}$ | $\times$ | $\times$ | $\checkmark$ | $\times$ | $\checkmark$ |
| FedZO Ours (Thm. 4) | $\mathcal{O}\left(T^{-2} + \mu^2\right)$ | $\eta_t \leq \frac{\eta_1}{t}$ | $\times$ | $\checkmark$ | $\checkmark$ | $\times$ | $\times$ |
| FedZO Ours (Thm. 6) | $\mathcal{O}\left(T^{-2} + \mu^2\right)$ | $\eta_t \leq \frac{\eta_1}{t}$ | $\times$ | $\checkmark$ | $\checkmark$ | $\times$ | $\times$ |

**Theorem 3.** *Let $\{w^t\}$ and $\{\bar{w}^t\}$ be produced by FedZO (3) on $S$ and $S^{(j_i)}$ respectively, where $\eta_t = \eta_1 t^{-1}, \eta_1 \leq \frac{1-d/b_2}{4a_3}$ with $a_3 = (1 + d/b_2)\beta$, and $0 \geq \frac{1-d/b_2}{2}\alpha\eta_1 - 1$. After $T$ iterations, we get the global parameters $A(S) = w^T$ and $A(S^{(j_i)}) = \bar{w}^T$ respectively. Under Assumptions 1 (b), 2 and 3, there holds*

$$\frac{1}{nN}\sum_{i=1}^{N}\sum_{j=1}^{n}\mathbb{E}\left[\left\|w^T - \bar{w}^T\right\|\right] \leq \left(e\,(T-1)\right)^{a_1\eta_1} a_4(T-1)\eta_1 \log\left(e\,(T-1)\right),$$

*where $a_4(T-1) = 2(nN)^{-1}\sqrt{\tau(T-1)} + \mu\beta + 2(nN)^{-1}\sqrt{\tau(T-1)}\sqrt{d/b_2}$ and $\tau(T-1) = \mathcal{O}\left(\mu^2 \log T\right)$.*

Theorem 3 states the stability bound $\mathcal{O}\left(\left((nN)^{-1}\sqrt{\log T} + 1\right)(nN)^{-1}T^{\frac{1}{4}}\log T\right)$ if taking $\mu = \mathcal{O}\left((nN)^{-1}\right)$ and $\frac{a_1(1-d/b_2)}{4a_3} \leq \frac{1}{4}$. Compared with Theorem 2, the current result removes the dependence on the Lipschitz parameter $L$ while the dependence on $\mu$ is improved from the partial dependence $L/(nN) + \mu$ to the full dependence $(\sqrt{\log T}/(nN) + 1)\mu$. The reason is that, motivated by [36], we decompose the gradient approximation into two parts, i.e., the difference between the unknown gradient and its expected estimator, the divergence between the expected and its empirical version. With the help of second-order Taylor expansion, the first part is bounded by Lemmas 4 and 5 in *Appendix B.1*. Meanwhile, the second part is bounded by the PL condition. The detailed proof is provided by Equation (8) in *Appendix B.4*.

Combining Theorem 1 (b) with Theorem 3, we derive the following generalization bound for the heavy-tailed FedZO.

**Corollary 2.** *Under the same conditions of Theorem 3, for FedZO (3), there holds*

$$\left|\mathbb{E}[F(w^T) - F_S(w^T)]\right| \leq \mathcal{O}\left(\left((nN)^{-1}\sqrt{\log T} + 1\right)(4\theta)^\theta \mu T^{a_1\eta_1}\log T + \mathbb{E}[F_S(w^T)]\right).$$

When $\mu = \mathcal{O}\left((nN)^{-1}\right), \frac{a_1(1-d/b_2)}{4a_3} \leq \frac{1}{4}$ and $\mathbb{E}\left[F_S(w^T)\right] = \mathcal{O}\left((nN)^{-1}\right)$, Corollary 2 shows the generalization bound $\mathcal{O}\left(\left((nN)^{-1}\sqrt{\log T} + 1\right)(4\theta)^\theta(nN)^{-1}T^{\frac{1}{4}}\log T\right)$. With bounded variance of gradient and bounded loss function, [38] stated the generalization bound

$\mathcal{O}\left((n\delta)^{-\frac{1}{2}}L^{\frac{1}{2}}\left(\left(\mathbb{E}\left[v^2\right]\right)^{\frac{1}{4}}+\sqrt{\log T}\right)\right)$ for SGD with high probability. Here, we further developed the analysis technique associated with the bounded variance of gradient to the federated learning setting by the fine-grained error analysis (see the proof of Theorem 1 (b)).

**Theorem 4.** *Let $\{w^t\}$ be produced by FedZO (3) on S, where $\eta_t = \eta_1 t^{-1}, \eta_1 \leq \frac{1-d/b_2}{4a_3}$ with $a_3 = (1+d/b_2)\beta$ and $0 \geq \frac{1-d/b_2}{2}\alpha\eta_1 - 1$. Under Assumptions 1 (b), 2 and 3, there hold*

$$\mathbb{E}[F_S(w^T) - F_S(w(S))] \leq \mathcal{O}\left(T^{-2} + \mu^2\right)$$

*and*

$$\begin{aligned}
&|\mathbb{E}[F(w^T) - F(w^*)]| \\
&\leq \mathcal{O}\left(T^{-2} + \mu^2 + \left((nN)^{-1}\sqrt{\log T} + 1\right)(4\theta)^\theta \mu T^{a_1\eta_1} \log T + \mathbb{E}[F_S(w^T)]\right),
\end{aligned}$$

*where $w(S), w^*$ are defined in (4).*

The PL condition can guarantee the identification of global minimizers. Therefore, we regard $\mathbb{E}\left[F_S(w^T) - F_S(w^*)\right]$, instead of $\left\|\nabla F_S(w^T)\right\|$, as the measure of optimization error in this paper. [40] provided the optimal optimization bound $\mathcal{O}\left(1/(T\alpha^2)\right)$ with uniform stability tool for gradient-dominated pairwise SGD. [16] characterized the convergence rate $\mathcal{O}\left((MT)^{-\frac{1}{2}} + d\mu^2\right)$ of the FedZO algorithm with partial device participation. As shown in Table 2, Theorem 4 guarantees the optimal decay rate $\mathcal{O}\left(T^{-1}\right)$ on the optimization error as $\mu = \mathcal{O}(T^{-\frac{1}{2}})$ without the dependence of the dimension of hypothesis function space $d$ and the random direction number $b_2$. Note that, our optimization bounds rely on the quality of the initial model like many previous work (e.g. [16]).

## 4 Learning Guarantees of Asynchronous FedZO

Following the line of Section 3.2, we further study theoretical foundations for the asynchronous case of the FedZO algorithm. Considering the asynchrony among the local model parameters of different clients in the same iteration, we modify Equation (3) as follows

$$w^{t+1} = w^t - \frac{\eta_t}{b_1 N}\sum_{i=1}^{N}\sum_{m=1}^{b_1}\tilde{\nabla}f_i\left(w_i^{t_i}; z_{i,m}^{t_i}, \left\{v_{i,l}^{t_i}\right\}_{l=1}^{b_2}, \mu\right), \tag{7}$$

where $t - t_i \in [t_0]$ denotes the delay of the $i$-th client in the $t$-th iteration, and $t_0 = \max_{t\in[T]}\left\{\max_{i\in[N]} t_i - \min_{i\in[N]} t_i\right\}$ denotes the maximum delay for all clients in the whole update process of the global model. Note that, if $t_i = t$ for some $i$, the parameter $w_i^{t_i}$ will be updated to $w^{t+1}$ at the end of the $t$-th iteration. We state the differences between Equations (3) and (7) in Remark 2.

**Theorem 5.** *Given S and $S^{(j_i)}$ in Definition 1, let $\{w^t\}$ and $\{\bar{w}^t\}$ be produced by asynchronous FedZO (7) on S and $S^{(j_i)}$ respectively, where $\eta_t = \eta_1 t^{-1}, \eta_1 \leq \frac{1-d/b_2}{4a_3}$ with $a_3 = (1+d/b_2)\beta$ and $0 \geq \frac{1-d/b_2}{2}\alpha\eta_1 - 1$. After T iterations, we get the global parameters $A(S) = w^T$ and $A(S^{(j_i)}) = \bar{w}^T$ respectively. Under Assumptions 1 (b), 2 and 3, there holds*

$$\frac{1}{nN}\sum_{i=1}^{N}\sum_{j=1}^{n}\mathbb{E}\left[\|w^T - \bar{w}^T\|\right] \leq (e(T-1))^{2\beta\eta_1}\left(a_5(T-1)\eta_1\log(e(T-1)) + 4a_6(T-1)\eta_1^2\right),$$

*where $a_5(T-1) = \mu\beta + 4(nN)^{-1}\sqrt{\hat{\tau}(T-1)}\sqrt{d/b_2}$, $a_6(T-1) = \mu\beta^2\left((b_1N)^{-1}\left(1+\frac{b_1-1}{n}\right) + 2t_0\right) + \beta\left(2 + 2\sqrt{d/b_2}\right)\left((b_1N)^{-1}\left(1+\frac{b_1-1}{n}\right) + 2t_0\right)\sqrt{\hat{\tau}(T-1)}$ and $\hat{\tau}(T-1) = \mathcal{O}\left(\mu^2\log T\right)$.*

Asynchrony can cause discrepancies among the local models of various clients $w^{t_i}, i = 1, ..., N$ and the global model $w^t$ in the $t$-th iteration. Therefore, the primary challenge of asynchronous learning theory is to establish the relationship between $\left\|w_i^{t_i} - \bar{w}_i^{t_i}\right\|$ and $\left\|w^t - \bar{w}^t\right\|$. To overcome this bottleneck, we design a new decomposition $\left\|w_i^{t_i} - \bar{w}_i^{t_i}\right\| \leq \left\|w_i^{t_i} - w^t - (\bar{w}_i^{t_i} - \bar{w}^t)\right\| + \left\|w^t - \bar{w}^t\right\|$

and then use second-order Taylor expansion to give an upper bound of the first term on the right-hand side of the inequality. When $\mu = \mathcal{O}\left((nN)^{-1}\right)$ and $\frac{\beta(1-d/b_2)}{2a_3} \leq \frac{1}{2}$, Theorem 5 provide a stability bound $\mathcal{O}\left(\left(\left(1 + (nN)^{-1}\sqrt{\log T}\right)\log T + \sqrt{\log T}t_0\right)(nN)^{-1}T^{\frac{1}{2}}\right)$.

**Corollary 3.** *Under the same conditions of Theorem 5, for asynchronous FedZO* (7)*, there holds*

$$|\mathbb{E}[F(w^T) - F_S(w^T)]|$$
$$\leq \mathcal{O}\left(\left(\left(1 + (nN)^{-1}\sqrt{\log T}\right)\log T + \sqrt{\log T}t_0\right)(4\theta)^\theta \mu T^{2\beta\eta_1} + \mathbb{E}[F_S(w^T)]\right).$$

Combining Theorem 1 (b) with Theorem 5, when $\mu = \mathbb{E}[F_S(w^T)] = \mathcal{O}\left((nN)^{-1}\right), \frac{\beta(1-d/b_2)}{2a_3} \leq \frac{1}{2}$, Corollary 3 yields the generalization bound $\mathcal{O}\left(\left(\left(1 + (nN)^{-1}\sqrt{\log T}\right)\log T + \sqrt{\log T}t_0\right)(4\theta)^\theta (nN)^{-1}T^{\frac{1}{2}}\right)$ which appears to be the first generalization bound developed for asynchronous federated learning algorithms. It should be noted that, due to the more complex communication structure, using the condition $t_0 = 0$ (the synchronous case) can not recover the generalization bound in Corollary 2.

**Theorem 6.** *Let* $\{w^t\}$ *be produced by FedZO* (7) *on* $S$*, where* $\eta_t = \eta_1 t^{-1}, \eta_1 \leq \frac{1-d/b_2}{4a_3}$ *with* $a_3 = (1 + d/b_2)\beta$ *and* $0 \geq \frac{1-d/b_2}{2}\alpha\eta_1 - 1$*. Under Assumptions 1 (b), 2 and 3, there hold*

$$\mathbb{E}[F_S(w^T) - F_S(w(S))] \leq \mathcal{O}\left(T^{-2} + \mu^2\right)$$

*and*

$$|\mathbb{E}[F(w^T) - F(w^*)]|$$
$$\leq \mathcal{O}\left(T^{-2} + \mu^2 + \left(\left(1 + (nN)^{-1}\sqrt{\log T}\right)\log T + \sqrt{\log T}t_0\right)(4\theta)^\theta \mu T^{2\beta\eta_1} + \mathbb{E}[F_S(w^T)]\right),$$

*where* $w(S), w^*$ *are defined in* (4).

Theorem 6 also develops the first optimal optimization bound $\mathcal{O}\left(T^{-2} + \mu^2\right)$ for the asynchronous FedZO. Based on Equation (5), the excess risk bound $\mathcal{O}\left(T^{-2} + \left(\left(1 + (nN)^{-1}\sqrt{\log T}\right)\log T + t_0\sqrt{\log T}\right)(4\theta)^\theta(nN)^{-1}T^{\frac{1}{2}}\right)$ can be directly derived by integrating Corollary 3 and this optimization bound when $\mu = \mathbb{E}[F_S(w^T)] = \mathcal{O}\left((nN)^{-1}\right)$.

## 5  Conclusion

This paper fills the gap of theoretical guarantees for both synchronous and asynchronous FedZO algorithms. We develop the first generalization bound for general FedZO after bridging the quantitative relationships between generalization error and $\ell_1$ on-average model stability. Moreover, fine-grained learning theory analysis is established by means of the heavy-tailed condition and the second-order Taylor expansion, where the derived error bounds are satisfactory even compared with the previous results for traditional SGD [37, 32, 38] and recent ZoSS [36].

## Acknowledgments

This work was supported in part by the National Natural Science Foundation of China (Nos. 12071166 and 62376104), the Fundamental Research Funds for the Central Universities of China (No. 2662023LXPY005), and HZAU-AGIS Cooperation Fund (No. SZYJY2023010).

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
