# Appendix for "Fine-Grained Theoretical Analysis of Federated Zeroth-Order Optimization"

37th Conference on Neural Information Processing Systems (NeurIPS 2023).

## A. Notations

The main notations of this paper are summarized in Table 1.

Table 1: Descriptions of the main notations used in this work.

| Notations | Descriptions |
|---|---|
| $N, n$ | the total number of clients and the total sample number of each client |
| $S, S_i$ | the total dataset and the $i$-th local dataset, $i = 1, ..., N$ |
| $S_i^{(j)}$ | the $i$-th client's dataset where the $j$-th sample $z_{ij}$ is replaced by $z'_{ij}$, $S^{(j_i)} = \left\{ S_1, ..., S_{i-1}, S_i^{(j)}, S_{i+1}, ..., S_N \right\}$ |
| $\mathcal{M}_t, M$ | the collection of selected client indices in the $t$-th iteration and its size |
| $z_{ij}$ | the $j$-th sample in $S_i$ over distribution $\mathcal{D}_i$, $j = 1, ..., n$ |
| $\mathcal{D}_i$ | the distribution of $z_{ij}$ ($\mathcal{D}_i$ is independent of $\mathcal{D}_{i'}$ if $i \neq i'$) |
| $w^t, w_i^t$ | the training parameters for the global model and the $i$-th local model in the $t$-th iteration respectively |
| $\mathcal{W}, d$ | the hypothesis function space and its dimension |
| $F(w), F_i(w_i)$ | the expected risks for the global model and the local model of the $i$-th client |
| $F_S(w), F_{S_i}(w_i)$ | the empirical risks for the global model and the local model of the $i$-th client |
| $f_i(w_i; z_i), \nabla f_i, \tilde{\nabla} f_i$ | the loss function of the $i$-th local client over sample $z_i$, its first-order gradient and the corresponding estimation |
| $\eta_t, \mu$ | the step sizes in the $t$-th iteration and the positive step size in the definition of the derivative |
| $b_1, b_2$ | the sizes of i.i.d. random samples and random direction vectors |
| $v_{i,l}^t$ | the $l$-th random direction vector for the $i$-th client in the $t$-th iteration |
| $w^*, w(S)$ | the expected optimal model and the empirical optimal model |
| $T$ | the total number of iterations |
| $A, A(S) = w^T$ | the federated learning algorithm and the parameter trained with $A$ on $S$ |
| $\epsilon$ | the parameter of $\ell_1$ on-average model stability |
| $\| \cdot \|$ | the Euclidean norm |
| $\theta, K$ | the parameters related to sub-Weibull distribution |
| $L, \beta, \alpha$ | the parameters of Lipschitz, smoothness and PL condition respectively |
| $t - t_i, t_0$ | the delay of the $i$-th client in the $t$-th iteration and the maximum delay |
| $\Gamma(x)$ | $\Gamma(x) = \int_0^\infty t^{x-1} e^{-t} dt$ |

## B. Proofs of Main Results

We first introduce the lemmas which will be used in our proofs.

### B.1. Lemmas

**Lemma 1.** *If the function $f$ is $\beta$-smooth, then we have for any $z, \bar{z}$,*

$$f(w; z, \bar{z}) - f(\bar{w}; z, \bar{z}) \leq \langle w - \bar{w}, \nabla f(\bar{w}; z, \bar{z}) \rangle + \frac{1}{2} \beta \|w - \bar{w}\|^2 \tag{1}$$

*and*

$$\frac{1}{2\beta} \|\nabla f(w; z, \bar{z})\|^2 \leq f(w; z, \bar{z}) - \inf_{\bar{w}} f(\bar{w}; z, \tilde{z}) \leq f(w; z, \bar{z}). \tag{2}$$

**Lemma 2.** *[1]. Assume $X$ is $K$-sub-Weibull$(\theta)$, i.e. $X \sim subW(\theta, K)$, then $\|X\|_p \leq (2\theta)^\theta K p^\theta$, where $p \geq 1/\theta$. In particular, $\|X\|_2 \leq (4\theta)^\theta K, \theta \geq 1/2$.*

**Lemma 3.** *[2]. Let $e$ be the base of the natural logarithm. The following inequalities hold:*

*(a) if $\alpha = 1$, then $\sum_{k=1}^{t} k^{-\alpha} \leq \log(et)$; (b) if $\alpha > 1$, then $\sum_{k=1}^{t} k^{-\alpha} \leq \frac{\alpha}{\alpha - 1}$.*

**Lemma 4.** *[3]. Assme a random vector $X \in \mathbb{R}^d$ is $d$-dimensional uniform distribution. For any $k \in \mathbb{N}$, there holds $\mathbb{E}\left[\|X\|^k\right] = d/(d + k)$.*

**Lemma 5.** *[3]. Let $v_l \in \mathbb{R}^d, l \in \{1, 2, ..., b_2\}$ be i.i.d. random vectors satisfying $d$-dimensional uniform distribution. For every random vector $u \in \mathbb{R}^d$ independent of all $v_l$, the following inequality holds*

$$\mathbb{E}\left[\left\|\frac{1}{b_2}\sum_{l=1}^{b_2}\langle u, v_l\rangle v_l - u\right\| \, \Big| \, u\right] \leq \sqrt{\frac{d}{b_2}}\|u\|.$$

## B.2.  Proof of Theorem 1

**Proof of Theorem 1**: (a) According to the symmetry, triangular inequality, $L$-Lipschitz continuity and $\ell_1$ on-average model stability, we deduce that

$$|\mathbb{E}[F(A(S)) - F_S(A(S))]|$$

$$= \left|\frac{1}{N}\sum_{i=1}^{N}\mathbb{E}[F_i(A(S)) - F_{S_i}(A(S))]\right|$$

$$\leq \frac{1}{N}\sum_{i=1}^{n}|\mathbb{E}[F_i(A(S)) - F_{S_i}(A(S))]|$$

$$= \frac{1}{N}\sum_{i=1}^{N}\left|\mathbb{E}\left[F_i(A(S)) - \frac{1}{n}\sum_{j=1}^{n}f_i(A(S); z_{ij})\right]\right|$$

$$= \frac{1}{N}\sum_{i=1}^{N}\left|\frac{1}{n}\sum_{j=1}^{n}\mathbb{E}\left[f_i(A(S^{(j_i)}); z_{ij}) - f_i(A(S); z_{ij})\right]\right|$$

$$\leq \frac{1}{nN}\sum_{i=1}^{N}\sum_{j=1}^{n}\mathbb{E}\left[\left|f_i(A(S^{(j_i)}); z_{ij}) - f_i(A(S); z_{ij})\right|\right]$$

$$\leq \frac{L}{nN}\sum_{i=1}^{N}\sum_{j=1}^{n}\mathbb{E}\left[\left\|A(S^{(j_i)}) - A(S)\right\|\right]$$

$$\leq L\epsilon.$$

This proves Part (a).

(b) Let $g(w) = f(w) - F_S(w)$. From Lemma 2 and Assumption 1(a), it is obvious that, for any $w, w' \in \mathcal{W}, w \neq w'$,

$$\|\nabla g(w)\| \leq (4\theta)^\theta K,$$

which means

$$|f(w) - F_S(w) - (f(w') - F_{S'}(w'))| \leq (4\theta)^\theta K\|w - w'\|.$$

Taking expectation with respect to (w.r.t.) all randomness, we get that

$$\mathbb{E}[|f(w) - F_S(w) - (f(w') - F_{S'}(w'))|] \leq (4\theta)^\theta K\mathbb{E}[\|w - w'\|]. \tag{3}$$

Then,

$$|\mathbb{E}[F(A(S)) - F_S(A(S))]|$$

$$\leq \frac{1}{N} \sum_{i=1}^{N} |\mathbb{E}[F_i(A(S)) - F_{S_i}(A(S))]|$$

$$\leq \frac{1}{nN} \sum_{i=1}^{N} \sum_{j=1}^{n} \mathbb{E}\left[\left|f_i(A(S^{(j_i)}); z_{ij}) - f_i(A(S); z_{ij})\right|\right]$$

$$\leq \frac{1}{nN} \sum_{i=1}^{N} \sum_{j=1}^{n} \left( \mathbb{E}\left[\left|f_i(A(S^{(j_i)}); z_{ij}) - F_{S^{(j_i)}}(A(S^{(j_i)})) - f_i(A(S); z_{ij}) + F_S(A(S))\right|\right] \right.$$

$$\left. + \mathbb{E}\left[\left\|F_{S^{(j_i)}}(A(S^{(j_i)})) - F_S(A(S))\right\|\right] \right)$$

$$\leq \frac{(4\theta)^\theta K}{nN} \sum_{i=1}^{N} \sum_{j=1}^{n} \mathbb{E}[\|A(S^{(j_i)}) - A(S)\|] + 2\mathbb{E}[F_S(A(S))]$$

$$\leq (4\theta)^\theta K \epsilon + 2\mathbb{E}[F_S(A(S))],$$

where the first three inequalities are caused by the triangular inequality and the fourth inequality is due to Equation (3). The stated result in Part (b) is proved. □

### B.3. Proof of Theorem 2

**Proof of Theorem 2**: Let $S^{(j_i)} = S^{(n_N)} = \{S_i\}_{i=1}^{N-1} \cup S_N^{(n)}$. Define $\alpha_{ij}^t = \left|\{m : z_{i,m}^t = z_{ij}^t\}\right|$, $\forall t \in \mathbb{N}, i \in \mathcal{M}_t, j \in [n], m \in [b_1]$. That is $\alpha_{ij}^t$ is the number of samples that are equal to $z_{ij}$ in the $t$-th global iteration for the $i$-th edge device. It is obvious that $\mathbb{E}\left[\alpha_{ij}^t\right] = b_1/n, \mathbb{E}\left[(\alpha_{ij}^t)^2\right] = \left(\mathbb{E}\left[\alpha_{ij}^t\right]\right)^2 + Var\left(\alpha_{ij}^t\right) = \frac{b_1}{n}\left(1 + \frac{b_1-1}{n}\right)$. Then, the update can be reformulated as

$$w^{t+1} = w^t - \frac{\eta_t}{b_1 M} \sum_{i \in \mathcal{M}_t} \sum_{j \in [n]} \alpha_{ij}^t \tilde{\nabla} f_i\left(w_i^t; z_{ij}^t, \{v_{i,l}^t\}_{l=1}^{b_2}, \mu\right). \tag{4}$$

For the sake of simplicity, we denote $\tilde{\nabla} f_i\left(w_i^t; z_{ij}^t, \{v_{i,l}^t\}_{l=1}^{b_2}, \mu\right)$ as $\tilde{\nabla} f_i\left(w_i^t; z_{ij}^t\right)$. According to the new formulation, we can get

$$\|w^{t+1} - \bar{w}^{t+1}\|$$

$$= \left\|w^t - \bar{w}^t - \frac{\eta_t}{b_1 M} \sum_{i \in \mathcal{M}_t} \sum_{j \in [n]} \alpha_{ij}^t \left(\tilde{\nabla} f_i\left(w_i^t; z_{ij}^t\right) - \tilde{\nabla} f_i\left(\bar{w}_i^t; \bar{z}_{ij}^t\right)\right)\right\|$$

$$\leq \left\|w^t - \bar{w}^t - \frac{\eta_t}{b_1 M} \sum_{i \in \mathcal{M}_t} \sum_{j \in [n]} \alpha_{ij}^t \left(\nabla f_i\left(w_i^t; z_{ij}^t\right) - \nabla f_i\left(\bar{w}_i^t; \bar{z}_{ij}^t\right)\right)\right\| \tag{5}$$

$$+ \left\|\frac{\eta_t}{b_1 M} \sum_{i \in \mathcal{M}_t} \sum_{j \in [n]} \alpha_{ij}^t \left(\tilde{\nabla} f_i\left(w_i^t; z_{ij}^t\right) - \tilde{\nabla} f_i\left(\bar{w}_i^t; \bar{z}_{ij}^t\right) - \nabla f_i\left(w_i^t; z_{ij}^t\right) + \nabla f_i\left(\bar{w}_i^t; \bar{z}_{ij}^t\right)\right)\right\|. \tag{6}$$

Considering the possibility of choosing a client who has a disturbed sample, we carry out the following discussion. When $N \notin \mathcal{M}_t$, we use smoothness, the fact that $w_i^t = w^t$ to get

$$(5)$$

$$\leq \|w^t - \bar{w}^t\| + \frac{\eta_t}{b_1 M} \sum_{i \in \mathcal{M}_t} \sum_{j \in [n]} \alpha_{ij}^t \|\nabla f_i\left(w_i^t; z_{ij}^t\right) - \nabla f_i\left(\bar{w}_i^t; z_{ij}^t\right)\|$$

$$\leq \|w^t - \bar{w}^t\| + \frac{\beta \eta_t}{b_1 M} \sum_{i \in \mathcal{M}_t} \sum_{j \in [n]} \alpha_{ij}^t \|w_i^t - \bar{w}_i^t\|$$

$$= \|w^t - \bar{w}^t\| + \frac{\beta \eta_t}{b_1 M} \sum_{i \in \mathcal{M}_t} \sum_{j \in [n]} \alpha_{ij}^t \|w^t - \bar{w}^t\|$$

$$= \left(1 + \frac{\beta \eta_t}{b_1 M} \sum_{i \in \mathcal{M}_t} \sum_{j \in [n]} \alpha_{ij}^t \right) \|w^t - \bar{w}^t\|,$$

and

(6)

$$\leq \frac{\eta_t}{b_1 M} \sum_{i \in \mathcal{M}_t} \sum_{j \in [n]} \alpha_{ij}^t \left\| \tilde{\nabla} f_i\left(w_i^t; z_{ij}^t\right) - \tilde{\nabla} f_i\left(\bar{w}_i^t; z_{ij}^t\right) - \nabla f_i\left(w_i^t; z_{ij}^t\right) + \nabla f_i\left(\bar{w}_i^t; z_{ij}^t\right) \right\|$$

$$= \frac{\eta_t}{b_1 M} \sum_{i \in \mathcal{M}_t} \sum_{j \in [n]} \alpha_{ij}^t \left\| \frac{1}{b_2} \sum_{l=1}^{b_2} \left( \langle \nabla f_i(w_i^t; z_{i,j}^t) - \nabla f_i(\bar{w}_i^t; z_{ij}^t), v_{i,l}^t \rangle v_{i,l}^t \right. \right.$$

$$\left. + \left( \frac{\mu}{2} (v_{i,l}^t)^\top \nabla_{w_i}^2 f_i(w_i; z_{ij}^t)|_{w_i = w_{i,l}^{t*}} v_{i,l}^t \right) v_{i,l}^t - \left( \frac{\mu}{2} (v_{i,l}^t)^\top \nabla_{w_i}^2 f_i(w_i; z_{ij}^t)|_{w_i = w_{i,l}^{t\dagger}} v_{i,l}^t \right) v_{i,l}^t \right)$$

$$\left. - \nabla f_i\left(w_i^t; z_{ij}^t\right) + \nabla f_i\left(\bar{w}_i^t; z_{ij}^t\right) \right\|$$

$$= \frac{\eta_t}{b_1 M} \sum_{i \in \mathcal{M}_t} \sum_{j \in [n]} \alpha_{ij}^t \left( \left\| \frac{1}{b_2} \sum_{l=1}^{b_2} \left( \left( \frac{\mu}{2} (v_{i,l}^t)^\top \nabla_{w_i}^2 f_i(w_i; z_{ij}^t)|_{w_i = w_{i,l}^{t*}} v_{i,l}^t \right) v_{i,l}^t \right. \right. \right.$$

$$\left. \left. - \left( \frac{\mu}{2} (v_{i,l}^t)^\top \nabla_{w_i}^2 f_i(w_i; z_{ij}^t)|_{w_i = w_{i,l}^{t\dagger}} v_{i,l}^t \right) v_{i,l}^t \right) \right\|$$

$$\left. + \left\| \frac{1}{b_2} \sum_{l=1}^{b_2} \langle \nabla f_i(w_i^t; z_{i,j}^t) - \nabla f_i(\bar{w}_i^t; z_{ij}^t), v_{i,l}^t \rangle v_{i,l}^t - \nabla f_i\left(w_i^t; z_{ij}^t\right) + \nabla f_i\left(\bar{w}_i^t; z_{ij}^t\right) \right\| \right)$$

$$\leq \frac{\eta_t}{b_1 M} \sum_{i \in \mathcal{M}_t} \sum_{j \in [n]} \alpha_{ij}^t \left( \frac{2}{b_2} \sum_{l=1}^{b_2} \frac{\mu \beta}{2} \|v_{i,l}^t\|^3 \right.$$

$$\left. + \left\| \frac{1}{b_2} \sum_{l=1}^{b_2} \langle \nabla f_i(w_i^t; z_{ij}^t) - \nabla f_i(\bar{w}_i^t; z_{ij}^t), v_{i,l}^t \rangle v_{i,l}^t - \nabla f_i\left(w_i^t; z_{ij}^t\right) + \nabla f_i\left(\bar{w}_i^t; z_{ij}^t\right) \right\| \right)$$

$$\leq \frac{\eta_t}{b_1 M} \sum_{i \in \mathcal{M}_t} \sum_{j \in [n]} \alpha_{ij}^t \left( \frac{\mu \beta}{b_2} \sum_{l=1}^{b_2} \|v_{i,l}^t\|^3 \right.$$

$$\left. + \left\| \frac{1}{b_2} \sum_{l=1}^{b_2} \langle \nabla f_i(w_i^t; z_{ij}^t) - \nabla f_i(\bar{w}_i^t; z_{ij}^t), v_{i,l}^t \rangle v_{i,l}^t - \nabla f_i\left(w_i^t; z_{ij}^t\right) + \nabla f_i\left(\bar{w}_i^t; z_{ij}^t\right) \right\| \right).$$

When $N \in \mathcal{M}_t$, let $P_t = \{(i,j) | i \in \mathcal{M}_t / \{N\}, j \in [n] \text{ or } i = N, j \in [n-1]\}$, then

(5)

$$\leq \|w^t - \bar{w}^t\| + \frac{\eta_t}{b_1 M} \sum_{P_t} \alpha_{ij}^t \left\| \nabla f_i\left(w_i^t; z_{ij}^t\right) - \nabla f_i\left(\bar{w}_i^t; z_{ij}^t\right) \right\|$$

$$+ \frac{\eta_t}{b_1 M} \alpha_{Nn}^t \left\| \nabla f_N\left(w_N^t; z_{Nn}^t\right) - \nabla f_N\left(\bar{w}_N^t; \bar{z}_{Nn}^t\right) \right\|$$

$$\leq \|w^t - \bar{w}^t\| + \frac{\beta \eta_t}{b_1 M} \sum_{P_t} \alpha_{ij}^t \|w^t - \bar{w}^t\| + \frac{2 \eta_t L}{b_1 M} \alpha_{Nn}^t$$

$$= \left(1 + \frac{\beta \eta_t}{b_1 M} \sum_{P_t} \alpha_{ij}^t \right) \|w^t - \bar{w}^t\| + \frac{2 \eta_t L}{b_1 M} \alpha_{Nn}^t,$$

and

(6)

$$\leq \frac{\eta_t}{b_1 M} \sum_{P_t} \alpha_{ij}^t \left\| \tilde{\nabla} f_i\left(w_i^t; z_{ij}^t\right) - \tilde{\nabla} f_i\left(\bar{w}_i^t; z_{ij}^t\right) - \nabla f_i\left(w_i^t; z_{ij}^t\right) + \nabla f_i\left(\bar{w}_i^t; z_{ij}^t\right) \right\|$$

$$+ \frac{\eta_t}{b_1 M} \alpha_{Nn}^t \left\| \tilde{\nabla} f_N\left(w_N^t; z_{Nn}^t\right) - \tilde{\nabla} f_N\left(\bar{w}_N^t; \bar{z}_{Nn}^t\right) - \nabla f_N\left(w_N^t; z_{Nn}^t\right) + \nabla f_N\left(\bar{w}_N^t; \bar{z}_{Nn}^t\right) \right\|$$

$$\leq \frac{\eta_t}{b_1 M} \sum_{i \in \mathcal{M}_t} \sum_{j \in [n]} \alpha_{ij}^t \frac{\mu\beta}{b_2} \sum_{l=1}^{b_2} \|v_{i,l}^t\|^3$$

$$+ \frac{\eta_t}{b_1 M} \sum_{P_t} \alpha_{ij}^t \left\| \frac{1}{b_2} \sum_{l=1}^{b_2} \langle \nabla f_i(w_i^t; z_{ij}^t) - \nabla f_i(\bar{w}_i^t; z_{ij}^t), v_{i,l}^t \rangle v_{i,l}^t - \nabla f_i\left(w_i^t; z_{ij}^t\right) + \nabla f_i\left(\bar{w}_i^t; z_{ij}^t\right) \right\|$$

$$+ \frac{\eta_t}{b_1 M} \alpha_{Nn}^t \left\| \frac{1}{b_2} \sum_{l=1}^{b_2} \langle \nabla f_N(w_N^t; z_{Nn}^t) - \nabla f_N(\bar{w}_N^t; \bar{z}_{Nn}^t), v_{N,l}^t \rangle v_{N,l}^t - \nabla f_N\left(w_N^t; z_{Nn}^t\right) + \nabla f_N\left(\bar{w}_N^t; \bar{z}_{Nn}^t\right) \right\|.$$

Then, combining the above four inequalities, we obtain that

$$\|w^{t+1} - \bar{w}^{t+1}\|$$

$$\leq \frac{N-M}{N} \left( \left( 1 + \frac{\beta\eta_t}{b_1 M} \sum_{i \in \mathcal{M}_t} \sum_{j \in [n]} \alpha_{ij}^t \right) \|w^t - \bar{w}^t\| + \frac{\eta_t}{b_1 M} \sum_{i \in \mathcal{M}_t} \sum_{j \in [n]} \alpha_{ij}^t \left( \frac{\mu\beta}{b_2} \sum_{l=1}^{b_2} \|v_{i,l}^t\|^3 \right. \right.$$

$$\left. \left. + \left\| \frac{1}{b_2} \sum_{l=1}^{b_2} \langle \nabla f_i(w_i^t; z_{ij}^t) - \nabla f_i(\bar{w}_i^t; z_{ij}^t), v_{i,l}^t \rangle v_{i,l}^t - \nabla f_i\left(w_i^t; z_{ij}^t\right) + \nabla f_i\left(\bar{w}_i^t; z_{ij}^t\right) \right\| \right) \right)$$

$$+ \frac{M}{N} \left( \left( 1 + \frac{\beta\eta_t}{b_1 M} \sum_{P_t} \alpha_{ij}^t \right) \|w^t - \bar{w}^t\| + \frac{2\eta_t L}{b_1 M} \alpha_{Nn}^t + \frac{\eta_t}{b_1 M} \sum_{i \in \mathcal{M}_t} \sum_{j \in [n]} \alpha_{ij}^t \frac{\mu\beta}{b_2} \sum_{l=1}^{b_2} \|v_{i,l}^t\|^3 \right.$$

$$+ \frac{\eta_t}{b_1 M} \sum_{P_t} \alpha_{ij}^t \left\| \frac{1}{b_2} \sum_{l=1}^{b_2} \langle \nabla f_i(w_i^t; z_{ij}^t) - \nabla f_i(\bar{w}_i^t; z_{ij}^t), v_{i,l}^t \rangle v_{i,l}^t - \nabla f_i\left(w_i^t; z_{ij}^t\right) + \nabla f_i\left(\bar{w}_i^t; z_{ij}^t\right) \right\|$$

$$+ \frac{\eta_t}{b_1 M} \alpha_{Nn}^t \left\| \frac{1}{b_2} \sum_{l=1}^{b_2} \langle \nabla f_N(w_N^t; z_{Nn}^t) - \nabla f_N(\bar{w}_N^t; \bar{z}_{Nn}^t), v_{N,l}^t \rangle v_{N,l}^t - \nabla f_N\left(w_N^t; z_{Nn}^t\right) + \nabla f_N\left(\bar{w}_N^t; \bar{z}_{Nn}^t\right) \right\| \right).$$

Define $J_i^t = \{z_{i,1}^t, ..., z_{i,b_1}^t\}, t \in \mathbb{N}, i \in [N]$. Taking conditional expectation w.r.t. $J_i^t$, we derive

$$\mathbb{E}_{J_i^t}[\|w^{t+1} - \bar{w}^{t+1}\|]$$

$$\leq \frac{N-M}{N} \left( \left( 1 + \frac{\beta\eta_t}{b_1 M} \sum_{i \in \mathcal{M}_t} \sum_{j \in [n]} \mathbb{E}_{J_i^t}[\alpha_{ij}^t] \right) \|w^t - \bar{w}^t\| + \frac{\eta_t}{b_1 M} \sum_{i \in \mathcal{M}_t} \sum_{j \in [n]} \mathbb{E}_{J_i^t}[\alpha_{ij}^t] \left( \frac{\mu\beta}{b_2} \sum_{l=1}^{b_2} \|v_{i,l}^t\|^3 \right. \right.$$

$$\left. \left. + \left\| \frac{1}{b_2} \sum_{l=1}^{b_2} \langle \nabla f_i(w_i^t; z_{ij}^t) - \nabla f_i(\bar{w}_i^t; z_{ij}^t), v_{i,l}^t \rangle v_{i,l}^t - \nabla f_i\left(w_i^t; z_{ij}^t\right) + \nabla f_i\left(\bar{w}_i^t; z_{ij}^t\right) \right\| \right) \right)$$

$$+ \frac{M}{N} \left( \left( 1 + \frac{\beta\eta_t}{b_1 M} \sum_{P_t} \mathbb{E}_{J_i^t}[\alpha_{ij}^t] \right) \|w^t - \bar{w}^t\| \right.$$

$$+ \frac{2\eta_t L}{b_1 M} \mathbb{E}_{J_N^t}[\alpha_{Nn}^t] + \frac{\eta_t}{b_1 M} \sum_{i \in \mathcal{M}_t} \sum_{j \in [n]} \mathbb{E}_{J_i^t}[\alpha_{ij}^t] \frac{\mu\beta}{b_2} \sum_{l=1}^{b_2} \|v_{i,l}^t\|^3$$

$$+ \frac{\eta_t}{b_1 M} \sum_{P_t} \mathbb{E}_{J_i^t}[\alpha_{ij}^t] \left\| \frac{1}{b_2} \sum_{l=1}^{b_2} \langle \nabla f_i(w_i^t; z_{ij}^t) - \nabla f_i(\bar{w}_i^t; z_{ij}^t), v_{i,l}^t \rangle v_{i,l}^t - \nabla f_i\left(w_i^t; z_{ij}^t\right) + \nabla f_i\left(\bar{w}_i^t; z_{ij}^t\right) \right\|$$

$$+ \frac{\eta_t}{b_1 M} \mathbb{E}_{J_N^t}[\alpha_{Nn}^t] \left\| \frac{1}{b_2} \sum_{l=1}^{b_2} \langle \nabla f_N(w_N^t; z_{Nn}^t) - \nabla f_N(\bar{w}_N^t; \bar{z}_{Nn}^t), v_{N,l}^t \rangle v_{N,l}^t - \nabla f_N\left(w_N^t; z_{Nn}^t\right) + \nabla f_N\left(\bar{w}_N^t; \bar{z}_{Nn}^t\right) \right\| \right)$$

$$=\frac{N-M}{N}(1+\eta_t\beta)\|w^t-\bar{w}^t\| + \frac{M}{N}(1+\eta_t\beta)\|w^t-\bar{w}^t\| + \frac{2\eta_t L}{nN} + \frac{\mu\eta_t\beta}{b_2}\sum_{l=1}^{b_2}\|v_{i,l}^t\|^3$$

$$+\frac{N-M}{N}\eta_t\left\|\frac{1}{b_2}\sum_{l=1}^{b_2}\langle\nabla f_i(w_i^t;z_{ij}^t)-\nabla f_i(\bar{w}_i^t;z_{ij}^t),v_{i,l}^t\rangle v_{i,l}^t - \nabla f_i\left(w_i^t;z_{ij}^t\right)+\nabla f_i\left(\bar{w}_i^t;z_{ij}^t\right)\right\|$$

$$+\frac{M}{N}\eta_t\left\|\frac{1}{b_2}\sum_{l=1}^{b_2}\langle\nabla f_i(w_i^t;z_{ij}^t)-\nabla f_i(\bar{w}_i^t;z_{ij}^t),v_{i,l}^t\rangle v_{i,l}^t - \nabla f_i\left(w_i^t;z_{ij}^t\right)+\nabla f_i\left(\bar{w}_i^t;z_{ij}^t\right)\right\|$$

$$+\frac{\eta_t}{nN}\left\|\frac{1}{b_2}\sum_{l=1}^{b_2}\langle\nabla f_N(w_N^t;z_{Nn}^t)-\nabla f_N(\bar{w}_N^t;\bar{z}_{Nn}^t),v_{N,l}^t\rangle v_{N,l}^t - \nabla f_N\left(w_N^t;z_{Nn}^t\right)+\nabla f_N\left(\bar{w}_N^t;\bar{z}_{Nn}^t\right)\right\|$$

$$\leq(1+\eta_t\beta)\|w^t-\bar{w}^t\| + \frac{2\eta_t L}{nN} + \frac{\mu\eta_t\beta}{b_2}\sum_{l=1}^{b_2}\|v_{i,l}^t\|^3$$

$$+\eta_t\left\|\frac{1}{b_2}\sum_{l=1}^{b_2}\langle\nabla f_i(w_i^t;z_{ij}^t)-\nabla f_i(\bar{w}_i^t;z_{ij}^t),v_{i,l}^t\rangle v_{i,l}^t - \nabla f_i\left(w_i^t;z_{ij}^t\right)+\nabla f_i\left(\bar{w}_i^t;z_{ij}^t\right)\right\|$$

$$+\frac{\eta_t}{nN}\left\|\frac{1}{b_2}\sum_{l=1}^{b_2}\langle\nabla f_N(w_N^t;z_{Nn}^t)-\nabla f_N(\bar{w}_N^t;\bar{z}_{Nn}^t),v_{N,l}^t\rangle v_{N,l}^t - \nabla f_N\left(w_N^t;z_{Nn}^t\right)+\nabla f_N\left(\bar{w}_N^t;\bar{z}_{Nn}^t\right)\right\|.$$

Further taking expectation w.r.t. all randomness and utilizing Lemmas 4, 5, we obtain that

$$\mathbb{E}[\|w^{t+1}-\bar{w}^{t+1}\|]$$

$$\leq(1+\eta_t\beta)\mathbb{E}[\|w^t-\bar{w}^t\|] + \frac{2\eta_t L}{nN} + \mu\eta_t\beta\mathbb{E}[\|v_{i,l}^t\|^3]$$

$$+\eta_t\mathbb{E}\left[\left\|\frac{1}{b_2}\sum_{l=1}^{b_2}\langle\nabla f_i(w_i^t;z_{ij}^t)-\nabla f_i(\bar{w}_i^t;z_{ij}^t),v_{i,l}^t\rangle v_{i,l}^t - \nabla f_i\left(w_i^t;z_{ij}^t\right)+\nabla f_i\left(\bar{w}_i^t;z_{ij}^t\right)\right\|\right]$$

$$+\frac{\eta_t}{nN}\mathbb{E}\left[\left\|\frac{1}{b_2}\sum_{l=1}^{b_2}\langle\nabla f_N(w_N^t;z_{Nn}^t)-\nabla f_N(\bar{w}_N^t;\bar{z}_{Nn}^t),v_{N,l}^t\rangle v_{N,l}^t - \nabla f_N\left(w_N^t;z_{Nn}^t\right)+\nabla f_N\left(\bar{w}_N^t;\bar{z}_{Nn}^t\right)\right\|\right]$$

$$\leq(1+\eta_t\beta)\mathbb{E}[\|w^t-\bar{w}^t\|] + \frac{2\eta_t L}{nN} + \frac{d\mu\eta_t\beta}{d+3} + \eta_t\sqrt{\frac{d}{b_2}}\mathbb{E}\left[\|\nabla f_i\left(w_i^t;z_{ij}^t\right)-\nabla f_i\left(\bar{w}_i^t;z_{ij}^t\right)\|\right]$$

$$+\frac{\eta_t}{nN}\sqrt{\frac{d}{b_2}}\mathbb{E}\left[\|\nabla f_N\left(w_N^t;z_{Nj}^t\right)-\nabla f_N\left(\bar{w}_N^t;\bar{z}_{Nj}^t\right)\|\right]$$

$$\leq\left(1+\left(1+\sqrt{\frac{d}{b_2}}\right)\eta_t\beta\right)\mathbb{E}[\|w^t-\bar{w}^t\|] + \left(\frac{2L}{nN}+\mu\beta+\frac{2L}{nN}\sqrt{\frac{d}{b_2}}\right)\eta_t.$$

Let $a_1=\left(1+\sqrt{\frac{d}{b_2}}\right)\beta$ and $a_2=\frac{2L}{nN}+\mu\beta+\frac{2L}{nN}\sqrt{\frac{d}{b_2}}$. Taking summation from $t=1$ to $T-1$, we deduce that

$$\mathbb{E}[\|w^T-\bar{w}^T\|]$$

$$\leq\sum_{t=1}^{T-1}\left(\prod_{s=t+1}^{T-1}(1+a_1\eta_s)\right)a_2\eta_t$$

$$\leq\sum_{t=1}^{T-1}\exp\left(\sum_{s=t+1}^{T-1}a_1\eta_s\right)a_2\eta_t$$

$$\leq\sum_{t=1}^{T-1}\exp\left(a_1\eta_1\sum_{s=1}^{T-1}s^{-1}\right)a_2\eta_t$$

$$=\exp\left(a_1\eta_1\sum_{s=1}^{T-1}s^{-1}\right)a_2\eta_1\sum_{t=1}^{T-1}t^{-1}$$

$$\leq (e\,(T-1))^{a_1\eta_1}\,a_2\eta_1\log(e(T-1))$$
$$\leq \mathcal{O}\left(\left((nN)^{-1}L + \mu\right)T^{\frac{1}{2}}\log T\right),$$

where the second inequality is derived by $1 + x \leq e^x$ and the fourth inequality follows by Lemma 3 (a). $\qquad\square$

**Proof of Corollary 1**: We integrate Theorem 1 (a) and Theorem 2 to obtain that

$$\left|\mathbb{E}\left[F(w^T) - F_S(w^T)\right]\right|$$
$$\leq \frac{L}{nN}\sum_{i=1}^{N}\sum_{j=1}^{n}\mathbb{E}\left[\left\|w^T - \bar{w}^T\right\|\right] = L\mathbb{E}\left[\left\|w^T - \bar{w}^T\right\|\right]$$
$$\leq \mathcal{O}\left(L\left((nN)^{-1}L + \mu\right)T^{\frac{1}{2}}\log T\right).$$

The proof is complete. $\qquad\square$

### B.4. Proofs of Theorem 3 and Thoerem 4

**Proof of Theorem 3**: Let $S^{(j_i)} = S^{(n_N)} = \{S_i\}_{i=1}^{N-1} \cup S_N^{(n)}$. Similar to the proof of Lemma 1, when $N \notin \mathcal{M}_t$,

$$(5) = \left(1 + \frac{\beta\eta_t}{b_1 M}\sum_{i\in\mathcal{M}_t}\sum_{j\in[n]}\alpha_{ij}^t\right)\left\|w^t - \bar{w}^t\right\|,$$

and

$$(6) \leq \frac{\eta_t}{b_1 M}\sum_{i\in\mathcal{M}_t}\sum_{j\in[n]}\alpha_{ij}^t\left(\frac{\mu\beta}{b_2}\sum_{l=1}^{b_2}\|v_{i,l}^t\|^3\right.$$
$$\left. + \left\|\frac{1}{b_2}\sum_{l=1}^{b_2}\langle\nabla f_i(w_i^t; z_{ij}^t) - \nabla f_i(\bar{w}_i^t; z_{ij}^t), v_{i,l}^t\rangle v_{i,l}^t - \nabla f_i\left(w_i^t; z_{ij}^t\right) + \nabla f_i\left(\bar{w}_i^t; z_{ij}^t\right)\right\|\right).$$

When $N \in \mathcal{M}_t$, let $P_t = \{(i,j)|i \in \mathcal{M}_t/\{N\}, j \in [n] \text{ or } i = N, j \in [n-1]\}$, then

$$(5) = \left(1 + \frac{\beta\eta_t}{b_1 M}\sum_{P_t}\alpha_{ij}^t\right)\left\|w^t - \bar{w}^t\right\| + \frac{2\eta_t}{b_1 M}\alpha_{Nn}^t\|\nabla f_N(w_N^t; z_{Nn}^t)\|,$$

and

$$(6)$$
$$\leq \frac{\eta_t}{b_1 M}\sum_{i\in\mathcal{M}_t}\sum_{j\in[n]}\alpha_{ij}^t\frac{\mu\beta}{b_2}\sum_{l=1}^{b_2}\|v_{i,l}^t\|^3$$
$$+ \frac{\eta_t}{b_1 M}\sum_{P_t}\alpha_{ij}^t\left\|\frac{1}{b_2}\sum_{l=1}^{b_2}\langle\nabla f_i(w_i^t; z_{ij}^t) - \nabla f_i(\bar{w}_i^t; z_{ij}^t), v_{i,l}^t\rangle v_{i,l}^t - \nabla f_i\left(w_i^t; z_{ij}^t\right) + \nabla f_i\left(\bar{w}_i^t; z_{ij}^t\right)\right\|$$
$$+ \frac{\eta_t}{b_1 M}\alpha_{Nn}^t\left\|\frac{1}{b_2}\sum_{l=1}^{b_2}\langle\nabla f_N(w_N^t; z_{Nn}^t) - \nabla f_N(\bar{w}_N^t; \bar{z}_{Nn}^t), v_{N,l}^t\rangle v_{N,l}^t - \nabla f_N\left(w_N^t; z_{Nn}^t\right) + \nabla f_N\left(\bar{w}_N^t; \bar{z}_{Nn}^t\right)\right\|.$$

Then, combining the above four inequalities, we obtain that

$$\left\|w^{t+1} - \bar{w}^{t+1}\right\|$$
$$\leq \frac{N-M}{N}\left(\left(1 + \frac{\beta\eta_t}{b_1 M}\sum_{i\in\mathcal{M}_t}\sum_{j\in[n]}\alpha_{ij}^t\right)\left\|w^t - \bar{w}^t\right\| + \frac{\eta_t}{b_1 M}\sum_{i\in\mathcal{M}_t}\sum_{j\in[n]}\alpha_{ij}^t\left(\frac{\mu\beta}{b_2}\sum_{l=1}^{b_2}\|v_{i,l}^t\|^3\right.\right.$$
$$\left.\left. + \left\|\frac{1}{b_2}\sum_{l=1}^{b_2}\langle\nabla f_i(w_i^t; z_{ij}^t) - \nabla f_i(\bar{w}_i^t; z_{ij}^t), v_{i,l}^t\rangle v_{i,l}^t - \nabla f_i\left(w_i^t; z_{ij}^t\right) + \nabla f_i\left(\bar{w}_i^t; z_{ij}^t\right)\right\|\right)\right)$$

$$+ \frac{M}{N}\left(\left(1 + \frac{\beta\eta_t}{b_1 M}\sum_{P_t}\alpha_{ij}^t\right)\|w^t - \bar{w}^t\| + \frac{2\eta_t}{b_1 M}\alpha_{Nn}^t\|\nabla f_N(w_N^t; z_{Nn}^t)\| + \frac{\eta_t}{b_1 M}\sum_{i \in \mathcal{M}_t}\sum_{j \in [n]}\alpha_{ij}^t\frac{\mu\beta}{b_2}\sum_{l=1}^{b_2}\|v_{i,l}^t\|^3\right.$$

$$+ \frac{\eta_t}{b_1 M}\sum_{P_t}\alpha_{ij}^t\left\|\frac{1}{b_2}\sum_{l=1}^{b_2}\langle\nabla f_i(w_i^t; z_{ij}^t) - \nabla f_i(\bar{w}_i^t; z_{ij}^t), v_{i,l}^t\rangle v_{i,l}^t - \nabla f_i\left(w_i^t; z_{ij}^t\right) + \nabla f_i\left(\bar{w}_i^t; z_{ij}^t\right)\right\|$$

$$\left.+ \frac{\eta_t}{b_1 M}\alpha_{Nn}^t\left\|\frac{1}{b_2}\sum_{l=1}^{b_2}\langle\nabla f_N(w_N^t; z_{Nn}^t) - \nabla f_N(\bar{w}_N^t; \bar{z}_{Nn}^t), v_{N,l}^t\rangle v_{N,l}^t - \nabla f_N\left(w_N^t; z_{Nn}^t\right) + \nabla f_N\left(\bar{w}_N^t; \bar{z}_{Nn}^t\right)\right\|\right).$$

Taking conditional expectation with respect to (w.r.t.) $J_i^t$, we derive

$$\mathbb{E}_{J_i^t}[\|w^{t+1} - \bar{w}^{t+1}\|]$$

$$\leq \frac{N-M}{N}\left(\left(1 + \frac{\beta\eta_t}{b_1 M}\sum_{i \in \mathcal{M}_t}\sum_{j \in [n]}\mathbb{E}_{J_i^t}[\alpha_{ij}^t]\right)\|w^t - \bar{w}^t\| + \frac{\eta_t}{b_1 M}\sum_{i \in \mathcal{M}_t}\sum_{j \in [n]}\mathbb{E}_{J_i^t}[\alpha_{ij}^t]\left(\frac{\mu\beta}{b_2}\sum_{l=1}^{b_2}\|v_{i,l}^t\|^3\right.\right.$$

$$\left.\left.+ \left\|\frac{1}{b_2}\sum_{l=1}^{b_2}\langle\nabla f_i(w_i^t; z_{ij}^t) - \nabla f_i(\bar{w}_i^t; z_{ij}^t), v_{i,l}^t\rangle v_{i,l}^t - \nabla f_i\left(w_i^t; z_{ij}^t\right) + \nabla f_i\left(\bar{w}_i^t; z_{ij}^t\right)\right\|\right)\right)$$

$$+ \frac{M}{N}\left(\left(1 + \frac{\beta\eta_t}{b_1 M}\sum_{P_t}\mathbb{E}_{J_i^t}[\alpha_{ij}^t]\right)\|w^t - \bar{w}^t\| + \frac{2\eta_t}{b_1 M}\mathbb{E}_{J_N^t}[\alpha_{Nn}^t]\|\nabla f_N(w_N^t; z_{Nn}^t)\|\right.$$

$$+ \frac{\eta_t}{b_1 M}\sum_{i \in \mathcal{M}_t}\sum_{j \in [n]}\mathbb{E}_{J_i^t}[\alpha_{ij}^t]\frac{\mu\beta}{b_2}\sum_{l=1}^{b_2}\|v_{i,l}^t\|^3 + \frac{\eta_t}{b_1 M}\sum_{P_t}\mathbb{E}_{J_i^t}[\alpha_{ij}^t]$$

$$\left\|\frac{1}{b_2}\sum_{l=1}^{b_2}\langle\nabla f_i(w_i^t; z_{ij}^t) - \nabla f_i(\bar{w}_i^t; z_{ij}^t), v_{i,l}^t\rangle v_{i,l}^t - \nabla f_i\left(w_i^t; z_{ij}^t\right) + \nabla f_i\left(\bar{w}_i^t; z_{ij}^t\right)\right\| + \frac{\eta_t}{b_1 M}\mathbb{E}_{J_N^t}[\alpha_{Nn}^t]$$

$$\left.\left\|\frac{1}{b_2}\sum_{l=1}^{b_2}\langle\nabla f_N(w_N^t; z_{Nn}^t) - \nabla f_N(\bar{w}_N^t; \bar{z}_{Nn}^t), v_{N,l}^t\rangle v_{N,l}^t - \nabla f_N\left(w_N^t; z_{Nn}^t\right) + \nabla f_N\left(\bar{w}_N^t; \bar{z}_{Nn}^t\right)\right\|\right)$$

$$\leq (1 + \eta_t\beta)\|w^t - \bar{w}^t\| + \frac{2\eta_t}{nN}\|\nabla f_N(w_N^t; z_{Nn}^t)\| + \frac{\mu\eta_t\beta}{b_2}\sum_{l=1}^{b_2}\|v_{i,l}^t\|^3$$

$$+ \eta_t\left\|\frac{1}{b_2}\sum_{l=1}^{b_2}\langle\nabla f_i(w_i^t; z_{ij}^t) - \nabla f_i(\bar{w}_i^t; z_{ij}^t), v_{i,l}^t\rangle v_{i,l}^t - \nabla f_i\left(w_i^t; z_{ij}^t\right) + \nabla f_i\left(\bar{w}_i^t; z_{ij}^t\right)\right\|$$

$$+ \frac{\eta_t}{nN}\left\|\frac{1}{b_2}\sum_{l=1}^{b_2}\langle\nabla f_N(w_N^t; z_{Nn}^t) - \nabla f_N(\bar{w}_N^t; \bar{z}_{Nn}^t), v_{N,l}^t\rangle v_{N,l}^t - \nabla f_N\left(w_N^t; z_{Nn}^t\right) + \nabla f_N\left(\bar{w}_N^t; \bar{z}_{Nn}^t\right)\right\|.$$

Further taking expectation w.r.t. all randomness and utilizing Lemmas 4, 5, we obtain that

$$\mathbb{E}[\|w^{t+1} - \bar{w}^{t+1}\|]$$

$$\leq (1 + \eta_t\beta)\mathbb{E}[\|w^t - \bar{w}^t\|] + \frac{2\eta_t}{nN}\mathbb{E}[\|\nabla f_N(w_N^t; z_{Nn}^t)\|] + \mu\eta_t\beta\mathbb{E}[\|v_{i,l}^t\|^3]$$

$$+ \eta_t\mathbb{E}\left[\left\|\frac{1}{b_2}\sum_{l=1}^{b_2}\langle\nabla f_i(w_i^t; z_{ij}^t) - \nabla f_i(\bar{w}_i^t; z_{ij}^t), v_{i,l}^t\rangle v_{i,l}^t - \nabla f_i\left(w_i^t; z_{ij}^t\right) + \nabla f_i\left(\bar{w}_i^t; z_{ij}^t\right)\right\|\right]$$

$$+ \frac{\eta_t}{nN}\mathbb{E}\left[\left\|\frac{1}{b_2}\sum_{l=1}^{b_2}\langle\nabla f_N(w_N^t; z_{Nn}^t) - \nabla f_N(\bar{w}_N^t; \bar{z}_{Nn}^t), v_{N,l}^t\rangle v_{N,l}^t - \nabla f_N\left(w_N^t; z_{Nn}^t\right) + \nabla f_N\left(\bar{w}_N^t; \bar{z}_{Nn}^t\right)\right\|\right]$$

$$\leq (1 + \eta_t\beta)\mathbb{E}[\|w^t - \bar{w}^t\|] + \frac{2\eta_t}{nN}\mathbb{E}[\|\nabla f_N(w_N^t; z_{Nn}^t)\|] + \frac{d\mu\eta_t\beta}{d+3}$$

$$+ \eta_t\sqrt{\frac{d}{b_2}}\mathbb{E}\left[\|\nabla f_i\left(w_i^t; z_{ij}^t\right) - \nabla f_i\left(\bar{w}_i^t; z_{ij}^t\right)\|\right] + \frac{\eta_t}{nN}\sqrt{\frac{d}{b_2}}\mathbb{E}\left[\|\nabla f_N\left(w_N^t; z_{Nn}^t\right) - \nabla f_N\left(\bar{w}_N^t; \bar{z}_{Nn}^t\right)\|\right]$$

$$\leq \left(1 + \left(1 + \sqrt{\frac{d}{b_2}}\right)\eta_t\beta\right)\mathbb{E}[\|w^t - \bar{w}^t\|]$$

$$+ \left(\frac{2}{nN}\mathbb{E}[\|\nabla f_N(w_N^t; z_{Nn}^t)\|] + \mu\beta + \frac{2}{nN}\sqrt{\frac{d}{b_2}}\mathbb{E}[\|\nabla f_N(w_N^t; z_{Nj}^t)\|]\right)\eta_t. \tag{7}$$

To measure the stability, we need to obtain the upper bound of $\mathbb{E}[\|\nabla f_N(w_N^t; z_{Nj}^t)\|], j = 1, ..., n$. Based on Equation (1), the update of FedZO, triangular inequality, Lemmas 4, 5 and Assumption 3, we provide that

$$\mathbb{E}[F_S(w^{t+1}) - F_S(w^t)]$$

$$\leq \mathbb{E}\left[\langle w^{t+1} - w^t, \nabla F_S(w^t)\rangle + \frac{1}{2}\beta\|w^{t+1} - w^t\|^2\right]$$

$$= \mathbb{E}\left[\left\langle -\frac{\eta_t}{b_1 M}\sum_{i\in\mathcal{M}_t}\sum_{m\in[b_1]}\tilde{\nabla}f_i\left(w_i^t; z_{i,m}^t\right), \nabla F_S(w^t)\right\rangle + \frac{1}{2}\beta\left\|\frac{\eta_t}{b_1 M}\sum_{i\in\mathcal{M}_t}\sum_{m\in[b_1]}\tilde{\nabla}f_i\left(w_i^t; z_{i,m}^t\right)\right\|^2\right]$$

$$= -\frac{\eta_t}{b_1 M}\sum_{i\in\mathcal{M}_t}\sum_{m\in[b_1]}\mathbb{E}\left[\left\langle\tilde{\nabla}f_i\left(w_i^t; z_{i,m}^t\right) - \nabla f_i\left(w_i^t; z_{i,m}^t\right) + \nabla f_i\left(w_i^t; z_{i,m}^t\right), \nabla F_S(w^t)\right\rangle\right]$$

$$+ \frac{\beta}{2}\mathbb{E}\left[\left\|\frac{\eta_t}{b_1 M}\sum_{i\in\mathcal{M}_t}\sum_{m\in[b_1]}\left(\tilde{\nabla}f_i\left(w_i^t; z_{i,m}^t\right) - \nabla f_i\left(w_i^t; z_{i,m}^t\right) + \nabla f_i\left(w_i^t; z_{i,m}^t\right)\right)\right\|^2\right]$$

$$\leq -\frac{\eta_t}{b_1 M}\sum_{i\in\mathcal{M}_t}\sum_{m\in[b_1]}\mathbb{E}\left[\left\langle\tilde{\nabla}f_i\left(w_i^t; z_{i,m}^t\right) - \nabla f_i\left(w_i^t; z_{i,m}^t\right), \nabla F_S(w^t)\right\rangle\right] - \eta_t\mathbb{E}\left[\|\nabla F_S(w^t)\|^2\right]$$

$$+ \beta\eta_t^2\mathbb{E}\left[\left\|\tilde{\nabla}f_i\left(w_i^t; z_{i,m}^t\right) - \nabla f_i\left(w_i^t; z_{i,m}^t\right)\right\|^2\right] + \beta\eta_t^2\mathbb{E}\left[\|\nabla F_S(w^t)\|^2\right]$$

$$\leq \frac{\eta_t}{2b_1 M}\sum_{i\in\mathcal{M}_t}\sum_{m\in[b_1]}\mathbb{E}\left[\left\|\tilde{\nabla}f_i\left(w_i^t; z_{i,m}^t\right) - \nabla f_i\left(w_i^t; z_{i,m}^t\right)\right\|^2\right] + \frac{\eta_t}{2}\mathbb{E}\left[\|\nabla F_S(w^t)\|^2\right]$$

$$- \eta_t\mathbb{E}\left[\|\nabla F_S(w^t)\|^2\right] + \beta\eta_t^2\mathbb{E}\left[\left\|\tilde{\nabla}f_i\left(w_i^t; z_{i,m}^t\right) - \nabla f_i\left(w_i^t; z_{i,m}^t\right)\right\|^2\right] + \beta\eta_t^2\mathbb{E}\left[\|\nabla F_S(w^t)\|^2\right]$$

$$\leq \left(\beta\eta_t^2 - \frac{\eta_t}{2}\right)\mathbb{E}\left[\|\nabla F_S(w^t)\|^2\right] + \left(\frac{\eta_t}{2b_1 M} + \frac{\beta\eta_t^2}{b_1 M}\right)\sum_{i\in\mathcal{M}_t}\sum_{m\in[b_1]}\left(\frac{\mu^2\beta^2}{4}\mathbb{E}\left[\|v_{i,l}^t\|^6\right]\right.$$

$$\left. + \frac{d}{b_2}\mathbb{E}\left[\|\nabla f_i\left(w_i^t; z_{i,m}^t\right)\|^2\right]\right)$$

$$= \left(\left(1 + \frac{d}{b_2}\right)\beta\eta_t^2 - \left(\frac{1}{2} - \frac{d}{2b_2}\right)\eta_t\right)\mathbb{E}\left[\|\nabla F_S(w^t)\|^2\right] + \frac{d\mu^2\beta^3\eta_t^2}{4(d+6)} + \frac{d\mu^2\beta^2\eta_t}{8(d+6)}$$

$$\leq -\left(\frac{1}{4} - \frac{d}{4b_2}\right)\eta_t\mathbb{E}\left[\|\nabla F_S(w^t)\|^2\right] + \frac{\mu^2\beta^3\eta_t^2}{4} + \frac{\mu^2\beta^2\eta_t}{8}$$

$$\leq -\left(\frac{1}{2} - \frac{d}{2b_2}\right)\alpha\eta_t\mathbb{E}\left[F_S(w^t) - F_S(w(S))\right] + \frac{\mu^2\beta^3\eta_t^2}{4} + \frac{\mu^2\beta^2\eta_t}{8}. \tag{8}$$

Then,

$$\mathbb{E}[F_S(w^{t+1}) - F_S(w(S))]$$

$$\leq \left(1 - \left(\frac{1}{2} - \frac{d}{2b_2}\right)\alpha\eta_t\right)\mathbb{E}\left[F_S(w^t) - F_S(w(S))\right] + \frac{\mu^2\beta^3\eta_t^2}{4} + \frac{\mu^2\beta^2\eta_t}{8}$$

$$\leq \mathbb{E}\left[F_S(w^t) - F_S(w(S))\right] + \frac{\mu^2\beta^3\eta_t^2}{4} + \frac{\mu^2\beta^2\eta_t}{8}$$

$$\leq \mathbb{E}\left[F_S(w^1) - F_S(w(S))\right] + \sum_{i=1}^{t}\left(\frac{\mu^2\beta^3\eta_t^2}{4} + \frac{\mu^2\beta^2\eta_t}{8}\right)$$

$$\leq \mathbb{E}\left[F_S(w^1) - F_S(w(S))\right] + \frac{\mu^2\beta^3\eta_1^2}{4}\sum_{i=1}^{t}i^{-2} + \frac{\mu^2\beta^2\eta_1}{8}\sum_{i=1}^{t}i^{-1}$$

$$\leq \mathbb{E}\left[F_S(w^1) - F_S(w(S))\right] + \frac{\mu^2\beta^3\eta_1^2}{2} + \frac{\mu^2\beta^2\eta_1}{8}\log(et),$$

where the last inequality is due to Lemma 3 (a), (b). It follows from Lemma 1 (2) that

$$\mathbb{E}[\|\nabla f_N(w_N^t; z_{Nj}^t)\|^2]$$

$$\leq 2\beta\mathbb{E}[F_{S_N}(w_N^t)] = \frac{2\beta}{N}\sum_{i=1}^{N}\mathbb{E}[F_{S_i}(w_i^t)] = 2\beta\mathbb{E}[F_S(w^t)]$$

$$\leq 2\beta\left(\mathbb{E}[F_S(w^1)] + \frac{\mu^2\beta^3\eta_1^2}{2} + \frac{\mu^2\beta^2\eta_1}{8}\log(e(t-1))\right)$$

For convenience, we denote $2\beta\left(\mathbb{E}[F_S(w^1)] + \frac{\mu^2\beta^3\eta_1^2}{2} + \frac{\mu^2\beta^2\eta_1}{8}\log(e(t-1))\right)$ as $\tau(t)$. Then, from Equation (7), we know that

$$\mathbb{E}[\|w^{t+1} - \bar{w}^{t+1}\|]$$

$$\leq \left(1 + \left(1 + \sqrt{\frac{d}{b_2}}\right)\eta_t\beta\right)\mathbb{E}[\|w^t - \bar{w}^t\|] + \left(\frac{2\sqrt{\tau(t)}}{nN} + \mu\beta + \frac{2\sqrt{\tau(t)}}{nN}\sqrt{\frac{d}{b_2}}\right)\eta_t.$$

Let $a_1 = \left(1 + \sqrt{\frac{d}{b_2}}\right)\beta$ and $a_4(t) = \frac{2\sqrt{\tau(t)}}{nN} + \mu\beta + \frac{2\sqrt{\tau(t)}}{nN}\sqrt{\frac{d}{b_2}}$. Taking summation from $t = 1$ to $T - 1$, we deduce that

$$\mathbb{E}[\|w^T - \bar{w}^T\|]$$

$$\leq \sum_{t=1}^{T-1}\left(\prod_{s=t+1}^{T-1}(1 + a_1\eta_s)\right)a_4(t)\eta_t$$

$$\leq \sum_{t=1}^{T-1}\exp\left(\sum_{s=t+1}^{T-1}a_1\eta_s\right)a_4(T-1)\eta_t$$

$$\leq \sum_{t=1}^{T-1}\exp\left(a_1\eta_1\sum_{s=1}^{T-1}s^{-1}\right)a_4(T-1)\eta_t$$

$$\leq \exp\left(a_1\eta_1\sum_{s=1}^{T-1}s^{-1}\right)a_4(T-1)\eta_1\sum_{t=1}^{T-1}t^{-1}$$

$$\leq (e(T-1))^{a_1\eta_1}a_4(T-1)\eta_1\log(e(T-1))$$

$$\leq \mathcal{O}\left(\left((nN)^{-1}\sqrt{\log T} + 1\right)\mu T^{\frac{1}{4}}\log T\right).$$

$\square$

**Proof of Corollary 2**: We integrate Theorem 1 (b) and Theorem 3 to obtain that

$$\left|\mathbb{E}\left[F(w^T) - F_S(w^T)\right]\right|$$

$$\leq \frac{(4\theta)^\theta K}{nN}\sum_{i=1}^{N}\sum_{j=1}^{n}\mathbb{E}\left[\|w^T - \bar{w}^T\|\right] + 2\mathbb{E}\left[F_S(w^T)\right]$$

$$= (4\theta)^\theta K\mathbb{E}\left[\|w^T - \bar{w}^T\|\right] + 2\mathbb{E}\left[F_S(w^T)\right]$$

$$\leq \mathcal{O}\left(\left((nN)^{-1}\sqrt{\log T} + 1\right)(4\theta)^\theta(nN)^{-1}T^{\frac{1}{4}}\log T\right).$$

The proof is complete. $\square$

**Proof of Theorem 4**: According to Equation (8),

$$\mathbb{E}[F_S(w^{t+1}) - F_S(w^t)]$$

$$\leq -\left(\frac{1}{2} - \frac{d}{2b_2}\right)\alpha\eta_t\mathbb{E}\left[F_S(w^t) - F_S(w(S))\right] + \frac{\mu^2\beta^3\eta_t^2}{4} + \frac{\mu^2\beta^2\eta_t}{8},$$

that is

$$\mathbb{E}[F_S(w^{t+1}) - F_S(w(S))]$$
$$\leq \left(1 - \left(\frac{1}{2} - \frac{d}{2b_2}\right)\alpha\eta_t\right)\mathbb{E}\left[F_S(w^t) - F(w(S))\right] + \frac{\mu^2\beta^3\eta_t^2}{4} + \frac{\mu^2\beta^2\eta_t}{8}$$
$$= \left(1 - \left(\frac{1}{2} - \frac{d}{2b_2}\right)\frac{\alpha\eta_1}{t}\right)\mathbb{E}\left[F_S(w^t) - F(w(S))\right] + \frac{\mu^2\beta^3\eta_1^2}{4}t^{-2} + \frac{\mu^2\beta^2\eta_1}{8}t^{-1}.$$

We multiply both sides of the above inequality by $t\left(t - \left(\frac{1}{4} - \frac{d}{4b_2}\right)\alpha\eta_1\right)$ to get

$$t\left(t - \left(\frac{1}{4} - \frac{d}{4b_2}\right)\alpha\eta_1\right)\mathbb{E}[F_S(w^{t+1}) - F(w(S))]$$
$$\leq \left(t - \left(\frac{1}{4} - \frac{d}{4b_2}\right)\alpha\eta_1\right)\left(t - \left(\frac{1}{2} - \frac{d}{2b_2}\right)\alpha\eta_1\right)\mathbb{E}\left[F_S(w^t) - F(w(S))\right]$$
$$+ \frac{\mu^2\beta^3\eta_1^2}{4} + \frac{\mu^2\beta^2\eta_1}{8}t$$
$$\leq \left(1 - \left(\frac{1}{4} - \frac{d}{4b_2}\right)\alpha\eta_1\right)\left(1 - \left(\frac{1}{2} - \frac{d}{2b_2}\right)\alpha\eta_1\right)\mathbb{E}\left[F_S(w^1) - F(w(S))\right]$$
$$+ \frac{\mu^2\beta^3\eta_1^2 t}{4} + \frac{\mu^2\beta^2\eta_1}{8}\sum_{t'=1}^{t}t'$$
$$\leq \left(1 - \left(\frac{1}{4} - \frac{d}{4b_2}\right)\alpha\eta_1\right)\left(1 - \left(\frac{1}{2} - \frac{d}{2b_2}\right)\alpha\eta_1\right)\mathbb{E}\left[F_S(w^1) - F(w(S))\right]$$
$$+ \frac{\mu^2\beta^3\eta_1^2 t}{4} + \frac{\mu^2\beta^2\eta_1}{16}t(t+1).$$

Therefore,

$$\mathbb{E}[F_S(w^T) - F_S(w(S))]$$
$$\leq \frac{\left(1 - \left(\frac{1}{4} - \frac{d}{4b_2}\right)\alpha\eta_1\right)\left(1 - \left(\frac{1}{2} - \frac{d}{2b_2}\right)\alpha\eta_1\right)}{(T-1)\left(T - 1 - \left(\frac{1}{4} - \frac{d}{4b_2}\right)\alpha\eta_1\right)}\mathbb{E}\left[F_S(w^1) - F(w(S))\right]$$
$$+ \frac{\mu^2\beta^3\eta_1^2}{4\left(T - 1 - \left(\frac{1}{4} - \frac{d}{4b_2}\right)\alpha\eta_1\right)} + \frac{\mu^2\beta^2\eta_1 T}{16\left(T - 1 - \left(\frac{1}{4} - \frac{d}{4b_2}\right)\alpha\eta_1\right)}$$
$$= \mathcal{O}\left(T^{-2} + \mu^2\right).$$

The optimization bound is given. By integrating this optimization upper bound and the generalization upper bound in Corollary 2, we can get the following expected excess risk bound

$$|\mathbb{E}\left[F(w^T) - F(w^*)\right]|$$
$$\leq |\mathbb{E}\left[F(w^T) - F_S(w^T)\right]| + |\mathbb{E}\left[F_S(w^T) - F_S(w(S))\right]|$$
$$\leq \mathcal{O}\left(T^{-2} + \mu^2 + \left((nN)^{-1}\sqrt{\log T} + 1\right)(4\theta)^\theta\mu T^{a_1\eta_1}\log T + \mathbb{E}[F_S(w^T)]\right).$$

$\square$

## B.5. Proofs of Theorem 5 and Theorem 6

**Proof of Theorem 5**: Let $S^{(j_i)} = S^{(n_N)} = \{S_i\}_{i=1}^{N-1} \cup S_N^{(n)}$. The update for asnchronous case can be reformulated as

$$w^{t+1} = w^t - \frac{\eta_t}{b_1 N}\sum_{i\in[N]}\sum_{j\in[n]}\alpha_{ij}^{t_i}\tilde{\nabla}f_i\left(w_i^{t_i}; z_{ij}^{t_i}, \left\{v_{i,l}^{t_i}\right\}_{l=1}^{b_2}, \mu\right). \tag{9}$$

For the sake of simplicity, we denote $\tilde{\nabla} f_i \left( w_i^{t_i}; z_{ij}^{t_i}, \left\{ v_{i,l}^{t_i} \right\}_{l=1}^{b_2}, \mu \right)$ as $\tilde{\nabla} f_i \left( w_i^{t_i}; z_{ij}^{t_i} \right)$. According to the new formulation, we can get

$$\| w^{t+1} - \bar{w}^{t+1} \|$$

$$= \left\| w^t - \bar{w}^t - \frac{\eta_t}{b_1 N} \sum_{i \in [N]} \sum_{j \in [n]} \alpha_{ij}^{t_i} \left( \tilde{\nabla} f_i \left( w_i^{t_i}; z_{ij}^{t_i} \right) - \tilde{\nabla} f_i \left( \bar{w}_i^{t_i}; \bar{z}_{ij}^{t_i} \right) \right) \right\|$$

$$\leq \left\| w^t - \bar{w}^t - \frac{\eta_t}{b_1 N} \sum_{i \in [N]} \sum_{j \in [n]} \alpha_{ij}^{t_i} \left( \nabla f_i \left( w_i^{t_i}; z_{ij}^{t_i} \right) - \nabla f_i \left( \bar{w}_i^{t_i}; \bar{z}_{ij}^{t_i} \right) \right) \right\| \tag{10}$$

$$+ \left\| \frac{\eta_t}{b_1 N} \sum_{i \in [N]} \sum_{j \in [n]} \alpha_{ij}^{t_i} \left( \tilde{\nabla} f_i \left( w_i^{t_i}; z_{ij}^{t_i} \right) - \tilde{\nabla} f_i \left( \bar{w}_i^{t_i}; \bar{z}_{ij}^{t_i} \right) - \nabla f_i \left( w_i^{t_i}; z_{ij}^{t_i} \right) + \nabla f_i \left( \bar{w}_i^{t_i}; \bar{z}_{ij}^{t_i} \right) \right) \right\|. \tag{11}$$

In the sequel, we seperately give the upper bounds for (9) and (10). Let $P_t = \{ (i,j) | i \in [N-1], j \in [n] \text{ or } i = N, j \in [n-1] \}$, then

$(10)$

$$\leq \| w^t - \bar{w}^t \| + \frac{\eta_t}{b_1 N} \alpha_{Nn}^{t_N} \| \nabla f_N(w_N^{t_N}; z_{Nn}^{t_N}) - \nabla f_N(\bar{w}_N^{t_N}; \bar{z}_{Nn}^{t_N}) \|$$

$$+ \frac{\eta_t}{b_1 N} \sum_{P_t} \alpha_{ij}^{t_i} \| \nabla f_i(w_i^{t_i}; z_{ij}^{t_i}) - \nabla f_i(\bar{w}_i^{t_i}; \bar{z}_{ij}^{t_i}) \|$$

$$\leq \| w^t - \bar{w}^t \| + \frac{2\eta_t}{b_1 N} \alpha_{Nn}^{t_N} \| \nabla f_N(w_N^{t_N}; z_{Nn}^{t_N}) \| + \frac{\beta \eta_t}{b_1 N} \sum_{P_t} \alpha_{ij}^{t_i} \| w_i^{t_i} - \bar{w}_i^{t_i} \|$$

$$\leq \| w^t - \bar{w}^t \| + \frac{2\eta_t}{b_1 N} \alpha_{Nn}^{t_N} \| \nabla f_N(w_N^{t_N}; z_{Nn}^{t_N}) \| + \frac{\beta \eta_t}{b_1 N} \sum_{P_t} \alpha_{ij}^{t_i} \left( \| w_i^{t_i} - w^t - \bar{w}_i^{t_i} + \bar{w}^t \| + \| w^t - \bar{w}^t \| \right)$$

$$\leq \left( 1 + \frac{\beta \eta_t}{b_1 N} \sum_{P_t} \alpha_{ij}^{t_i} \right) \| w^t - \bar{w}^t \| + \frac{2\eta_t}{b_1 N} \alpha_{Nn}^{t_N} \| \nabla f_N(w_N^{t_N}; z_{Nn}^{t_N}) \|$$

$$+ \frac{\beta \eta_t}{b_1 N} \sum_{P_t} \alpha_{ij}^{t_i} \left\| \sum_{t'=t_i}^{t-1} \left( \frac{\eta_{t'}}{b_1 N} \sum_{i' \in [N]} \sum_{j' \in [n]} \alpha_{i'j'}^{t'} \left( \tilde{\nabla} f_{i'} \left( w_{i'}^{t'_{i'}}; z_{i'j'}^{t'_{i'}} \right) - \tilde{\nabla} f_{i'} \left( \bar{w}_{i'}^{t'_{i'}}; \bar{z}_{i'j'}^{t'_{i'}} \right) \right) \right) \right\|$$

$$\leq \left( 1 + \frac{\beta \eta_t}{b_1 N} \sum_{P_t} \alpha_{ij}^{t_i} \right) \| w^t - \bar{w}^t \| + \frac{2\eta_t}{b_1 N} \alpha_{Nn}^{t_N} \| \nabla f_N(w_N^{t_N}; z_{Nn}^{t_N}) \|$$

$$+ \frac{\beta \eta_t}{b_1 N} \sum_{P_t} \alpha_{ij}^{t_i} \left( \sum_{t'=t_i}^{t-1} \left( \frac{\eta_{t'}}{b_1 N} \sum_{i' \in [N]} \sum_{j' \in [n]} \alpha_{i'j'}^{t'} \left( \left\| \nabla f_{i'} \left( w_{i'}^{t'_{i'}}; z_{i'j'}^{t'_{i'}} \right) - \nabla f_{i'} \left( \bar{w}_{i'}^{t'_{i'}}; \bar{z}_{i'j'}^{t'_{i'}} \right) \right\| \right. \right. \right.$$

$$\left. \left. \left. + \left\| \tilde{\nabla} f_{i'} \left( w_{i'}^{t'_{i'}}; z_{i'j'}^{t'_{i'}} \right) - \tilde{\nabla} f_{i'} \left( \bar{w}_{i'}^{t'_{i'}}; \bar{z}_{i'j'}^{t'_{i'}} \right) - \nabla f_{i'} \left( w_{i'}^{t'_{i'}}; z_{i'j'}^{t'_{i'}} \right) + \nabla f_{i'} \left( \bar{w}_{i'}^{t'_{i'}}; \bar{z}_{i'j'}^{t'_{i'}} \right) \right\| \right) \right) \right)$$

$$\leq \left( 1 + \frac{\beta \eta_t}{b_1 N} \sum_{P_t} \alpha_{ij}^{t_i} \right) \| w^t - \bar{w}^t \| + \frac{2\eta_t}{b_1 N} \alpha_{Nn}^{t_N} \| \nabla f_N(w_N^{t_N}; z_{Nn}^{t_N}) \|$$

$$+ \frac{\beta \eta_t}{b_1 N} \sum_{P_t} \alpha_{ij}^{t_i} \left( \sum_{t'=t_i}^{t-1} \left( \frac{\eta_{t'}}{b_1 N} \sum_{i' \in [N]} \sum_{j' \in [n]} \alpha_{i'j'}^{t'} \left( \frac{\mu \beta}{b_2} \sum_{l=1}^{b_2} \| v_{i',l}^{t'_{i'}} \|^3 + 2 \left\| \nabla f_{i'} \left( w_{i'}^{t'_{i'}}; z_{i'j'}^{t'_{i'}} \right) \right\| \right. \right. \right.$$

$$\left. \left. \left. + \left\| \frac{1}{b_2} \sum_{l=1}^{b_2} \left\langle \nabla f_{i'} \left( w_{i'}^{t'_{i'}}; z_{i'j'}^{t'_{i'}} \right) - \nabla f_{i'} \left( \bar{w}_{i'}^{t'_{i'}}; \bar{z}_{i'j'}^{t'_{i'}} \right), v_{i',l}^{t'_{i'}} \right\rangle v_{i',l}^{t'_{i'}} - \nabla f_{i'} \left( w_{i'}^{t'_{i'}}; z_{i'j'}^{t'_{i'}} \right) + \nabla f_{i'} \left( \bar{w}_{i'}^{t'_{i'}}; \bar{z}_{i'j'}^{t'_{i'}} \right) \right\| \right) \right) \right).$$

$(11)$

$$
\leq \frac{\eta_t}{b_1 N} \sum_{P_t} \alpha_{ij}^{t_i} \left\| \tilde{\nabla} f_i \left( w_i^{t_i}; z_{ij}^{t_i} \right) - \tilde{\nabla} f_i \left( \bar{w}_i^{t_i}; z_{ij}^{t_i} \right) - \nabla f_i \left( w_i^{t_i}; z_{ij}^{t_i} \right) + \nabla f_i \left( \bar{w}_i^{t_i}; z_{ij}^{t_i} \right) \right\|
$$

$$
+ \frac{\eta_t}{b_1 N} \alpha_{Nn}^{t_N} \left\| \tilde{\nabla} f_N \left( w_N^{t_N}; z_{Nn}^{t_N} \right) - \tilde{\nabla} f_N \left( \bar{w}_N^{t_N}; \bar{z}_{Nn}^{t_N} \right) - \nabla f_N \left( w_N^{t_N}; z_{Nn}^{t_N} \right) + \nabla f_N \left( \bar{w}_N^{t_N}; \bar{z}_{Nn}^{t_N} \right) \right\|
$$

$$
\leq \frac{\eta_t}{b_1 N} \sum_{i \in [N]} \sum_{j \in [n]} \alpha_{ij}^{t_i} \frac{\mu \beta}{b_2} \sum_{l=1}^{b_2} \| v_{i,l}^{t_i} \|^3 + \frac{\eta_{t_i}}{b_1 N} \sum_{P_t} \alpha_{ij}^{t_i}
$$

$$
\left\| \frac{1}{b_2} \sum_{l=1}^{b_2} \left\langle \nabla f_i(w_i^{t_i}; z_{ij}^{t_i}) - \nabla f_i(\bar{w}_i^{t_i}; z_{ij}^{t_i}), v_{i,l}^{t_i} \right\rangle v_{i,l}^{t_i} - \nabla f_i \left( w_i^{t_i}; z_{ij}^{t_i} \right) + \nabla f_i \left( \bar{w}_i^{t_i}; z_{ij}^{t_i} \right) \right\| + \frac{\eta_t}{b_1 N} \alpha_{Nn}^{t_N}
$$

$$
\left\| \frac{1}{b_2} \sum_{l=1}^{b_2} \langle \nabla f_N(w_N^{t_N}; z_{Nn}^{t_N}) - \nabla f_N(\bar{w}_N^{t_N}; \bar{z}_{Nn}^{t_N}), v_{N,l}^{t_N} \rangle v_{N,l}^{t_N} - \nabla f_N \left( w_N^{t_N}; z_{Nn}^{t_N} \right) + \nabla f_N \left( \bar{w}_N^{t_N}; \bar{z}_{Nn}^{t_N} \right) \right\|.
$$

Then, combining the above two inequalities, we obtain that

$$
\| w^{t+1} - \bar{w}^{t+1} \|
$$

$$
\leq \left( 1 + \frac{\beta \eta_t}{b_1 N} \sum_{P_t} \alpha_{ij}^{t_i} \right) \| w^t - \bar{w}^t \| + \frac{2 \eta_t}{b_1 N} \alpha_{Nn}^{t_N} \| \nabla f_N(w_N^{t_N}; z_{Nn}^{t_N}) \|
$$

$$
+ \frac{\beta \eta_t}{b_1 N} \sum_{P_t} \alpha_{ij}^{t_i} \left( \sum_{t'=t_i}^{t-1} \left( \frac{\eta_{t'}}{b_1 N} \sum_{i' \in [N]} \sum_{j' \in [n]} \alpha_{i'j'}^{t'} \left( \frac{\mu \beta}{b_2} \sum_{l=1}^{b_2} \| v_{i',l}^{t'} \|^3 + 2 \left\| \nabla f_{i'} \left( w_{i'}^{t'}; z_{i'j'}^{t'} \right) \right\| \right. \right. \right.
$$

$$
\left. \left. \left. + \left\| \frac{1}{b_2} \sum_{l=1}^{b_2} \left\langle \nabla f_{i'} \left( w_{i'}^{t'}; z_{i'j'}^{t'} \right) - \nabla f_{i'} \left( \bar{w}_{i'}^{t'}; \bar{z}_{i'j'}^{t'} \right), v_{i',l}^{t'} \right\rangle v_{i',l}^{t'} - \nabla f_{i'} \left( w_{i'}^{t'}; z_{i'j'}^{t'} \right) + \nabla f_{i'} \left( \bar{w}_{i'}^{t'}; \bar{z}_{i'j'}^{t'} \right) \right\| \right) \right) \right)
$$

$$
+ \frac{\eta_t}{b_1 N} \sum_{i \in [N]} \sum_{j \in [n]} \alpha_{ij}^{t_i} \frac{\mu \beta}{b_2} \sum_{l=1}^{b_2} \| v_{i,l}^{t_i} \|^3 + \frac{\eta_{t_i}}{b_1 N} \sum_{P_t} \alpha_{ij}^{t_i}
$$

$$
\left\| \frac{1}{b_2} \sum_{l=1}^{b_2} \left\langle \nabla f_i(w_i^{t_i}; z_{ij}^{t_i}) - \nabla f_i(\bar{w}_i^{t_i}; z_{ij}^{t_i}), v_{i,l}^{t_i} \right\rangle v_{i,l}^{t_i} - \nabla f_i \left( w_i^{t_i}; z_{ij}^{t_i} \right) + \nabla f_i \left( \bar{w}_i^{t_i}; z_{ij}^{t_i} \right) \right\| + \frac{\eta_t}{b_1 N} \alpha_{Nn}^{t_N}
$$

$$
\left\| \frac{1}{b_2} \sum_{l=1}^{b_2} \langle \nabla f_N(w_N^{t_N}; z_{Nn}^{t_N}) - \nabla f_N(\bar{w}_N^{t_N}; \bar{z}_{Nn}^{t_N}), v_{N,l}^{t_N} \rangle v_{N,l}^{t_N} - \nabla f_N \left( w_N^{t_N}; z_{Nn}^{t_N} \right) + \nabla f_N \left( \bar{w}_N^{t_N}; \bar{z}_{Nn}^{t_N} \right) \right\|.
$$

Define $J_i = \{ J_i^1, ..., J_i^t \}$, $J_i^{t'} = \{ z_{i,1}^{t'}, ..., z_{i,b_1}^{t'} \}$, $t' \in [t], t \in \mathbb{N}, i \in [N]$. Taking conditional expectation w.r.t. $J_i$, we derive

$$
\mathbb{E}_{J_i}[\| w^{t+1} - \bar{w}^{t+1} \|]
$$

$$
\leq \left( 1 + \frac{\beta \eta_t}{b_1 N} \sum_{P_t} \mathbb{E}_{J_i}[\alpha_{ij}^{t_i}] \right) \| w^t - \bar{w}^t \| + \frac{2 \eta_t}{b_1 N} \mathbb{E}_{J_i}[\alpha_{Nn}^{t_N}] \| \nabla f_N(w_N^{t_N}; z_{Nn}^{t_N}) \|
$$

$$
+ \frac{\beta \eta_t}{b_1 N} \sum_{P_t} \mathbb{E}_{J_i}[(\alpha_{ij}^{t_i})^2] \left( \frac{\eta_{t_i}}{b_1 N} \left( \frac{\mu \beta}{b_2} \sum_{l=1}^{b_2} \| v_{i,l}^{t_i} \|^3 + 2 \left\| \nabla f_i \left( w_i^{t_i}; z_{ij}^{t_i} \right) \right\| \right. \right.
$$

$$
\left. \left. + \left\| \frac{1}{b_2} \sum_{l=1}^{b_2} \left\langle \nabla f_i \left( w_i^{t_i}; z_{ij}^{t_i} \right) - \nabla f_i \left( \bar{w}_i^{t_i}; \bar{z}_{ij}^{t_i} \right), v_{i,l}^{t_i} \right\rangle v_{i,l}^{t_i} - \nabla f_i \left( w_i^{t_i}; z_{ij}^{t_i} \right) + \nabla f_i \left( \bar{w}_i^{t_i}; \bar{z}_{ij}^{t_i} \right) \right\| \right) \right)
$$

$$
+ \frac{\beta \eta_t}{b_1 N} \sum_{P_t} \left( \mathbb{E}_{J_i}[\alpha_{ij}^{t_i}] \right)^2 \left( \frac{\eta_{t_i}}{b_1 N} \sum_{\substack{i' \in [N] \\ i' \neq i}} \sum_{\substack{j' \in [n] \\ j' \neq j}} \left( \frac{\mu \beta}{b_2} \sum_{l=1}^{b_2} \| v_{i',l}^{t_{i'}} \|^3 + 2 \left\| \nabla f_{i'} \left( w_{i'}^{t_{i'}}; z_{i'j'}^{t_{i'}} \right) \right\| \right. \right.
$$

$$
\left. \left. + \left\| \frac{1}{b_2} \sum_{l=1}^{b_2} \left\langle \nabla f_{i'} \left( w_{i'}^{t_{i'}}; z_{i'j'}^{t_{i'}} \right) - \nabla f_{i'} \left( \bar{w}_{i'}^{t_{i'}}; \bar{z}_{i'j'}^{t_{i'}} \right), v_{i',l}^{t_{i'}} \right\rangle v_{i',l}^{t_{i'}} - \nabla f_{i'} \left( w_{i'}^{t_{i'}}; z_{i'j'}^{t_{i'}} \right) + \nabla f_{i'} \left( \bar{w}_{i'}^{t_{i'}}; \bar{z}_{i'j'}^{t_{i'}} \right) \right\| \right) \right)
$$

$$+ \frac{\beta \eta_t}{b_1 N} \sum_{P_t} \left( \mathbb{E}_{J_i}[\alpha_{ij}^{t_i}] \right)^2 \left( \sum_{t'=t_i+1}^{t-1} \left( \frac{\eta_{t'}}{b_1 N} \sum_{i' \in [N]} \sum_{j' \in [n]} \left( \frac{\mu \beta}{b_2} \sum_{l=1}^{b_2} \|v_{i',l}^{t'_{i'}}\|^3 + 2 \left\| \nabla f_{i'} \left( w_{i'}^{t'_{i'}}; z_{i'j'}^{t'_{i'}} \right) \right\| \right. \right.$$

$$+ \left. \left. \left\| \frac{1}{b_2} \sum_{l=1}^{b_2} \left\langle \nabla f_{i'} \left( w_{i'}^{t'_{i'}}; z_{i'j'}^{t'_{i'}} \right) - \nabla f_{i'} \left( \bar{w}_{i'}^{t'_{i'}}; \bar{z}_{i'j'}^{t'_{i'}} \right), v_{i',l}^{t'_{i'}} \right\rangle v_{i',l}^{t'_{i'}} - \nabla f_{i'} \left( w_{i'}^{t'_{i'}}; z_{i'j'}^{t'_{i'}} \right) + \nabla f_{i'} \left( \bar{w}_{i'}^{t'_{i'}}; \bar{z}_{i'j'}^{t'_{i'}} \right) \right\| \right) \right) \right)$$

$$+ \frac{\eta_t}{b_1 N} \sum_{i \in [N]} \sum_{j \in [n]} \mathbb{E}_{J_i}[\alpha_{ij}^{t_i}] \frac{\mu \beta}{b_2} \sum_{l=1}^{b_2} \|v_{i,l}^{t_i}\|^3 + \frac{\eta_{t_i}}{b_1 N} \sum_{P_t} \mathbb{E}_{J_i}[\alpha_{ij}^{t_i}]$$

$$\left\| \frac{1}{b_2} \sum_{l=1}^{b_2} \left\langle \nabla f_i(w_i^{t_i}; z_{ij}^{t_i}) - \nabla f_i(\bar{w}_i^{t_i}; z_{ij}^{t_i}), v_{i,l}^{t_i} \right\rangle v_{i,l}^{t_i} - \nabla f_i \left( w_i^{t_i}; z_{ij}^{t_i} \right) + \nabla f_i \left( \bar{w}_i^{t_i}; z_{ij}^{t_i} \right) \right\| + \frac{\eta_t}{b_1 N} \mathbb{E}_{J_i}[\alpha_{Nn}^{t_N}]$$

$$\left\| \frac{1}{b_2} \sum_{l=1}^{b_2} \left\langle \nabla f_N(w_N^{t_N}; z_{Nn}^{t_N}) - \nabla f_N(\bar{w}_N^{t_N}; \bar{z}_{Nn}^{t_N}), v_{N,l}^{t_N} \right\rangle v_{N,l}^{t_N} - \nabla f_N \left( w_N^{t_N}; z_{Nn}^{t_N} \right) + \nabla f_N \left( \bar{w}_N^{t_N}; \bar{z}_{Nn}^{t_N} \right) \right\|$$

$$\leq (1 + \beta \eta_t) \|w^t - \bar{w}^t\| + \frac{2\eta_t}{nN} \|\nabla f_N(w_N^{t_N}; z_{Nn}^{t_N})\|$$

$$+ \frac{\beta \eta_t}{b_1 n N^2} \left( 1 + \frac{b_1 - 1}{n} \right) \sum_{P_t} \eta_{t_i} \left( \frac{\mu \beta}{b_2} \sum_{l=1}^{b_2} \|v_{i,l}^{t_i}\|^3 + 2 \left\| \nabla f_i \left( w_i^{t_i}; z_{ij}^{t_i} \right) \right\| \right.$$

$$+ \left. \left\| \frac{1}{b_2} \sum_{l=1}^{b_2} \left\langle \nabla f_i \left( w_i^{t_i}; z_{ij}^{t_i} \right) - \nabla f_i \left( \bar{w}_i^{t_i}; \bar{z}_{ij}^{t_i} \right), v_{i,l}^{t_i} \right\rangle v_{i,l}^{t_i} - \nabla f_i \left( w_i^{t_i}; z_{ij}^{t_i} \right) + \nabla f_i \left( \bar{w}_i^{t_i}; \bar{z}_{ij}^{t_i} \right) \right\| \right)$$

$$+ \frac{\beta \eta_t}{n^2 N^2} \sum_{P_t} \eta_{t_i} \left( \sum_{\substack{i' \in [N], \, j' \in [n], \\ i' \neq i \quad j' \neq j}} \left( \frac{\mu \beta}{b_2} \sum_{l=1}^{b_2} \|v_{i',l}^{t_{i'}}\|^3 + 2 \left\| \nabla f_{i'} \left( w_{i'}^{t_{i'}}; z_{i'j'}^{t_{i'}} \right) \right\| \right. \right.$$

$$+ \left. \left. \left\| \frac{1}{b_2} \sum_{l=1}^{b_2} \left\langle \nabla f_{i'} \left( w_{i'}^{t_{i'}}; z_{i'j'}^{t_{i'}} \right) - \nabla f_{i'} \left( \bar{w}_{i'}^{t_{i'}}; \bar{z}_{i'j'}^{t_{i'}} \right), v_{i',l}^{t_{i'}} \right\rangle v_{i',l}^{t_{i'}} - \nabla f_{i'} \left( w_{i'}^{t_{i'}}; z_{i'j'}^{t_{i'}} \right) + \nabla f_{i'} \left( \bar{w}_{i'}^{t_{i'}}; \bar{z}_{i'j'}^{t_{i'}} \right) \right\| \right) \right)$$

$$+ \frac{\beta \eta_t}{n^2 N^2} \sum_{P_t} \left( \sum_{t'=t_i+1}^{t-1} \left( \eta_{t'} \sum_{i' \in [N]} \sum_{j' \in [n]} \left( \frac{\mu \beta}{b_2} \sum_{l=1}^{b_2} \|v_{i',l}^{t'_{i'}}\|^3 + 2 \left\| \nabla f_{i'} \left( w_{i'}^{t'_{i'}}; z_{i'j'}^{t'_{i'}} \right) \right\| \right. \right. \right.$$

$$+ \left. \left. \left. \left\| \frac{1}{b_2} \sum_{l=1}^{b_2} \left\langle \nabla f_{i'} \left( w_{i'}^{t'_{i'}}; z_{i'j'}^{t'_{i'}} \right) - \nabla f_{i'} \left( \bar{w}_{i'}^{t'_{i'}}; \bar{z}_{i'j'}^{t'_{i'}} \right), v_{i',l}^{t'_{i'}} \right\rangle v_{i',l}^{t'_{i'}} - \nabla f_{i'} \left( w_{i'}^{t'_{i'}}; z_{i'j'}^{t'_{i'}} \right) + \nabla f_{i'} \left( \bar{w}_{i'}^{t'_{i'}}; \bar{z}_{i'j'}^{t'_{i'}} \right) \right\| \right) \right) \right)$$

$$+ \frac{\eta_t}{nN} \sum_{i \in [N]} \sum_{j \in [n]} \frac{\mu \beta}{b_2} \sum_{l=1}^{b_2} \|v_{i,l}^{t_i}\|^3$$

$$+ \frac{\eta_t}{nN} \sum_{P_t} \left\| \frac{1}{b_2} \sum_{l=1}^{b_2} \left\langle \nabla f_i(w_i^{t_i}; z_{ij}^{t_i}) - \nabla f_i(\bar{w}_i^{t_i}; z_{ij}^{t_i}), v_{i,l}^{t_i} \right\rangle v_{i,l}^{t_i} - \nabla f_i \left( w_i^{t_i}; z_{ij}^{t_i} \right) + \nabla f_i \left( \bar{w}_i^{t_i}; z_{ij}^{t_i} \right) \right\|$$

$$+ \frac{\eta_t}{nN} \left\| \frac{1}{b_2} \sum_{l=1}^{b_2} \left\langle \nabla f_N(w_N^{t_N}; z_{Nn}^{t_N}) - \nabla f_N(\bar{w}_N^{t_N}; \bar{z}_{Nn}^{t_N}), v_{N,l}^{t_N} \right\rangle v_{N,l}^{t_N} - \nabla f_N \left( w_N^{t_N}; z_{Nn}^{t_N} \right) + \nabla f_N \left( \bar{w}_N^{t_N}; \bar{z}_{Nn}^{t_N} \right) \right\|.$$

Further taking expectation w.r.t. randomness and utilizing Lemmas 4, 5, we obtain that

$$\mathbb{E} \left[ \|w^{t+1} - \bar{w}^{t+1}\| \right]$$

$$\leq (1 + \beta \eta_t) \mathbb{E} \left[ \|w^t - \bar{w}^t\| \right] + \frac{2\eta_t}{nN} \mathbb{E} \left[ \|\nabla f_N(w_N^{t_N}; z_{Nn}^{t_N})\| \right]$$

$$+ \frac{\beta \eta_t}{b_1 N} \left( 1 + \frac{b_1 - 1}{n} \right) \eta_{t-t_0} \left( \mu \beta \mathbb{E} \left[ \|v_{i,l}^{t_i}\|^3 \right] + \left( 2 + 2\sqrt{\frac{d}{b_2}} \right) \mathbb{E} \left[ \|\nabla f_i \left( w_i^{t_i}; z_{ij}^{t_i} \right)\| \right] \right)$$

$$+ \beta \eta_t \eta_{t-t_0} \left( \mu\beta \mathbb{E}\left[ \left\| v_{i',l}^{t_{i'}} \right\|^3 \right] + \left( 2 + 2\sqrt{\frac{d}{b_2}} \right) \mathbb{E}\left[ \left\| \nabla f_{i'}\left( w_{i'}^{t_{i'}}; z_{i'j'}^{t_{i'}} \right) \right\| \right] \right)$$

$$+ \beta \eta_t (t_0 - 1) \eta_{t-t_0} \left( \mu\beta \mathbb{E}\left[ \left\| v_{i',l}^{t'_{i'}} \right\|^3 \right] + \left( 2 + 2\sqrt{\frac{d}{b_2}} \right) \mathbb{E}\left[ \left\| \nabla f_{i'}\left( w_{i'}^{t'_{i'}}; z_{i'j'}^{t'_{i'}} \right) \right\| \right] \right) + \mu\beta\eta_t \mathbb{E}\left[ \left\| v_{i,l}^{t_i} \right\|^3 \right]$$

$$+ \frac{nN-1}{nN} \eta_t \mathbb{E}\left[ \left\| \nabla f_i\left( w_i^{t_i}; z_{ij}^{t_i} \right) - \nabla f_i\left( \bar{w}_i^{t_i}; z_{ij}^{t_i} \right) \right\| \right] + \frac{2\eta_t}{nN} \sqrt{\frac{d}{b_2}} \mathbb{E}\left[ \left\| \nabla f_N\left( \bar{w}_N^{t_N}; z_{Nn}^{t_N} \right) \right\| \right]$$

$$\leq (1 + \beta\eta_t) \mathbb{E}\left[ \left\| w^t - \bar{w}^t \right\| \right] + \left( \mu\beta^2 \left( \frac{1}{b_1 N} \left( 1 + \frac{b_1 - 1}{n} \right) + t_0 \right) \eta_t \eta_{t-t_0} + \mu\beta\eta_t \right) \mathbb{E}\left[ \left\| v_{i,l}^{t_i} \right\|^3 \right]$$

$$+ \left( \beta \left( 2 + 2\sqrt{\frac{d}{b_2}} \right) \left( \left( \frac{1}{b_1 N} \left( 1 + \frac{b_1 - 1}{n} \right) + t_0 \right) \eta_t \eta_{t-t_0} + \frac{1}{nN} \eta_t \right) \right) \mathbb{E}\left[ \left\| \nabla f_i\left( w_i^{t_i}; z_{ij}^{t_i} \right) \right\| \right]$$

$$+ \beta\eta_t \mathbb{E}\left[ \left\| w_i^{t_i} - \bar{w}_i^{t_i} \right\| \right]$$

$$\leq (1 + \beta\eta_t) \mathbb{E}\left[ \left\| w^t - \bar{w}^t \right\| \right] + \left( \mu\beta^2 \left( \frac{1}{b_1 N} \left( 1 + \frac{b_1 - 1}{n} \right) + t_0 \right) \eta_t \eta_{t-t_0} + \mu\beta\eta_t \right) \mathbb{E}\left[ \left\| v_{i,l}^{t_i} \right\|^3 \right]$$

$$+ \left( \beta \left( 2 + 2\sqrt{\frac{d}{b_2}} \right) \left( \left( \frac{1}{b_1 N} \left( 1 + \frac{b_1 - 1}{n} \right) + t_0 \right) \eta_t \eta_{t-t_0} + \frac{1}{nN} \eta_t \right) \right) \mathbb{E}\left[ \left\| \nabla f_i\left( w_i^{t_i}; z_{ij}^{t_i} \right) \right\| \right]$$

$$+ \beta\eta_t \left( \mathbb{E}\left[ \left\| w_i^{t_i} - \bar{w}_i^{t_i} - w^t + \bar{w}^t \right\| \right] + \mathbb{E}\left[ \left\| w^t - \bar{w}^t \right\| \right] \right)$$

$$\leq (1 + 2\beta\eta_t) \mathbb{E}\left[ \left\| w^t - \bar{w}^t \right\| \right] + \left( \mu\beta^2 \left( \frac{1}{b_1 N} \left( 1 + \frac{b_1 - 1}{n} \right) + t_0 \right) \eta_t \eta_{t-t_0} + \mu\beta\eta_t \right) \mathbb{E}\left[ \left\| v_{i,l}^{t_i} \right\|^3 \right]$$

$$+ \left( \beta \left( 2 + 2\sqrt{\frac{d}{b_2}} \right) \left( \left( \frac{1}{b_1 N} \left( 1 + \frac{b_1 - 1}{n} \right) + t_0 \right) \eta_t \eta_{t-t_0} + \frac{1}{nN} \eta_t \right) \right) \mathbb{E}\left[ \left\| \nabla f_i\left( w_i^{t_i}; z_{ij}^{t_i} \right) \right\| \right]$$

$$+ \beta\eta_t \mathbb{E}\left[ \sum_{t'=t_i}^{t-1} \left( \frac{\eta_{t'}}{b_1 N} \sum_{i'\in[N]} \sum_{j'\in[n]} \alpha_{i'j'}^{t'} \left( \frac{\mu\beta}{b_2} \sum_{l=1}^{b_2} \left\| v_{i',l}^{t'_{i'}} \right\|^3 + 2 \left\| \nabla f_{i'}\left( w_{i'}^{t'_{i'}}; z_{i'j'}^{t'_{i'}} \right) \right\| \right. \right. \right.$$

$$\left. \left. \left. + \left\| \frac{1}{b_2} \sum_{l=1}^{b_2} \left\langle \nabla f_{i'}\left( w_{i'}^{t'_{i'}}; z_{i'j'}^{t'_{i'}} \right) - \nabla f_{i'}\left( \bar{w}_{i'}^{t'_{i'}}; \bar{z}_{i'j'}^{t'_{i'}} \right), v_{i',l}^{t'_{i'}} \right\rangle v_{i',l}^{t'_{i'}} - \nabla f_{i'}\left( w_{i'}^{t'_{i'}}; z_{i'j'}^{t'_{i'}} \right) + \nabla f_{i'}\left( \bar{w}_{i'}^{t'_{i'}}; \bar{z}_{i'j'}^{t'_{i'}} \right) \right\| \right) \right) \right]$$

$$\leq (1 + 2\beta\eta_t) \mathbb{E}\left[ \left\| w^t - \bar{w}^t \right\| \right] + \left( \mu\beta^2 \left( \frac{1}{b_1 N} \left( 1 + \frac{b_1 - 1}{n} \right) + t_0 \right) \eta_t \eta_{t-t_0} + \mu\beta\eta_t \right) \mathbb{E}\left[ \left\| v_{i,l}^{t_i} \right\|^3 \right]$$

$$+ \left( \beta \left( 2 + 2\sqrt{\frac{d}{b_2}} \right) \left( \left( \frac{1}{b_1 N} \left( 1 + \frac{b_1 - 1}{n} \right) + t_0 \right) \eta_t \eta_{t-t_0} + \frac{1}{nN} \eta_t \right) \right) \mathbb{E}\left[ \left\| \nabla f_i\left( w_i^{t_i}; z_{ij}^{t_i} \right) \right\| \right]$$

$$+ \beta t_0 \eta_t \eta_{t-t_0} \left( \mu\beta \mathbb{E}\left[ \left\| v_{i',l}^{t'_{i'}} \right\|^3 \right] + \left( 2 + 2\sqrt{\frac{d}{b_2}} \right) \mathbb{E}\left[ \left\| \nabla f_{i'}\left( w_{i'}^{t'_{i'}}; z_{i'j'}^{t'_{i'}} \right) \right\| \right] \right)$$

$$= (1 + 2\beta\eta_t) \mathbb{E}\left[ \left\| w^t - \bar{w}^t \right\| \right] + \left( \mu\beta^2 \left( \frac{1}{b_1 N} \left( 1 + \frac{b_1 - 1}{n} \right) + 2t_0 \right) \eta_t \eta_{t-t_0} + \mu\beta\eta_t \right) \mathbb{E}\left[ \left\| v_{i,l}^{t_i} \right\|^3 \right]$$

$$+ \left( \beta \left( 2 + 2\sqrt{\frac{d}{b_2}} \right) \left( \left( \frac{1}{b_1 N} \left( 1 + \frac{b_1 - 1}{n} \right) + 2t_0 \right) \eta_t \eta_{t-t_0} + \frac{1}{nN} \eta_t \right) \right) \mathbb{E}\left[ \left\| \nabla f_i\left( w_i^{t_i}; z_{ij}^{t_i} \right) \right\| \right]$$

$$= (1 + 2\beta\eta_t) \mathbb{E}\left[ \left\| w^t - \bar{w}^t \right\| \right] + \left( \mu\beta^2 \left( \frac{1}{b_1 N} \left( 1 + \frac{b_1 - 1}{n} \right) + 2t_0 \right) \eta_t \eta_{t-t_0} + \mu\beta\eta_t \right) \frac{d}{d+3}$$

$$+ \left( \beta \left( 2 + 2\sqrt{\frac{d}{b_2}} \right) \left( \left( \frac{1}{b_1 N} \left( 1 + \frac{b_1 - 1}{n} \right) + 2t_0 \right) \eta_t \eta_{t-t_0} + \frac{1}{nN} \eta_t \right) \right) \mathbb{E}\left[ \left\| \nabla f_i\left( w_i^{t_i}; z_{ij}^{t_i} \right) \right\| \right]$$

$$\leq (1 + 2\beta\eta_t) \mathbb{E}\left[ \left\| w^t - \bar{w}^t \right\| \right] + \mu\beta^2 \left( \frac{1}{b_1 N} \left( 1 + \frac{b_1 - 1}{n} \right) + 2t_0 \right) \eta_t \eta_{t-t_0} + \mu\beta\eta_t$$

$$+ \left( \beta \left( 2 + 2\sqrt{\frac{d}{b_2}} \right) \left( \left( \frac{1}{b_1 N} \left( 1 + \frac{b_1 - 1}{n} \right) + 2t_0 \right) \eta_t \eta_{t-t_0} + \frac{1}{nN} \eta_t \right) \right) \mathbb{E} \left[ \left\| \nabla f_i \left( w_i^{t_i}; z_{ij}^{t_i} \right) \right\| \right].$$

(12)

To measure the stability, we need to obtain the upper bound of $\mathbb{E}[\|\nabla f_i(w_i^{t_i}; z_{ij}^{t_i})\|], i \in [N], j \in [n]$. Based on Equation (1), the update of asynchronous FedZO, triangular inequality, Lemmas 4, 5 and Assumption 3, we provide that

$$\mathbb{E}\left[ F_S\left( w^{t+1} \right) - F_S\left( w^t \right) \right]$$

$$\leq \mathbb{E}\left[ \left\langle w^{t+1} - w^t, \nabla F_S\left( w^t \right) \right\rangle + \frac{1}{2}\beta \left\| w^{t+1} - w^t \right\|^2 \right]$$

$$= \mathbb{E}\left[ \left\langle -\frac{\eta_t}{b_1 N} \sum_{i \in [N]} \sum_{m \in [b_1]} \tilde{\nabla} f_i\left( w_i^{t_i}; z_{i,m}^{t_i} \right), \nabla F_S(w^t) \right\rangle + \frac{1}{2}\beta \left\| \frac{\eta_t}{b_1 N} \sum_{i \in [N]} \sum_{m \in [b_1]} \tilde{\nabla} f_i\left( w_i^{t_i}; z_{i,m}^{t_i} \right) \right\|^2 \right]$$

$$= -\frac{\eta_t}{b_1 N} \sum_{i \in [N]} \sum_{m \in [b_1]} \mathbb{E}\left[ \left\langle \tilde{\nabla} f_i\left( w_i^{t_i}; z_{i,m}^{t_i} \right) - \nabla f_i\left( w_i^{t_i}; z_{i,m}^{t_i} \right) + \nabla f_i\left( w_i^{t_i}; z_{i,m}^{t_i} \right), \nabla F_S(w^t) \right\rangle \right]$$

$$+ \frac{\beta}{2} \mathbb{E}\left[ \left\| \frac{\eta_t}{b_1 N} \sum_{i \in [N]} \sum_{m \in [b_1]} \left( \tilde{\nabla} f_i\left( w_i^{t_i}; z_{i,m}^{t_i} \right) - \nabla f_i\left( w_i^{t_i}; z_{i,m}^{t_i} \right) + \nabla f_i\left( w_i^{t_i}; z_{i,m}^{t_i} \right) \right) \right\|^2 \right]$$

$$\leq -\frac{\eta_t}{b_1 N} \sum_{i \in [N]} \sum_{m \in [b_1]} \mathbb{E}\left[ \left\langle \tilde{\nabla} f_i\left( w_i^{t_i}; z_{i,m}^{t_i} \right) - \nabla f_i\left( w_i^{t_i}; z_{i,m}^{t_i} \right), \nabla F_S(w^t) \right\rangle \right] - \eta_t \mathbb{E}\left[ \left\langle \nabla f_i\left( w_i^{t_i}; z_{i,m}^{t_i} \right), \nabla F_S(w^t) \right\rangle \right]$$

$$+ \beta \eta_t^2 \mathbb{E}\left[ \left\| \tilde{\nabla} f_i\left( w_i^{t_i}; z_{i,m}^{t_i} \right) - \nabla f_i\left( w_i^{t_i}; z_{i,m}^{t_i} \right) \right\|^2 \right] + \beta \eta_t^2 \mathbb{E}\left[ \|\nabla F_S(w_i^{t_i})\|^2 \right]$$

$$\leq \frac{\eta_t}{2b_1 N} \sum_{i \in [N]} \sum_{m \in [b_1]} \mathbb{E}\left[ \left\| \tilde{\nabla} f_i\left( w_i^{t_i}; z_{i,m}^{t_i} \right) - \nabla f_i\left( w_i^{t_i}; z_{i,m}^{t_i} \right) \right\|^2 \right] + \frac{\eta_t}{2} \mathbb{E}\left[ \|\nabla F_S(w^t)\|^2 \right]$$

$$- \eta_t \mathbb{E}\left[ \left\langle \nabla f_i\left( w_i^{t_i}; z_{i,m}^{t_i} \right), \nabla F_S(w^t) \right\rangle \right] + \beta \eta_t^2 \mathbb{E}\left[ \left\| \tilde{\nabla} f_i\left( w_i^t; z_{i,m}^t \right) - \nabla f_i\left( w_i^t; z_{i,m}^t \right) \right\|^2 \right] + \beta \eta_t^2 \mathbb{E}\left[ \|\nabla F_S(w_i^{t_i})\|^2 \right]$$

$$\leq \left( \beta \eta_t^2 - \frac{\eta_t}{2} \right) \mathbb{E}\left[ \|\nabla F_S(w_i^{t_i})\|^2 \right] + \left( \frac{\eta_t}{2b_1 N} + \frac{\beta \eta_t^2}{b_1 N} \right) \sum_{i \in [N]} \sum_{m \in [b_1]} \left( \frac{\mu^2 \beta^2}{4} \mathbb{E}\left[ \left\| v_{i,l}^{t_i} \right\|^6 \right] \right.$$

$$\left. + \frac{d}{b_2} \mathbb{E}\left[ \left\| \nabla f_i\left( w_i^{t_i}; z_{i,m}^{t_i} \right) \right\|^2 \right] \right)$$

$$= \left( \left( 1 + \frac{d}{b_2} \right) \beta \eta_t^2 - \left( \frac{1}{2} - \frac{d}{2b_2} \right) \eta_t \right) \mathbb{E}\left[ \|\nabla F_S(w_i^{t_i})\|^2 \right] + \frac{d\mu^2 \beta^3 \eta_t^2}{4(d+6)} + \frac{d\mu^2 \beta^2 \eta_t}{8(d+6)}$$

$$\leq -\left( \frac{1}{4} - \frac{d}{4b_2} \right) \eta_t \mathbb{E}\left[ \|\nabla F_S(w_i^{t_i})\|^2 \right] + \frac{\mu^2 \beta^3 \eta_t^2}{4} + \frac{\mu^2 \beta^2 \eta_t}{8}$$

$$= -\left( \frac{1}{4} - \frac{d}{4b_2} \right) \mathbb{E}\left[ \|\nabla F_S(w^t)\|^2 \right] + \frac{\mu^2 \beta^3 \eta_t^2}{4} + \frac{\mu^2 \beta^2 \eta_t}{8}$$

$$\leq -\left( \frac{1}{2} - \frac{d}{2b_2} \right) \alpha \mathbb{E}\left[ F_S(w^t) - F_S(w(S)) \right] + \frac{\mu^2 \beta^3 \eta_t^2}{4} + \frac{\mu^2 \beta^2 \eta_t}{8}. \tag{13}$$

Then,

$$\mathbb{E}\left[ F_S\left( w^{t+1} \right) - F_S\left( w(S) \right) \right]$$

$$\leq \left( 1 - \left( \frac{1}{2} - \frac{d}{2b_2} \right) \alpha \right) \mathbb{E}\left[ F_S(w^t) - F_S(w(S)) \right] + \frac{\mu^2 \beta^3 \eta_t^2}{4} + \frac{\mu^2 \beta^2 \eta_t}{8}$$

$$\leq \mathbb{E}\left[ F_S(w^t) - F_S(w(S)) \right] + \frac{\mu^2 \beta^3 \eta_t^2}{4} + \frac{\mu^2 \beta^2 \eta_t}{8}$$

$$\leq \mathbb{E}\left[ F_S(w^1) - F_S(w(S)) \right] + \sum_{i=1}^{t} \left( \frac{\mu^2 \beta^3}{4} \eta_i^2 + \frac{\mu^2 \beta^2}{8} \eta_i \right)$$

$$\leq \mathbb{E}\left[F_S(w^1) - F_S(w(S))\right] + \frac{\mu^2\beta^3\eta_1^2}{2} + \frac{\mu^2\beta^2\eta_1}{8}\log(et),$$

where the last inequality is due to Lemma 3 (a), (b). It follows from Lemma 1 (2) that

$$\mathbb{E}[\|\nabla f_i(w_i^{t_i}; z_{ij}^{t_i})\|^2]$$

$$\leq 2\beta\mathbb{E}[F_{S_i}(w_i^{t_i})] = \frac{2\beta}{N}\sum_{i=1}^{N}\mathbb{E}[F_{S_i}(w_i^{t_i})] = 2\beta\mathbb{E}[F_S(w^t)]$$

$$\leq 2\beta\left(\mathbb{E}[F_S(w^1)] + \frac{\mu^2\beta^3\eta_1^2}{2} + \frac{\mu^2\beta^2\eta_1}{8}\log(e(t-1))\right).$$

For convenience, we denote $2\beta\left(\mathbb{E}[F_S(w^1)] + \frac{\mu^2\beta^3\eta_1^2}{2} + \frac{\mu^2\beta^2\eta_1}{8}\log(e(t-1))\right)$ as $\hat{\tau}(t)$. Then, from Equation (12), we know that

$$\mathbb{E}\left[\|w^{t+1} - \bar{w}^{t+1}\|\right]$$

$$\leq (1+2\beta\eta_t)\mathbb{E}\left[\|w^t - \bar{w}^t\|\right] + \left(\mu\beta + \frac{4\sqrt{\hat{\tau}(t)}}{nN}\sqrt{\frac{d}{b_2}}\right)\eta_t + \left(\mu\beta^2\left(\frac{1}{b_1 N}\left(1 + \frac{b_1-1}{n}\right) + 2t_0\right)\right)$$

$$+ \beta\left(2 + 2\sqrt{\frac{d}{b_2}}\right)\left(\frac{1}{b_1 N}\left(1 + \frac{b_1-1}{n}\right) + 2t_0\right)\sqrt{\hat{\tau}(t)}\right)\eta_{t-t_0}^2.$$

Let $a_5(t) = \mu\beta + \frac{4\sqrt{\hat{\tau}(t)}}{nN}\sqrt{\frac{d}{b_2}}$ and $a_6(t) = \mu\beta^2\left(\frac{1}{b_1 N}\left(1 + \frac{b_1-1}{n}\right) + 2t_0\right) + \beta\left(2 + 2\sqrt{\frac{d}{b_2}}\right)$ $\left(\frac{1}{b_1 N}\left(1 + \frac{b_1-1}{n}\right) + 2t_0\right)\sqrt{\hat{\tau}(t)}$. Taking summation from $t=1$ to $T-1$, we deduce that

$$\mathbb{E}[\|w^T - \bar{w}^T\|]$$

$$\leq \sum_{t=1}^{T-1}\left(\prod_{s=t+1}^{T-1}(1+2\beta\eta_s)\right)\left(a_5(t)\eta_t + a_6(t)\eta_{t-t_0}^2\right)$$

$$\leq \exp\left(2\beta\eta_1\sum_{s=1}^{T-1}s^{-1}\right)\left(a_5(T-1)\eta_1\sum_{t=1}^{T-1}t^{-1} + a_6(T-1)\eta_1^2\sum_{t=1}^{T-1}(t-t_0)^{-2}\right)$$

$$\leq (e(T-1))^{2\beta\eta_1}\left(a_5(T-1)\eta_1\log(e(T-1)) + 4a_6(T-1)\eta_1^2\right)$$

$$\leq \mathcal{O}\left(\mu T^{\frac{1}{2}}\left((nN)^{-1}(\log T)^{\frac{3}{2}} + \log T + t_0\sqrt{\log T}\right)\right).$$

$\square$

**Proof of Corollary 3**: We integrate Theorem 1 (b) and Theorem 5 to obtain that

$$|\mathbb{E}\left[F(w^T) - F_S(w^T)\right]|$$

$$\leq \frac{(4\theta)^\theta K}{nN}\sum_{i=1}^{N}\sum_{j=1}^{n}\mathbb{E}\left[\|w^T - \bar{w}^T\|\right] + 2\mathbb{E}\left[F_S(w^T)\right]$$

$$= (4\theta)^\theta K\mathbb{E}\left[\|w^T - \bar{w}^T\|\right] + 2\mathbb{E}\left[F_S(w^T)\right]$$

$$\leq \mathcal{O}\left(\left(\left(1 + (nN)^{-1}\sqrt{\log T}\right)\log T + \sqrt{\log T}t_0\right)(4\theta)^\theta\mu T^{\frac{1}{2}} + \mathbb{E}[F_S(w^T)]\right).$$

The proof is complete. $\square$

**Proof of Theorem 6**: According to Equation (13),

$$\mathbb{E}[F_S(w^{t+1}) - F_S(w^t)]$$

$$\leq -\left(\frac{1}{2} - \frac{d}{2b_2}\right)\alpha\eta_t\mathbb{E}\left[F_S(w^t) - F_S(w(S))\right] + \frac{\mu^2\beta^3\eta_t^2}{4} + \frac{\mu^2\beta^2\eta_t}{8},$$

that is

$$\mathbb{E}[F_S(w^{t+1}) - F(w(S))]$$

$$\leq \left(1 - \left(\frac{1}{2} - \frac{d}{2b_2}\right)\alpha\eta_t\right)\mathbb{E}\left[F_S(w^t) - F(w(S))\right] + \frac{\mu^2\beta^3\eta_t^2}{4} + \frac{\mu^2\beta^2\eta_t}{8}$$

$$= \left(1 - \left(\frac{1}{2} - \frac{d}{2b_2}\right)\frac{\alpha\eta_1}{t}\right)\mathbb{E}\left[F_S(w^t) - F(w(S))\right] + \frac{\mu^2\beta^3\eta_1^2}{4}t^{-2} + \frac{\mu^2\beta^2\eta_1}{8}t^{-1}.$$

We multiply both sides of the above inequality by $t\left(t - \left(\frac{1}{4} - \frac{d}{4b_2}\right)\alpha\eta_1\right)$ to get

$$t\left(t - \left(\frac{1}{4} - \frac{d}{4b_2}\right)\alpha\eta_1\right)\mathbb{E}[F_S(w^{t+1}) - F(w(S))]$$

$$\leq \left(t - \left(\frac{1}{4} - \frac{d}{4b_2}\right)\alpha\eta_1\right)\left(t - \left(\frac{1}{2} - \frac{d}{2b_2}\right)\alpha\eta_1\right)\mathbb{E}\left[F_S(w^t) - F(w(S))\right]$$

$$+ \frac{\mu^2\beta^3\eta_1^2}{4} + \frac{\mu^2\beta^2\eta_1}{8}t$$

$$\leq \left(1 - \left(\frac{1}{4} - \frac{d}{4b_2}\right)\alpha\eta_1\right)\left(1 - \left(\frac{1}{2} - \frac{d}{2b_2}\right)\alpha\eta_1\right)\mathbb{E}\left[F_S(w^1) - F(w(S))\right]$$

$$+ \frac{\mu^2\beta^3\eta_1^2 t}{4} + \frac{\mu^2\beta^2\eta_1}{8}\sum_{t'=1}^{t}t'$$

$$\leq \left(1 - \left(\frac{1}{4} - \frac{d}{4b_2}\right)\alpha\eta_1\right)\left(1 - \left(\frac{1}{2} - \frac{d}{2b_2}\right)\alpha\eta_1\right)\mathbb{E}\left[F_S(w^1) - F(w(S))\right]$$

$$+ \frac{\mu^2\beta^3\eta_1^2 t}{4} + \frac{\mu^2\beta^2\eta_1}{16}t(t+1).$$

Therefore,

$$\mathbb{E}[F_S(w^T) - F_S(w(S))]$$

$$\leq \frac{\left(1 - \left(\frac{1}{4} - \frac{d}{4b_2}\right)\alpha\eta_1\right)\left(1 - \left(\frac{1}{2} - \frac{d}{2b_2}\right)\alpha\eta_1\right)}{(T-1)\left(T - 1 - \left(\frac{1}{4} - \frac{d}{4b_2}\right)\alpha\eta_1\right)}\mathbb{E}\left[F_S(w^1) - F(w(S))\right]$$

$$+ \frac{\mu^2\beta^3\eta_1^2}{4\left(T - 1 - \left(\frac{1}{4} - \frac{d}{4b_2}\right)\alpha\eta_1\right)} + \frac{\mu^2\beta^2\eta_1 T}{16\left(T - 1 - \left(\frac{1}{4} - \frac{d}{4b_2}\right)\alpha\eta_1\right)}$$

$$= \mathcal{O}\left(T^{-2} + \mu^2\right).$$

The optimization bound is given. By integrating this optimization upper bound and the generalization upper bound in Corollary 3, we can get the following expected excess risk bound

$$\left|\mathbb{E}\left[F(w^T) - F(w^*)\right]\right|$$

$$\leq \left|\mathbb{E}\left[F(w^T) - F_S(w^T)\right]\right| + \left|\mathbb{E}\left[FS(w^T) - F_S(w(S))\right]\right|$$

$$\leq \mathcal{O}\left(T^{-2} + \mu^2 + \left(\left(1 + (nN)^{-1}\sqrt{\log T}\right)\log T + \sqrt{\log T}t_0\right)(4\theta)^\theta\mu T^{\frac{1}{2}} + \mathbb{E}[F_S(w^T)]\right).$$

$\square$