# OpenReview forum: "Fine-Grained Theoretical Analysis of Federated Zeroth-Order Optimization"
_NeurIPS.cc/2023/Conference — NeurIPS 2023 poster_

### Official Review · Reviewer_obsy · 2023-07-07

**Soundness:** 3 good
**Presentation:** 3 good
**Contribution:** 3 good
**Rating:** 6
**Confidence:** 3

**Summary:**

This paper provides generalization analysis of federated zeroth order optimization.

**Strengths:**

The paper is well-written and addresses a relevant problem. The theoretical results appear correct, although I haven't thoroughly checked them.

**Weaknesses:**

Since I'm not super familiar with this direction, I just have a few clarification questions I have asked below. The one limitation in my opinion is the lack of multiple-local updates at the clients.

**Questions:**

Theory:
- Line 105: what is the intuitive reason for the requirement $b_2 \geq d$?
- In (3), aren't we considering multiple local steps at the clients?
- In (5), shouldn't there be an additional difference $\mathbb E[F_S(w(S))] - F(w^*)$?
- In the paragraph following Theorem 1 in lines 187-88, it is said that $\mathbb E[F_S(A(S))]$ has no adverse impact on the upper bound. Why so? If this term is not close to zero, doesn't it introduce a bias, irrespective of $\epsilon$?
- Line 206: one can always find constant $c$ such that $\beta c \geq 1$ holds. Shouldn't there be some more conditions for this statement to hold?
- Line 228 "while introducing the dependence $(nN)^{-1}$": but theorem 2 also has $(nN)^{-1}$
- Line 249: what is $\alpha$ in this bound?
- Are all the results stated assuming full-client participation, since only $N$ appears in the bounds, even though (3) is with partial client participation ($M$).

---

> ### Author Rebuttal · Authors · 2023-08-08
>
> **Q1:** Line 105: what is the intuitive reason for the requirement $b_2\geq d$?
>
> **A1:** Thanks for your constructive comments. The requirement $b_2\geq d$ is adopted by many previous work. For example, as mentioned in the second paragraph of Introduction in [1], “deterministic zeroth-order approaches require at least $b_2\geq d+1$ queries”. In Section 3.3 of [2], the objective function is also evaluated $2d$ times to estimate gradients of all $d$ coordinates ($b_2=2d$). We have modified our explanation in line 105 of the main paper.
>
> [1] K. Nikolakakis, et al. Black-box generalization: Stability of zeroth-order learning. NeurIPS, 2022.
>
> [2] P. Chen, et al.  Zoo: Zeroth order optimization based black-box attacks to deep neural networks without training substitute models. AISec, 2017.
> ***
> **Q2:** In (3), aren't we considering multiple local steps at the clients?
>
> **A2:** Due to some obstacles of proof technology, we don’t expand our analysis to the multiple-local case (H>1). If these obstacles are overcome in the future, we will further analyze the case you mentioned.
> ***
> **Q3:** In (5), shouldn't there be an additional difference $\mathbb E[F_S(w(S))] - F(w^*)$?
>
> **A3:** The complete form of Equation (5) should be $\mathbb{E}[F(A(S))-F(w^*)]\leq \mathbb{E}[F(A(S))-F_S(A(S))]+\mathbb{E}[F_S(A(S))-F_S(w(S))]$ since $\mathbb{E}[F_S(w(S))-F(w^*)]=F(w(S))-F(w^*)\leq 0$. To avoid ambiguity, we have corrected it with the complete inequality.
> ***
> **Q4:** ... Why $\mathbb E[F_S(A(S))]$ has no adverse impact on the upper bound? If this term is not close to zero, doesn't it introduce a bias, irrespective of $\epsilon$?
>
> **A4:** Thanks. In this paper, we assume the output global model has a small empirical risk [3][4] on the training set to study the theoretical generalization performance on the unknown testing set. If this term $\mathbb E[F_S(A(S))]$ is not small enough, we don’t wish that the performance on the training set is generalized to the unknown testing set. Thus, we let the term $\mathbb E[F_S(A(S))]=\mathcal{O}(1/(nN))$. An explanation has been added in the paragraph following Theorem 1.
>
> [3] Y. Lei et al. Fine-grained analysis of stability and generalization for stochastic gradient descent. ICML, 2020.
>
> [4] S. Li, Y. Liu. High probability guarantees for nonconvex stochastic gradient descent with heavy tails. ICML, 2022.
> ***
> **Q5:** Line 206: one can always find constant $c$ such that $\beta c\geq 1$ holds. Shouldn't there be some more conditions for this statement to hold?
>
> **A5:** These results [5][6][7] in Table 1 are not concise due to some parameters, such as $\beta$ and constant c. Our results can’t be directly compared with them. Thus, we select some specific cases, (e.g., $\beta c \geq 1$ ) to show the advantage of our results, which is a standard strategy for comparisons of error bounds (see e.g., Table 1 in [8], Table 1 in [9]). We can’t ensure that one can always find constant $c$ such that $\beta c \geq 1$ holds. Following your valuable comments, we have added a remark for the parameter conditions in line 206 of the main paper.
>
> [5] M. Hardt, et al. Train faster, generalize better: Stability of stochastic gradient descent. ICML, 2016.
>
> [6] W. Shen, et al. Stability and optimization error of stochastic gradient descent for pairwise learning. arXiv, 2019.
>
> [7] K. Nikolakakis, et al. Black-box generalization: Stability of zeroth-order learning. NeurIPS, 2022.
>
> [8]W. Fang, et al. Communication-efficient stochastic zeroth-order optimization for federated learning. TSP, 2022.
>
> [9] K. Nikolakakis, et al. Black-box generalization: Stability of zeroth-order learning. NeurIPS, 2022.
> ***
> **Q6:** Line 228 "while introducing the dependence $(nN)^{-1}$": but theorem 2 also has $(nN)^{-1}$
>
> **A6:** When we don’t take some specific value of $\mu$, the order of Theorem 3 is $\mathcal{O}(((nN)^{-1}\sqrt{\log T}+1)\mu T^{\frac{1}{2}}\log T)$. Compared with Theorem 2, the bound in Theorem 3 is independent of Lipschitz parameter. Meanwhile, the dependence on $\mu$ is improved from the partial dependence $(L/(nN) + \mu)$ to full dependence $((\sqrt{\log T}/(nN) + 1) \mu)$. We have corrected this statement in line 28 of our main paper.
> ***
> **Q7:** Line 249: what is $\alpha$ in this bound?
>
> **A7:** $\alpha$ is the parameter of PL condition (Assumption 3 in lines 171 and 172 of the main paper).
> ***
> **Q8:** Are all the results stated assuming full-client participation, since only $N$ appears in the bounds, even though (3) is with partial client participation ($M$).
>
> **A8:** Firstly, synchronous FedZO is with partial client participation. During the whole training process, there is a probability $\frac{M}{N}$ for each client at each iteration to be selected to update the global model, which is the reason why $N$ appears in our bounds. While, for a single iteration, only M clients are used to update the global model. In fact, our bounds would have the dependence $\frac{1}{M}$ if the term $\frac{M}{N}$ is relaxed to 1 due to $M\in[1, N]$. The details can be found in the proofs of Theorems 2, 3 (see lines 41 and 56). Secondly, asynchronous FedZO is with full-client participation, which is presented in Equation (7) of the main paper.

---

> > ### Comment · Reviewer_obsy · 2023-08-18
> > **Thanks for the response**
> >
> > As I said earlier, I'm not super familiar with this space, no I just had clarification questions, all of which the authors have answered satisfactorily. I apologize for not increasing the score (owing to my own limited knowledge of the field, I cannot advocate for the paper too strongly), but support the paper.

---

> > > ### Author Response · Authors · 2023-08-18
> > >
> > > Thank you very much for your support of our work.

---

### Official Review · Reviewer_YCFd · 2023-07-07

**Soundness:** 3 good
**Presentation:** 3 good
**Contribution:** 3 good
**Rating:** 5
**Confidence:** 2

**Summary:**

The analysis of Federated Zeroth-Order Optimization is limited now. This work considers the zeroth-order optimization in federated learning and establishes the generalization error bound of FedZO under the Lipschitz continuity and smoothness conditions.

**Strengths:**

1. The analysis of Federated Zeroth-Order Optimization is limited now. This work provides generalization bounds with theoretical analysis and asynchronous FedZO.


**Weaknesses:**

There is no experimental analysis.

1. In deep learning, the first-order stochastic optimizer is very popular. Although some cases are mentioned where gradient information is expensive to obtain, it is necessary to use experiments to verify the necessity of the zero-order algorithm.

2. Experiment about the efficiency of algorithms are also missing.

**Questions:**

1. Could you provide more motivation to study zeroth-order optimization in federated learning? In practice, we have many stochastic optimizers in federated learning. Does zeroth-order optimization have any advantage compared with these optimizers?

2. What is the key challenge of zeroth-order optimization in federated learning compared with the optimization in the single-machine setting?

**Limitations:**

No experiments are provided to verify their theoretical analysis

---

> ### Author Rebuttal · Authors · 2023-08-08
>
> **Q1:** ... it is necessary to use experiments to verify the necessity of the zero-order algorithm.
>
> **A1:** Thanks for your constructive comments. As your mentioned, there are some cases where gradient information is expensive to obtain and even unavailable [1], such as federated hyperparameter tuning [2] or distributed black box attack of deep neural networks (DNN) [3]. First-order optimizer is not suitable for these cases, which had been validated in many previous work [4][5][6].
>
> [1]W. Fang, et al. Communication-efficient stochastic zeroth-order optimization for federated learning. TSP, 2022.
>
> [2]Z. Dai, et al. Federated bayesian optimization via thompson sampling. NeurIPS, 2020.
>
> [3]X. Yi, et al. Zerothorder algorithms for stochastic distributed nonconvex optimization. arXiv, 2021.
>
> [4]J. Nocedal and S. Wright, Numerical optimization, Springer Science & Business Media, 2006.
>
> [5]A. Conn, et al. Introduction to derivative-free optimization, SIAM, 2009.
>
> [6]L. Rios and N. Sahinidis. Derivative-free optimization: a review of algorithms and comparison of software implementations. Journal of Global Optimization, 2013.
> ***
> **Q2:** Experiment about the efficiency of algorithms are also missing.
>
> **A2:** Thanks. Some relevant experiments on FedZO had been provided in [7]. In addition, for the asynchronous version, there are many related works [8][9] validating that the asynchronous strategy can take full advantage of the clients computation capabilities. It should be noted that, as far as we know, there is a gap in the theoretical generalization guarantee of federated zeroth-order optimization algorithm, especially for the stability-based analysis, and our major contribution is providing the related stability-based theoretical generalization analysis. Our theoretical results conform to these empirical behaviors [7][8][9] from both generalization and optimization perspectives which are listed as follows.
>
> **Generalization:** Our generalization bounds (Theorems 2, 3) are negatively correlated with the number M of selected clients at each iteration (the dependence $\frac{1}{M}$ can be seen in lines 41 and 56 in Appendix). It is consistent with Figures 1(b) and 4 of [7].
>
> **Optimization:** Our optimization bounds (Theorems 4 and 6) are negatively correlated with the total iteration number $T$, which is presented in all figures of [7].
>
> To further explicitly show the efficiency of the asynchronous version, some necessary experiments (similar to the ones of synchronous FedZO [7]) will be conducted later. We have utilized “global response” to provide Figure 1 to show the structure of asynchronous FedZO.
>
> [7]W. Fang, et al. Communication-efficient stochastic zeroth-order optimization for federated learning. TSP, 2022.
>
> [8]X. Lian, et al. Asynchronous decentralized parallel stochastic gradient descent. ICML, 2018.
>
> [9]X. Lian, Y. Huang, Y. Li, and J. Liu. Asynchronous parallel stochastic gradient for nonconvex optimization. NIPS, 2015.
> ***
> **Q3:** Could you provide more motivation to study zeroth-order optimization in federated learning? ... Does zeroth-order optimization have any advantage compared with these optimizers?
>
> **A3:** This paper aims to fill the gap in the generalization guarantee of zeroth-order optimization in federated learning, which is valuable to the development of related algorithms. Compared with many other stochastic optimizers in federated learning, zeroth-order optimization algorithm can tackle some special cases with unknown gradient information [10] (e.g., federated hyperparameter tuning [11] or distributed blackbox attack of DNN [12]) and achieve satisfactory performance (e.g., FedZO can serve as a satisfactory alternative for the FedAvg algorithm [10]).
>
> [10]W. Fang, et al. Communication-efficient stochastic zeroth-order optimization for federated learning. TSP, 2022.
>
> [11]Z. Dai, et al. Federated bayesian optimization via thompson sampling. NeurIPS, 2020.
>
> [12]X. Yi, et al. Zerothorder algorithms for stochastic distributed nonconvex optimization. Automatica, 2022.
> ***
> **Q4:** What is the key challenge of zeroth-order optimization in federated learning compared with the optimization in the single-machine setting?
>
> **A4:** In federated learning, multiple clients use their own data to train a global model. Thus, the key challenge compared with single-machine learning is how to tackle the relationship among all clients, especially for the asynchronous version. We deal with it by introducing new formulas of the update of FedZO and asynchronous FedZO (see Equation (4) in Appendix B.3 and Equation (9) in Appendix B.5), and designing a new error decomposition strategy (see line 271). Besides, the unavailability of gradient information leads to another key challenge, i.e., estimating the real gradient of loss function. In this paper, we use the standard finite difference method to estimate the gradient and the second order Taylor expansion to make an approximation, respectively. Following your constructive comments, we have added a remark in line 118 of the main paper to demonstrate the above statements.

---

> > ### Comment · Reviewer_YCFd · 2023-08-17
> >
> > Thanks for your response. I improve my score. Thanks.

---

> > > ### Author Response · Authors · 2023-08-17
> > >
> > > Thank you very much for your recognition of our work.

---

### Official Review · Reviewer_oYUF · 2023-07-07

**Soundness:** 3 good
**Presentation:** 3 good
**Contribution:** 3 good
**Rating:** 6
**Confidence:** 3

**Summary:**

This paper presents a detailed analysis of Federated Zeroth-Order Optimization (FedZO) by developing the analysis technique of on-average model stability. The authors establish generalization error bounds for FedZO and refine them using heavy-tailed gradient noise and second-order Taylor expansion. They extend the analysis to the asynchronous case and contribute to systematic assessments, on-average model stability technique, and improved bounds for practical FedZO applications. In short, the main contributions of the paper are the establishment of systematic theoretical assessments of FedZO, the development of the analysis technique of on-average model stability, and the refinement of generalization and optimization bounds for practical applications of FedZO.


**Strengths:**

1. To my best knowledge, the authors provide the first generalization error bound of FedZO under the Lipschitz continuity and smoothness conditions, and also refine generalization and optimization bounds by replacing bounded gradient with heavy-tailed gradient noise and utilizing the second-order Taylor expansion for gradient approximation.
2. This paper has also provided the theoretical analysis for asynchronous FedZO, which has in fact seldomly considered in the federated zeroth-order optimization field.
3. The theoretical results are generally sound and can be inspiring for the follow-up works on federated zeroth order optimization.


**Weaknesses:**

1. While the authors provide a new error decomposition strategy for the asynchronous case, they do not compare their results with existing error bounds for other asynchronous optimization algorithms.
2. This paper does not provide any experimental results to validate the theoretical findings, which may limit its practical relevance. Empirical verification may make this paper more sound. So, I encourage the authors to provide certain empirical experiments to validate their main theoretical results.

**Questions:**

1. Can the authors provide more intuition behind the on-average model stability analysis technique? It may be helpful for readers who are not familiar with this concept to have a more intuitive understanding of how it relates to the generalization error.
2. The authors mention that their refined generalization and optimization bounds have important implications for practical applications of FedZO. Can they provide some examples of these practical applications and how their results could be used in these scenarios?
3. The paper focuses on the theoretical analysis of FedZO. Can the authors provide some insights into how their results could be used in practical applications of federated learning? For example, how could their results be used to improve the performance of real-world federated learning systems?

---

> ### Author Rebuttal · Authors · 2023-08-08
>
> **Q1:** ... compare with other asynchronous optimization algorithms.
>
> **A1:** Thanks. Following your constructive comments, we have added comparisons with existing error bounds of asynchronous optimization algorithms [1] and [1][2] in Tables 1 and 2, respectively. Limited by the length of Rebuttal, parts of these new comparisons are listed below and in Tables 1, 2 of “global response”.
>
> **Table 1**
>
> | | | | | | | |
> |:----:|:----:|:----:|:----:|:----:|:----:|:----:|
> |**Algorithm+Reference**|**Generalization bound**|**Tool**|**L**|**$\theta$**|**$v^2$**|**B.**|
> |**AD-SGD**[1]|$\mathcal{O}(\frac{n-\lambda}{n(1-\lambda)}(1+\frac{\beta \eta_1 }{M})^T)$|Uni.|$\surd$|$\times$|$\times$|$\times$|
> |**AD-SGD**[1]|$\mathcal{O}(\frac{nM-\lambda}{n(1-\lambda)}L^2T)$|Uni.|$\surd$|$\times$|$\times$|$\times$|
> | | | | | | | |
>
> where $\lambda$ characterizes the properties of decentralized topology.
>
> **Table 2**
>
> | | | | | | | | |
> |:----:|:----:|:----:|:----:|:----:|:----:|:----:|:----:|
> |**Algorithm+Reference**|**Optimization bound**|**Step size**|**L**|**$\theta$**|**$\beta$**|**B.**|**$\sigma$**|
> |**AD-SGD**[1]|$\mathcal{O}((r+\frac{C_{\lambda}}{\lambda^{t_0}}+\frac{t_0}{M})(\log T)^{-1})$|$\eta_t=\mathcal{O}(\frac{Mc}{t+1})$|$\surd$|$\times$|$\surd$|$\times$|$\times$|
> |**AD-PSGD**[2]|$\mathcal{O}(T^{-1/2})$|$\eta_t=\mathcal{O}(\frac{n}{b_1(\sqrt{T}+1)})$|$\times$|$\times$|$\surd$|$\times$|$\surd$|
> | | | | | | | | |
>
> where $C_{\lambda}=\frac{4}{\lambda e^2\log\lambda^{-1}}+\frac{2}{\lambda\log\lambda^{-1}}$.
>
> [1]X. Deng, et al. Stability-based generalization analysis of the asynchronous decentralized SGD. AAAI, 2023.
>
> [2]X. Lian, et al. Asynchronous decentralized parallel stochastic gradient descent. ICML, 2018.
> ***
> **Q2:** ... provide empirical experiments ...
>
> **A2:** There are some works providing some relevant experiments on FedZO [3]. For the asynchronous version, many related work [4][5] validates that the asynchronous strategy can take full advantage of the clients computation capabilities. It should be noted that our major contribution is exploring the theoretical generalization guarantee of federated zeroth-order optimization algorithm which is a gap as far as we know, especially for the stability-based analysis. Our theoretical results conform to these empirical behaviors [3] from both generalization and optimization perspectives which are listed as follows.
>
> **Generalization:** Our generalization bounds (Theorems 2, 3) are negatively correlated with the number M of selected clients at each iteration (the dependence $\frac{1}{M}$ can be seen in lines 41 and 56 in Appendix), which is consistent with Figures 1(b) and 4 of [3].
>
> **Optimization:** Our optimization bounds (Theorems 4 and 6) are negatively correlated with the total iteration number $T$, which is presented in all figures of [3].
>
> To further explicitly show the efficiency of the asynchronous version, some necessary experiments (similar to the ones of synchronous FedZO [3]) will be conducted later. We have utilized “global response” to provide Figure 1 to show the structure of asynchronous FedZO.
>
> [3]W. Fang, et al. Communication-efficient stochastic zeroth-order optimization for federated learning. TSP, 2022.
>
> [4]X. Lian, et al. Asynchronous decentralized parallel stochastic gradient descent. ICML, 2018.
>
> [5]X. Lian, Y. Huang, Y. Li, and J. Liu. Asynchronous parallel stochastic gradient for nonconvex optimization. NIPS, 2015.
> ***
> **Q3:** ... provide more intuition behind the on-average model stability ...
>
> **A3:** Compared with other tools including uniform convergence, algorithmic stability enjoys the dimension-independence of hypothesis parameter space [6][7]. We list the advantages of on-average model stability over several other stability tools as follows [8]:
>
> **Vs. Uniform (model) stability:** On-average model stability is a weaker stability tool than uniform (model) stability.
>
> **Vs. On-average stability:** On-average model stability measures the stability of model parameters $w$ instead of function values $f(w)$, which can improve our analysis.
>
> More comparisons among stability tools can be found in Appendix C of [9]. We have added the above statements in line 137 of the main paper.
>
> [6]W. Rogers et al. A finite sample distribution-free performance bound for local discrimination rules. The Annals of Statistics, 1978.
>
> [7]L. Devroye et al. Distribution-free performance bounds for potential function rules. TIT, 1979.
>
> [8]Y. Lei et al. Fine-grained analysis of stability and generalization for stochastic gradient descent. ICML, 2020.
>
> [9]J. Chen, et al. On the Stability and Generalization of Triplet Learning. AAAI, 2023.
> ***
> **Q4:** ... provide some practical examples and how their results used?
>
> **A4:** Our theoretical results conform to the related experimental results of FedZO in [10]. Please see **Q2** for some examples. In addition, our results are also consistent with some distributed zeroth-order optimization algorithms, e.g., [11].
>
> [10]W. Fang, et al. Communication-efficient stochastic zeroth-order optimization for federated learning. TSP, 2022.
>
> [11]E. Kaya, et al. Communication-efficient zeroth-order distributed online optimization: Algorithm, theory, and applications. Access, 2023.
> ***
> **Q5:** ... how their results could be used in practice ...
>
> **A5:** Firstly, from the perspective of generalization theory, our results are consistent with the experimental results of previous work [12]. Secondly, from the perspective of practical application, our results can provide guidance for parameter choices (e.g. the number M of selected clients at each iteration, the total iteration number T and the step size $\mu$ of gradient estimate) under some special accuracy requirement.
>
> [12]W. Fang, et al. Communication-efficient stochastic zeroth-order optimization for federated learning. TSP, 2022.

---

### Official Review · Reviewer_E5dV · 2023-07-11

**Soundness:** 3 good
**Presentation:** 3 good
**Contribution:** 3 good
**Rating:** 8
**Confidence:** 3

**Summary:**

This paper studies the theoretical analysis for federated zeroth-order optimization (FedZO). The main contributions of this paper include 1) deriving the generalization bound of synchronous FedZO with different assumptions (bound gradient, heavy tail gradient noise). The main technical is to establish the relationship between generalization error and $\ell_1$ on-average model stability; 2) deriving the generalization and optimization bounds for asynchronous FedZO.

**Strengths:**

1. This paper derives the generalization bound of synchronous FedZO with different assumptions (bound gradient, heavy tail gradient noise). The main technical is to establish the relationship between generalization error and $\ell_1$ on-average model stability;

2. This paper derives the generalization and optimization bounds for asynchronous FedZO with novel technique by a new error decomposition strategy.

3. The paper is very well written. In particular, it explains the main technical challenges, main techniques and the implication of the results very well.

**Weaknesses:**

I didn't find major weaknesses in this paper. Perhaps, it will be beneficial if the authors could provide some experimental study to validate the theories.

**Questions:**

1. It seems that (5) does not equal to $\mathbb{E}[F(A(S)) - F(\omega^*)]$. Could you please clarify?

2. In the paragraph below (4), do you mean Algorithm 1 instead of Algorithm A?

3. In Assumption 2, does $z_i^t$ means a data sample?

4. In Theorems 2, 3 and so on, what is the physical meaning of $\mu$?

**Limitations:**

There are no much discussion on the limitation of this paper.

---

> ### Author Rebuttal · Authors · 2023-08-08
>
> **Q1:** Perhaps, it will be beneficial if the authors could provide some experimental study to validate the theories.
>
> **A1:** Thanks for your constructive comments. Considering the gap in the generalization analysis of federated zeroth-order optimization algorithm, the major contribution of this paper is providing its stability-based theoretical guarantee, which is meaningful to the development of this algorithm. At present, there are some works providing some relevant experiments on FedZO [1]. For the asynchronous version, many related work [2][3] validates that the asynchronous strategy can take full advantage of the clients computation capabilities. Our theoretical results conform to these empirical behaviors [1][2][3] from both generalization and optimization perspectives which are listed as follows.
>
> **Generalization:** Our generalization bounds (Theorems 2, 3) are negatively correlated with the number M of selected clients at each iteration (the dependence $\frac{1}{M}$ can be seen in lines 41 and 56 in Appendix). It is consistent with Figures 1(b) and 4 of [1].
>
> **Optimization:** Our optimization bounds (Theorems 4 and 6) are negatively correlated with the total iteration number $T$, which is presented in all figures of [1].
>
> To further explicitly show the efficiency of the asynchronous version, some necessary experiments (similar to the ones of synchronous FedZO [1]) will be conducted later. We have utilized “global response” to provide Figure 1 to show the structure of asynchronous FedZO.
>
> [1]W. Fang, et al. Communication-efficient stochastic zeroth-order optimization for federated learning. TSP, 2022.
>
> [2]X. Lian, et al. Asynchronous decentralized parallel stochastic gradient descent. ICML, 2018.
>
> [3]X. Lian, Y. Huang, Y. Li, and J. Liu. Asynchronous parallel stochastic gradient for nonconvex optimization. NIPS, 2015.
> ***
> **Q2:** It seems that (5) does not equal to $\mathbb{E}[F(A(S)) - F(\omega^*)]$.
>
> **A2:** The complete form of Equation (5) should be $\mathbb{E}[F(A(S)) - F(w^*)] \leq \mathbb{E}[F(A(S))-F_S(A(S))]+\mathbb{E}[F_S(A(S))-F_S(w(S))]$ since $\mathbb{E}[F_S(w(S))-F(w^*)]=F(w(S))-F(w^*)\leq 0$. To avoid ambiguity, we have corrected it with the complete inequality.
> ***
> **Q3:** ... do you mean Algorithm 1 instead of Algorithm A?
>
> **A3:** Algorithm A indicates the general federated learning algorithm including Algorithm 1 (synchronous FedZO). Theorem 1 is developed for Algorithm A, while the rest results (Theorems 2-6) are developed for Algorithm 1 and its asynchronous version (synchronous FedZO and asynchronous FedZO). We have made the above explanation in line 111 of the main paper.
> ***
> **Q4:** In Assumption 2, does $z_i^t$ means a data sample?
>
> **A4:** Your understanding is correct. $z_i^t$ means the sample of the i-th client used at the t-th iteration.
> ***
> **Q5:** ... what is the physical meaning of $\mu$?
>
> **A5:** The meaning of $\mu$ had been briefly explained before Equation (3). According to the definition of gradient, $\mu$ represents the distance between two parameters used to estimate gradient. We have modified our related explanation to improve the readability of our paper.

---

> > ### Comment · Reviewer_E5dV · 2023-08-18
> >
> > I would like to thank the authors for answering my questions. I am keeping my original score.

---

> > > ### Author Response · Authors · 2023-08-19
> > >
> > > Thank you very much for your recognition and support of our work. Thanks.

---

### Official Review · Reviewer_hpvH · 2023-07-19

**Soundness:** 3 good
**Presentation:** 3 good
**Contribution:** 3 good
**Rating:** 7
**Confidence:** 3

**Summary:**

This paper fills the gap of theoretical guarantee for the Federated zeroth-order optimization (FedZO) algorithm. It provides the initial generalization error bound for FedZO and presents refined generalization and optimization bounds. The structure of the paper is logical and the accompanying theoretical proofs in the supplementary material are comprehensive and robust.

**Strengths:**

1. The need for filling the theoretical gap is important.
2. The Author's contribution is solid.
3. This paper is well-organized and easy to follow.


**Weaknesses:**


This paper addresses a topic that isn't directly within my area of expertise, so my comments may not fully capture the nuances specific to this field, but I hope my remarks will prove useful to the authors.

**1.** In Assumption 3, the authors refer to the PL condition but they do not explain what PL stands for earlier in the text.

**2.** Regarding the assumptions made throughout the paper, it would be helpful if the authors could elaborate on whether each assumption is considered strong, mild, or weak, or if it is a relaxed version of a concept from previous work.

**3.** The term "heavy-tailed" is used but not clearly defined. It would benefit readers if the authors could briefly describe what they mean by "heavy-tailed" early in the paper. In the context of imbalanced regression problems, is "heavy-tailed" equivalent to "long-tailed"? In my understanding, both terms refer to highly biased or imbalanced data.

**4.** There are minor grammatical errors that need to be corrected.

In conclusion, this paper is well grounded and presents detailed, solid findings.

**Questions:**

See above

**Limitations:**

Authors did not provide the limitation section.

---

> ### Author Rebuttal · Authors · 2023-08-08
>
> **Q1:** ... not explain what PL stands for ...
>
> **A1:** Thanks for your constructive comments. As we all know, under the non-convex condition, local optimal model isn’t equivalent to global optimal model. Assumption 3 simply requires that the gradient grows faster than a quadratic function as we move away from the optimal function value [1], which stands for the fact that every stationary point ($|\nabla F_S(w)|=0$) is a global minimum [2]. With this assumption, we can study the optimization error with the form $F_S(w)-F_S(w(S))$ rather than $|\nabla F_S(w)|$ [3] under the non-convex condition. We have added a remark to clarify the meaning of PL condition following Assumption 3.
>
> [1]H. Karimi, et al. Linear convergence of gradient and proximalgradient methods under the polyak-łojasiewicz condition. ECML, 2016.
>
> [2]L. Lei, et al. Non-convex finite-sum optimization via scsg methods. NeurIPS, 2017.
>
> [3]S. Li, Y. Liu. High probability guarantees for nonconvex stochastic gradient descent with heavy tails. ICML, 2022.
> ***
> **Q2:** ... it would be helpful if the authors could elaborate on whether each assumption is considered strong, mild, or weak, or if it is a relaxed version of a concept from previous work.
>
> **A2:** The strength of assumptions is crucial for learning theoretical analysis. The related illustrations for all assumptions are listed as follows.
>
> (1)**Lipschitz continuity (bounded gradient):** It is one of the most general assumptions which is considered strong. Some milder related condition are provided in some previous work, e.g., Assumption 2.8 in [4]. We just assume the Lipschitz condition in our first case. In the rest cases, we introduce a milder assumption, i.e., bounded gradient noise (i.e., heavy-tailed gradient noise) to avoid the dependence on $L$.
>
> (2)**Smoothness:** It can be called bounded second-order gradient assumption which is also considered strong. Generally, Holder continuity is a milder condition than smoothness [5].
>
> (3)**Heavy-tailed gradient noise:** As mentioned in (1), heavy-tailed gradient noise can be regarded as a milder condition than Lipschitz continuity. It stands for a refined bounded variance of gradient which is similar to [6].
>
> (4)**PL condition:** It is commonly assumed in non-convex optimization [1][2][3]. [1] shows that PL condition is weaker than the main conditions that have been explored to show linear convergence rates without strong convexity, e.g., essential strong convexity[7][8].
> We have added the above four illustrations following the corresponding assumptions respectively. In the last part of Appendix, we have discussed whether we can relax our current assumptions, e.g., whether we can relax smoothness to Holder continuity.
>
> [4]S. Li, Y. Liu. High probability guarantees for nonconvex stochastic gradient descent with heavy tails. ICML, 2022.
>
> [5]Y. Lei, Y. Ying. Fine-grained analysis of stability and generalization for stochastic gradient descent. ICML, 2020.
>
> [6]Y. Zhou, et al. Understanding generalization error of SGD in nonconvex optimization. Machine Learning, 2022.
>
> [7]J. Liu, et al. An asynchronous parallel stochastic coordinate descent algorithm, ICML, 2014.
>
> [8]I. Necoara, et al. Linear convergence of first order methods for non-strongly convex optimization. Math Program, 2019.
> ***
> **Q3:** ... It would benefit readers if the authors could briefly describe what they mean by "heavy-tailed" early in the paper. ...
>
> **A3:** Your understanding of "heavy-tailed" is correct. "Heavy-tailed" is equivalent to "long-tailed". In this paper, we use a special heavy-tailed distribution, i.e., sub-Weibull distribution. We have briefly described the meaning of this distribution following Definition 2.
> ***
> **Q4:** There are minor grammatical errors that need to be corrected.
>
> **A4:** Thanks. We have carefully checked the whole manuscript and correct all grammatical errors. For example, we have corrected Equation (5) to “$\mathbb{E}[F(A(S))-F(w^*)]\leq\mathbb{E}[F(A(S))-F_S(A(S))]+\mathbb{E}[F_S(A(S))-F_S(w(S))]$”.

---

> > ### Comment · Reviewer_hpvH · 2023-08-11
> > **Thank you for your rebuttal**
> >
> > Thank you for your detailed responses.

---

> > > ### Author Response · Authors · 2023-08-17
> > >
> > > Thank you very much for your recognition of our work.

---

### Official Review · Reviewer_QoKb · 2023-07-26

**Soundness:** 3 good
**Presentation:** 3 good
**Contribution:** 3 good
**Rating:** 6
**Confidence:** 3

**Summary:**

This paper studies the generalization and optimization analysis of the Federated zeroth-order optimization (FedZO) algorithm. It develops tailored techniques for the federated setting to establish generalization bounds. One bound relies on Lipschitz continuity, and the second removes this dependency. The optimization error in the order of $\mathcal{O}(1/T)$ is also developed for FedZO. The authors extend results to the asynchronous federated setting, where all the workers participate throughout the update process and asynchrony may cause inconsistency in local workers within the same iteration.

**Strengths:**

The paper is well-crafted, and the authors effectively elucidate the distinctions from prior methods.

The theoretical results are quite comprehensive. Existing techniques in the literature may not be directly applicable to the FedZO algorithm. The authors fill in this gap by first developing error decomposition and estimation techniques and then presenting the first algorithmic stability-based generalization analysis for FedZO.


**Weaknesses:**

The paper demonstrates a thorough analysis of the existing federated learning algorithm, FedZO, which is commendable. However, to further enhance its novelty and impact, it would have been valuable if the authors had introduced a new algorithm and conducted a comparative analysis against FedZO.

Furthermore, to gain deeper insights into the analysis, it is recommended to include empirical evaluations exploring the effects of different parameters on the bounds. This empirical investigation would provide valuable practical implications and strengthen the overall findings.

**Questions:**

Unlike other existing papers, all the findings in this paper rely on the $\theta$ tail parameter of the Sub-Weibull distribution. The extent of this dependence's impact is uncertain in practical settings, and it would be helpful if the authors could provide insights or comments on this matter.

---

> ### Author Rebuttal · Authors · 2023-08-08
>
> **Q1:** ... introduced a new algorithm and conducted a comparative analysis against FedZO.
>
> **A1:** Thanks for your constructive comments. Our major contribution is exploring the theoretical generalization upper bounds of federated zeroth-order optimization algorithm which seem to be a gap, especially for stability-based analysis. Not limited to FedZO [1], we extend FedZO to the asynchronous version and give a similar theoretical generalization bound. From the theoretical perspective, the bounds of asynchronous FedZO can recover the bounds of FedZO, which indicates this extension is theoretically reasonable. From the practical perspective, there are many related work [2][3] validating that the asynchronous strategy can take full advantage of the clients computation capabilities. To further explicitly show its effectiveness, some necessary experiments (similar to the ones of synchronous FedZO [1]) will be conducted later. We have used “global response” to provide Figure 1 to show the structure of asynchronous FedZO.
>
> [1]W. Fang, et al. Communication-efficient stochastic zeroth-order optimization for federated learning. TSP, 2022.
>
> [2]C. Xu, et al. Asynchronous federated learning on heterogeneous devices: A survey. arXiv, 2021.
>
> [3]A. Koloskova, et al. Decentralized stochastic optimization and gossip algorithms with compressed communication. ICML, 2019.
> ***
> **Q2:** ... it is recommended to include empirical evaluations exploring the effects of different parameters on the bounds ...
>
> **A2:** Thanks. Our theoretical results are consistent with the relevant experiments in [4]. For example, our generalization bounds indeed have negative dependence on the number M of selected clients at each iteration (see lines 41, 56 in Appendix) which is presented in Figures 1(b) and 4 of [4]. Our optimization bounds (Theorems 4 and 6) are also negatively dependent on the total iteration number $T$, which is presented in all figures of [4]. As mentioned in A1, we will provide more empirical evaluations.
>
> [4]W. Fang, et al. Communication-efficient stochastic zeroth-order optimization for federated learning. TSP, 2022.
> ***
> **Q3:** ... The extent of the impact of the dependence on $\theta$ is uncertain in practical settings ...
>
> **A3:** Some previous work indicated that the distribution of gradient noise of SGD exhibits heavier tail than sub-Gaussian (e.g., sub-Weibull [5][6][7]). In our paper, the distribution of gradient noise of each local client is equivalent to the one of SGD with the same model. Thus it is reasonable to assume that the real first order gradient noise for each local client is a sub-Weibull random vector (Assumption 2). It should be noted that the purpose of considering the heavy-tail condition is to provide the theoretical impact of heavy-tailed phenomena on the generalization performance of the FedZO algorithm. As your mentioned, due to the unknown gradient information, the dependence on the heavy-tail parameters $\theta$ and $K$ is uncertain for practice yet. Luckily, our generalization bounds indicate that, in the federated zeroth-order optimization, the degeneration of generalization performance caused by the heavy-tailed phenomena is mild, which is similar to some previous theoretical results (e.g., Theorems 3.3, 3.9 in [5]). We have provided some sub-Weibull survival curves with varying tail parameters $\theta$, which inspired by [8], via “global response” (Figure 2).
>
> [5]S. Li, Y. Liu. High probability guarantees for nonconvex stochastic gradient descent with heavy tails. ICML, 2022.
>
> [6]L. Madden, et al. Highprobability convergence bounds for non-convex stochastic gradient descent. arXiv, 2020.
>
> [7] M. Gurbuzbalaban, et al. The heavy-tail phenomenon in sgd. ICML, 2021.
>
> [8]M. Vladimirova, et al. Sub-weibull distributions: Generalizing sub-gaussian and sub-exponential properties to heavier tailed distributions. Stat, 2020.

---

> > ### Comment · Reviewer_QoKb · 2023-08-15
> >
> > Your responses are greatly appreciated. Taking into account the responses, I will uphold my current score.

---

> > > ### Author Response · Authors · 2023-08-17
> > >
> > > Thank you very much for your recognition of our work.

---

### Official Review · Reviewer_xnUD · 2023-07-27

**Soundness:** 2 fair
**Presentation:** 2 fair
**Contribution:** 2 fair
**Rating:** 5
**Confidence:** 3

**Summary:**

This paper provides theoretical guarantees for both synchronous and asynchronous FedZO algorithms. It first establishes a generalization error bound of FedZO under conventional assumptions. Then, the bounds are further improved by using the second-order Taylor expansion and heavy-tailed gradient noise. The theoretical results seem promising; however, some technical concerns remain unresolved.

**Strengths:**

This paper focuses on providing theoretical guarantees for the newly proposed FedZO algorithms. The mathematical tools used in the analysis are relatively advanced in recent literature. In addition to analyzing the original FedZO algorithm, this paper also extends its analysis to include asynchronous FedZO.

**Weaknesses:**

This paper primarily focuses on providing further theoretical guarantees for a recently proposed algorithm, which addresses a minor problem. As a result, the contributions and impacts to the federated learning community are limited. Furthermore, some assumptions and results are not well-justified, as discussed below.

[Related to federated learning]

In federated optimization literature, heterogeneous data distribution across clients is a significant challenge. In the original FedZO paper [16], Assumptions 3 and 4 were used to describe the impacts of such heterogeneity. However, this paper seems to have overlooked this necessary aspect in the analysis, which could significantly affect the proof of Theorem 2.

[Experiment]

This paper introduces asynchronous FedZO and provides learning guarantees compared to FedZO. However, the original FedZO does not evaluate the asynchronous version. It is crucial to include experimental evaluations of both synchronous FedZO and asynchronous FedZO in this paper to support the proposed theories.

[Organization]

One key aspect of this paper is the approximation of the first-order gradient using a second-order Taylor expansion and Assumption 2. However, the details of these methods are missing in the main paper, making it challenging to follow the arguments.

**Questions:**

[Comparison of generalization bounds]

This paper has mentioned, "Due to the essential difference between FedZO and ZoSS, the previous analysis technique in [36] cannot be used for federated learning directly." So, why are the generalization bounds comparable with ZoSS in Table 1? It seems more reasonable to compare them with distributed zero-order optimization.

[Comparison of optimization bounds]

A vital challenge of zero-order optimization is the dimension issue of model parameters, as shown in Table 1 in FedZO. In this paper, how does it remove the dimension $d$ from the optimization bound? Is this improvement applicable to arbitrary zero-order optimization?

**Limitations:**

No limitations are discussed in the main paper.

---

> ### Author Rebuttal · Authors · 2023-08-08
>
> **Q1:** ... overlooked Assumptions 3 and 4 of [1] ...
>
> **A1:** Thanks for your constructive comments. In the original FedZO [1], the reason for making Assumptions 3 and 4 is to connect the gradient of local empirical loss and the gradient of global population risk. The gradient of population risk characterizes the convergence of the FedZO algorithm which is one of the major theoretical contributions of [1].
>
> Compared with [1], we have the following two reasons why we didn’t make such assumptions.
>
> (1)Theorem 2: We can directly bound the gradient of local empirical loss with the Lipschitz parameter $L$ (see line 42 in Appendix).
>
> (2)Theorems 3, 4, 5, 6: We consider the gradients of all clients simultaneously (see Equations (8) and (13) in lines 60 and 90 of Appendix) rather than the one of single client like Assumption 4 in [1]. After careful checking, we find some mistakes in (8) and (13) (e.g., the decomposition strategy of norm). Luckily, they don’t make a difference in the order of our results. We have corrected them in our new manuscript.
>
> Following your comment, we will try to improve our results by further considering the impact of such heterogeneity.
>
> [1]W. Fang, et al. Communication-efficient stochastic zeroth-order optimization for federated learning. TSP, 2022.
> ***
> **Q2:** ... experimental evaluations...
>
> **A2:** Thanks. Some relevant experiments on FedZO had been provided in [2]. For the asynchronous version, many related work validates that the asynchronous strategy can take full advantage of the clients computation capabilities [3][4].
>
> Filling the gap of theoretical generalization analysis is crucial for the development of federated zeroth-order algorithm, which is the major contribution in this paper. Our results conform to these empirical behaviors from both generalization and optimization perspectives which are listed as follows.
>
> **Generalization:** Our generalization bounds (Theorems 2, 3) are negatively correlated with the number M of selected clients at each iteration (the dependence $\frac{1}{M}$ can be seen in lines 41 and 56 in Appendix). It is consistent with Figures 1(b) and 4 of [2].
>
> **Optimization:** Our optimization bounds (Theorems 4 and 6) are negatively correlated with the total iteration number $T$, which is presented in all figures of [2].
>
> To further give empirical observation of the asynchronous version, some necessary experiments (similar to the ones of synchronous FedZO [2]) will be conducted later. We have utilized “global response” to provide Figure 1 to show the structure of asynchronous FedZO.
>
> [2]W. Fang, et al. Communication-efficient stochastic zeroth-order optimization for federated learning. TSP, 2022.
>
> [3]X. Lian, et al. Asynchronous decentralized parallel stochastic gradient descent. ICML, 2018.
>
> [4]X. Lian, Y. Huang, Y. Li, and J. Liu. Asynchronous parallel stochastic gradient for nonconvex optimization. NIPS, 2015.
> ***
> **Q3:** ... Taylor expansion and Assumption 2 are missing in the main paper ...
>
> **A3:** Thanks. As shown in line 36 of Appendix, the unknown gradient is estimated by the second-order Taylor expansion. Following your valuable comment, we have supplemented the detailed steps of Taylor expansion in line 100 of the main paper. As for Assumption 2, we utilize some properties of sub-Weibull distribution to bound the gradient noise after estimating the gradient. We have also added a related remark following Assumption 2 to improve the readability of our paper. Considering the reader who is not familiar with this distribution, we have provided some sub-Weibull survival curves with varying tail parameters $\theta$ inspired by [5], via “global response” (Figure 2).
>
> [5]M. Vladimirova, et al. Sub-weibull distributions: Generalizing sub-gaussian and sub-exponential properties to heavier tailed distributions. Stat, 2020.
> ***
> **Q4:** ...why comparable with ZoSS rather than distributed zero-order optimization?
>
> **A4:** Table 1 shows the comparisons of stability-based generalization bounds. The reason to compare with ZoSS is that it is the only one using stability tools to analyze the theoretical generalization performance for zeroth-order optimization algorithm. Indeed, as you mentioned, it is reasonable to make comparisons with distributed zero-order algorithms. Thus, we have added some comparisons about optimization bound, e.g., [6][7], in Table 2. Limited by the length of Rebuttal, parts of these new comparisons are provided in Table 3 of “global response”.
>
> [6]E. Kaya, et al. Communication-efficient zeroth-order distributed online optimization: Algorithm, theory, and applications. Access, 2023.
>
> [7]X. Yi, et al. Zerothorder algorithms for stochastic distributed nonconvex optimization. Automatica, 2022.
> ***
> **Q5:** ... how remove $d$ from the optimization bound? Is this improvement applicable to arbitrary zero-order optimization?
>
> **A5:** For the optimization bound, the common technology is dimension related tool (e.g., uniform convergence) which is the major reason why removing the dimension $d$ is challenging. Different from this technology, the key of our proof is to directly build the iteration sequence $t(t-c)\mathcal{E}[F_S(w^{t+1})-F_S(w(S))]$ (c denotes a constant) via the smoothness assumption, the PL condition and the step size setting $\eta_t=\eta_1/(\alpha(t+a))$. After iterative computation, we can obtain the optimization bound $\mathcal{E}[F_S(w^T)-F_S(w(S))]$. Our proof can’t be applied to the zeroth-order algorithm which is without these settings. Note that, our optimization bounds rely on the quality of initial model like many previous work (e.g., [8][9]). We have added a remark behind Theorem 4 to make the above statement.
>
> [8]W. Fang, et al. Communication-efficient stochastic zeroth-order optimization for federated learning. TSP, 2022.
>
> [9]H. Yu, et al. Parallel restarted SGD with faster convergence and less communication: Demystifying why model averaging works for deep learning. AAAI, 2019.

---

> > ### Comment · Reviewer_xnUD · 2023-08-17
> >
> > Thank you for the response. Most of my concerns have been addressed.   I can improve the score. Given that data heterogeneity stands out as a crucial assumption, I would greatly appreciate it if the authors could conduct a thorough and comprehensive analysis of this aspect, as they have also mentioned in their rebuttal.

---

> > > ### Author Response · Authors · 2023-08-17
> > >
> > > Thank you very much for your constructive comments and recognition of our work. Our current results may be refined by further considering data heterogeneity which measures the dissimilarity among the data of different clients. As mentioned in our rebuttal, we will try to conduct a thorough and comprehensive analysis of this aspect.

---

### Author Rebuttal · Authors · 2023-08-09

Thanks for the comments of all reviewers. Considering the limitation of character count, we provide two figures and three tables in "global response".

Figure 1 denotes the structure of the asynchronous FedZO algorithm.

Figure 2 denotes some sub-Weibull survival curves with varying tail parameters $\theta$ inspired by [3].

Table 1 denotes some new comparisons with the stability-based generalization bounds of asynchronous optimization algorithms [1].

Table 2 denotes some new comparisons with the optimization bounds of asynchronous optimization algorithms [1][2].

Table 3 denotes some new comparisons with the optimization bounds of distributed zero-order optimization algorithms [6][7].

**Table 1**
| | | | | | | |
|:----:|:----:|:----:|:----:|:----:|:----:|:----:|
|**Algorithm+Reference**|**Generalization bound**|**Tool**|**L**|**$\theta$**|**$v^2$**|**B.**|
|**AD-SGD**[1]|$\mathcal{O}(\frac{n-\lambda}{n(1-\lambda)}(1+\frac{\beta \eta_1 }{M})^T)$|Uni.|$\surd$|$\times$|$\times$|$\times$|
|**AD-SGD**[1]|$\mathcal{O}(\frac{nM-\lambda}{n(1-\lambda)}L^2T)$|Uni.|$\surd$|$\times$|$\times$|$\times$|
| | | | | | | |

**Table 2**
| | | | | | | | |
|:----:|:----:|:----:|:----:|:----:|:----:|:----:|:----:|
|**Algorithm+Reference**|**Optimization bound**|**Step size**|**L**|**$\theta$**|**$\beta$**|**B.**|**$\sigma$**|
|**AD-SGD**[1]|$\mathcal{O}((r+\frac{C_{\lambda}}{\lambda^{t_0}}+\frac{t_0}{M})(\log T)^{-1})$|$\eta_t=\mathcal{O}(\frac{Mc}{t+1})$|$\surd$|$\times$|$\surd$|$\times$|$\times$|
|**AD-PSGD**[2]|$\mathcal{O}(T^{-1/2})$|$\eta_t=\mathcal{O}(\frac{n}{b_1(\sqrt{T}+1)})$|$\times$|$\times$|$\surd$|$\times$|$\surd$|
| | | | | | | | |

**Table 3**
| | | | | | | | |
|:----:|:----:|:----:|:----:|:----:|:----:|:----:|:----:|
|**Algorithm+Reference**|**Optimization bound**|**Step size**|**L**|**$\theta$**|**$\beta$**|**B.**|**$\sigma$**|
|**EF-ZO-SGD**[6]|$\mathcal{O}((d/T)^{1/2} + d/T)$|$\eta_t=\mathcal{O}(1/\sqrt{dT})$|$\surd$|$\times$|$\surd$|$\times$|$\times$|
|**FED-EF-ZO-SGD**[6]|$\mathcal{O}((d/T)^{1/2} + (d/T)^{3/2})$|$\eta_t=\mathcal{O}(1/\sqrt{dT})$|$\surd$|$\times$|$\surd$|$\times$|$\times$|
|**Distributed ZO Primal–Dual**[7]|$\mathcal{O}((d/(MT))^{1/2})$|$\eta_t=\mathcal{O}(d^{-1/2}(t+d^{1/(2\theta)})^{-\theta})$|$\times$|$\times$|$\surd$|$\times$|$\surd$|
| | | | | | | | |

[1]X. Deng, et al. Stability-based generalization analysis of the asynchronous decentralized SGD. AAAI, 2023.

[2]X. Lian, et al. Asynchronous decentralized parallel stochastic gradient descent. ICML, 2018.

[3]M. Vladimirova, et al. Sub-weibull distributions: Generalizing sub-gaussian and sub-exponential properties to heavier tailed distributions. Stat, 2020.

[6]E. Kaya, et al. Communication-efficient zeroth-order distributed online optimization: Algorithm, theory, and applications. Access, 2023.

[7]X. Yi, et al. Zerothorder algorithms for stochastic distributed nonconvex optimization. Automatica, 2022.

---

### Decision · Program_Chairs · 2023-09-21

**Decision:**

Accept (poster)

**Comment:**

This paper presents a fine-grained theoretical analysis of federated zeroth-order optimization techniques. The discussion phase improves everyone's understanding about this paper and we hope the authors revise it according to the reviewers' suggestion for the final version.